

# In-situ estimation of ice crystal properties at the South Pole using LED calibration data from the IceCube Neutrino Observatory

Rasha Abbasi[17,*], Markus Ackermann[63,*], Jenni Adams[18,*], Nakul Aggarwal[25,*], Juanan Aguilar[12,*], Markus Ahlers[22,*], Maryon Ahrens[53,*], Jean-Marco Alameddine[23,*], Antonio Augusto Alves Junior[31,*], Najia Moureen Binte Amin[43,*], Karen Andeen[41,*], Tyler Anderson[59,60,*], Gisela Anton[26,*], Carlos Argüelles[14,*], Yosuke Ashida[39,*], Sofia Athanasiadou[63,*], Spencer Axani[15,*], Xinhua Bai[49,*], Aswathi Balagopal V[39,*], Moreno Baricevic[39,*], Steve Barwick[30,*], Vedant Basu[39,*], Ryan Bay[8,*], James Beatty[20,21,*], Karl Heinz Becker[62,*], Julia Becker Tjus[11,*], Jakob Beise[61,*], Chiara Bellenghi[27,*], Samuel Benda[39,*], Segev BenZvi[51,*], David Berley[19,*], Elisa Bernardini[47,*], Dave Besson[34,*], Gary Binder[8,9,*], Daniel Bindig[62,*], Erik Blaufuss[19,*], Summer Blot[63,*], Federico Bontempo[31,*], Julia Book[14,*], Jürgen Borowka[1,*], Caterina Boscolo Meneguolo[47,*], Sebastian Böser[40,*], Olga Botner[61,*], Jakob Böttcher[1,*], Etienne Bourbeau[22,*], Jim Braun[39,*], Bennett Brinson[6,*], Jannes Brostean-Kaiser[63,*], Ryan Burley[2,*], Raffaela Busse[42,*], Michael Campana[48,*], Erin Carnie-Bronca[2,*], Chujie Chen[6,*], Zheyang Chen[54,*], Dmitry Chirkin[39,*], Koun Choi[55,*], Brian Clark[24,*], Lew Classen[42,*], Alan Coleman[43,*], Gabriel Collin[15,*], Amy Connolly[20,21,*], Janet Conrad[15,*], Paul Coppin[13,*], Pablo Correa[13,*], Stefan Countryman[45,*], Doug Cowen[59,60,*], Robert Cross[51,*], Christian Dappen[1,*], Pranav Dave[6,*], Catherine De Clercq[13,*], James DeLaunay[58,*], Diyaselis Delgado López[14,*], Hans Dembinski[43,*], Kunal Deoskar[53,*], Abhishek Desai[39,*], Paolo Desiati[39,*], Krijn de Vries[13,*], Gwenhael de Wasseige[36,*], Tyce DeYoung[24,*], Alejandro Diaz[15,*], Juan Carlos Díaz-Vélez[39,*], Markus Dittmer[42,*], Hrvoje Dujmovic[31,*], Michael DuVernois[39,*], Thomas Ehrhardt[40,*], Philipp Eller[27,*], Ralph Engel[31,32,*], Hannah Erpenbeck[1,*], John Evans[19,*], Paul Evenson[43,*], Kwok Lung Fan[19,*], Ali Fazely[7,*], Anatoli Fedynitch[57,*], Nora Feigl[10,*], Sebastian Fiedlschuster[26,*], Aaron Fienberg[60,*], Chad Finley[53,*], Leander Fischer[63,*], Derek Fox[59,*], Anna Franckowiak[11,*], Elizabeth Friedman[19,*], Alexander Fritz[40,*], Philipp Fürst[1,*], Tom Gaisser[43,†,*], Jay Gallagher[38,*], Erik Ganster[1,*], Alfonso Garcia[14,*], Simone Garrappa[63,*], Lisa Gerhardt[9,*], Ava Ghadimi[58,*], Christian Glaser[61,*], Thorsten Glüsenkamp[27,*], Theo Glauch[26,*], Noah Goehlke[32,*], Javier Gonzalez[43,*], Sreetama Goswami[58,*], Darren Grant[24,*], Shannon Gray[19,*], Timothée Grégoire[60,*], Spencer Griswold[51,*], Christoph Günther[1,*], Pascal Gutjahr[23,*], Christian Haack[27,*], Allan Hallgren[61,*], Robert Halliday[24,*], Lasse Halve[1,*], Francis Halzen[39,*], Hassane Hamdaoui[54,*], Martin Ha Minh[27,*], Kael Hanson[39,*], John Hardin[15,39,*], Alexander Harnisch[24,*], Patrick Hatch[33,*], Andreas Haungs[31,*], Klaus Helbing[62,*], Jonas Hellrung[1,*], Felix Henningsen[27,*], Lars Heuermann[1,*], Stephanie Hickford[62,*], Colton Hill[16,*], Gary Hill[2,*], Kara Hoffman[19,*], Kotoyo Hoshina[39,a,*], Wenjie Hou[31,*], Thomas Huber[31,*], Klas Hultqvist[53,*], Mirco Hünnefeld[23,*], Raamis Hussain[39,*], Karolin Hymon[23,*], Seongjin In[55,*], Nadege Iovine[12,*], Aya Ishihara[16,*], Matti Jansson[53,*], George Japaridze[5,*], Minjin Jeong[55,*], Miaochen Jin[14,*], Ben Jones[4,*], Donghwa Kang[31,*], Woosik Kang[55,*], Xinyue Kang[48,*], Alexander Kappes[42,*], David Kappesser[40,*], Leonora Kardum[23,*], Timo Karg[63,*], Martina Karl[27,*], Albrecht Karle[39,*], Uli Katz[26,*], Matt Kauer[39,*], John Kelley[39,*], Ali Kheirandish[60,*], Ken'ichi Kin[16,*], Joanna Kiryluk[54,*], Spencer Klein[8,9,*], Alina Kochocki[24,*], Ramesh Koirala[43,*], Hermann Kolanoski[10,*], Tomas Kontrimas[27,*], Lutz Köpke[40,*],





Claudio Kopper[24,*], Jason Koskinen[22,*], Paras Koundal[31,*], Michael Kovacevich[48,*], Marek Kowalski[10,63,*], Tetiana Kozynets[22,*], Emmett Krupczak[24,*], Emma Kun[11,*], Naoko Kurahashi[48,*], Neha Lad[63,*], Cristina Lagunas Gualda[63,*], Michael Larson[19,*], Frederik Lauber[62,*], Jeffrey Lazar[14,39,*], Jiwoong Lee[55,*], Kayla Leonard[39,*], Agnieszka Leszczyńska[43,*], Massimiliano Lincetto[11,*], Qinrui Liu[39,*], Maria Liubarska[25,*], Elisa Lohfink[40,*], Christina Love[48,*], Cristian Jesus Lozano Mariscal[42,*], Lu Lu[39,*], Francesco Lucarelli[28,*], Andrew Ludwig[24,35,*], William Luszczak[39,*], Yang Lyu[8,9,*], Wing Yan Ma[63,*], Jim Madsen[39,*], Kendall Mahn[24,*], Yuya Makino[39,*], Sarah Mancina[39,*], Wenceslas Marie Sainte[39,*], Ioana Mariş[12,*], Szabolcs Marka[45,*], Zsuzsa Marka[45,*], Matthew Marsee[58,*], Ivan Martinez-Soler[14,*], Reina Maruyama[44,*], Thomas McElroy[25,*], Frank McNally[37,*], James Vincent Mead[22,*], Kevin Meagher[39,*], Sarah Mechbal[63,*], Andres Medina[21,*], Maximilian Meier[16,*], Stephan Meighen-Berger[27,*], Yarno Merckx[13,*], Jessie Micallef[24,*], Daniela Mockler[12,*], Teresa Montaruli[28,*], Roger Moore[25,*], Bob Morse[39,*], Marjon Moulai[39,*], Tista Mukherjee[31,*], Richard Naab[63,*], Ryo Nagai[16,*], Uwe Naumann[62,*], Amid Nayerhoda[47,*], Jannis Necker[63,*], Miriam Neumann[42,*], Hans Niederhausen[24,*], Mehr Nisa[24,*], Sarah Nowicki[24,*], Anna Obertacke Pollmann[62,*], Marie Oehler[31,*], Bob Oeyen[29,*], Alex Olivas[19,*], Rasmus Orsoe[27,*], Jesse Osborn[39,*], Erin O'Sullivan[61,*], Hershal Pandya[43,*], Daria Pankova[60,*], Nahee Park[33,*], Grant Parker[4,*], Ek Narayan Paudel[43,*], Larissa Paul[41,*], Carlos Pérez de los Heros[61,*], Lilly Peters[1,*], Josh Peterson[39,*], Saskia Philippen[1,*], Sarah Pieper[62,*], Alex Pizzuto[39,*], Matthias Plum[49,*], Yuiry Popovych[40,*], Alessio Porcelli[29,*], Maria Prado Rodriguez[39,*], Brandon Pries[24,*], Rachel Procter-Murphy[19,*], Gerald Przybylski[9,*], Christoph Raab[12,*], John Rack-Helleis[40,*], Mohamed Rameez[22,*], Katherine Rawlins[3,*], Zoe Rechav[39,*], Abdul Rehman[43,*], Patrick Reichherzer[11,*], Giovanni Renzi[12,*], Elisa Resconi[27,*], Simeon Reusch[63,*], Wolfgang Rhode[23,*], Mike Richman[48,*], Benedikt Riedel[39,*], Ella Roberts[2,*], Sally Robertson[8,9,*], Steven Rodan[55,*], Gerrit Roellinghoff[55,*], Martin Rongen[40,*], Carsten Rott[52,55,*], Tim Ruhe[23,*], Li Ruohan[27,*], Dirk Ryckbosch[29,*], Devyn Rysewyk Cantu[24,*], Ibrahim Safa[14,39,*], Julian Saffer[32,*], Daniel Salazar-Gallegos[24,*], Pranav Sampathkumar[31,*], Sebastian Sanchez Herrera[24,*], Alexander Sandrock[23,*], Marcos Santander[58,*], Sourav Sarkar[25,*], Subir Sarkar[46,*], Merlin Schaufel[1,*], Harald Schieler[31,*], Sebastian Schindler[26,*], Berit Schlüter[42,*], Torsten Schmidt[19,*], Judith Schneider[26,*], Frank Schröder[31,43,*], Lisa Schumacher[27,*], Georg Schwefer[1,*], Steve Sclafani[48,*], Dave Seckel[43,*], Surujhdeo Seunarine[50,*], Ankur Sharma[61,*], Shefali Shefali[32,*], Nobuhiro Shimizu[16,*], Manuel Silva[39,*], Barbara Skrzypek[14,*], Ben Smithers[4,*], Robert Snihur[39,*], Jan Soedingrekso[23,*], Andreas Søgaard[22,*], Dennis Soldin[32,*], Christian Spannfellner[27,*], Glenn Spiczak[50,*], Christian Spiering[63,*], Michael Stamatikos[21,*], Todor Stanev[43,*], Robert Stein[63,*], Thorsten Stezelberger[9,*], Timo Stürwald[62,*], Thomas Stuttard[22,*], Greg Sullivan[19,*], Ignacio Taboada[6,*], Samvel Ter-Antonyan[7,*], Will Thompson[14,*], Jessie Thwaites[39,*], Serap Tilav[43,*], Kirsten Tollefson[24,*], Christoph Tönnis[56,*], Simona Toscano[12,*], Delia Tosi[39,*], Alexander Trettin[63,*], Chun Fai Tung[6,*], Roxanne Turcotte[31,*], Jean Pierre Twagirayezu[24,*], Bunheng Ty[39,*], Martin Unland Elorrieta[42,*], Karriem Upshaw[7,*], Nora Valtonen-Mattila[61,*], Justin Vandenbroucke[39,*], Nick van Eijndhoven[13,*], David Vannerom[15,*], Jakob van Santen[63,*], Javi Vara[42,*], Joshua Veitch-Michaelis[39,*], Stef Verpoest[29,*], Doga Veske[45,*], Christian Walck[53,*], Winnie Wang[39,*], Timothy Blake Watson[4,*], Chris Weaver[24,*], Philip Weigel[15,*], Andreas Weindl[31,*], Jan Weldert[40,*], Chris Wendt[39,*], Johannes Werthebach[23,*], Mark Weyrauch[31,*], Nathan Whitehorn[24,35,*], Christopher Wiebusch[1,*], Nathan Willey[24,*], Dawn Williams[58,*], Martin Wolf[39,*], Gerrit Wrede[26,*], Johan Wulff[11,*], Xianwu Xu[7,*], Juan



Pablo Yanez[25,*], Emre Yildizci[39,*], Shigeru Yoshida[16,*], Shiqi Yu[24,*], Tianlu Yuan[39,*], Zelong Zhang[54,*], and Pavel Zhelnin[14,*]

[1]III. Physikalisches Institut, RWTH Aachen University, D-52056 Aachen, Germany
[2]Department of Physics, University of Adelaide, Adelaide, 5005, Australia
[3]Dept. of Physics and Astronomy, University of Alaska Anchorage, 3211 Providence Dr., Anchorage, AK 99508, USA
[4]Dept. of Physics, University of Texas at Arlington, 502 Yates St., Science Hall Rm 108, Box 19059, Arlington, TX 76019, USA
[5]CTSPS, Clark-Atlanta University, Atlanta, GA 30314, USA
[6]School of Physics and Center for Relativistic Astrophysics, Georgia Institute of Technology, Atlanta, GA 30332, USA
[7]Dept. of Physics, Southern University, Baton Rouge, LA 70813, USA
[8]Dept. of Physics, University of California, Berkeley, CA 94720, USA
[9]Lawrence Berkeley National Laboratory, Berkeley, CA 94720, USA
[10]Institut für Physik, Humboldt-Universität zu Berlin, D-12489 Berlin, Germany
[11]Fakultät für Physik & Astronomie, Ruhr-Universität Bochum, D-44780 Bochum, Germany
[12]Université Libre de Bruxelles, Science Faculty CP230, B-1050 Brussels, Belgium
[13]Vrije Universiteit Brussel (VUB), Dienst ELEM, B-1050 Brussels, Belgium
[14]Department of Physics and Laboratory for Particle Physics and Cosmology, Harvard University, Cambridge, MA 02138, USA
[15]Dept. of Physics, Massachusetts Institute of Technology, Cambridge, MA 02139, USA
[16]Dept. of Physics and The International Center for Hadron Astrophysics, Chiba University, Chiba 263-8522, Japan
[17]Department of Physics, Loyola University Chicago, Chicago, IL 60660, USA
[18]Dept. of Physics and Astronomy, University of Canterbury, Private Bag 4800, Christchurch, New Zealand
[19]Dept. of Physics, University of Maryland, College Park, MD 20742, USA
[20]Dept. of Astronomy, Ohio State University, Columbus, OH 43210, USA
[21]Dept. of Physics and Center for Cosmology and Astro-Particle Physics, Ohio State University, Columbus, OH 43210, USA
[22]Niels Bohr Institute, University of Copenhagen, DK-2100 Copenhagen, Denmark
[23]Dept. of Physics, TU Dortmund University, D-44221 Dortmund, Germany
[24]Dept. of Physics and Astronomy, Michigan State University, East Lansing, MI 48824, USA
[25]Dept. of Physics, University of Alberta, Edmonton, Alberta, Canada T6G 2E1
[26]Erlangen Centre for Astroparticle Physics, Friedrich-Alexander-Universität Erlangen-Nürnberg, D-91058 Erlangen, Germany
[27]Physik-department, Technische Universität München, D-85748 Garching, Germany
[28]Département de physique nucléaire et corpusculaire, Université de Genève, CH-1211 Genève, Switzerland
[29]Dept. of Physics and Astronomy, University of Gent, B-9000 Gent, Belgium
[30]Dept. of Physics and Astronomy, University of California, Irvine, CA 92697, USA
[31]Karlsruhe Institute of Technology, Institute for Astroparticle Physics, D-76021 Karlsruhe, Germany
[32]Karlsruhe Institute of Technology, Institute of Experimental Particle Physics, D-76021 Karlsruhe, Germany
[33]Dept. of Physics, Engineering Physics, and Astronomy, Queen's University, Kingston, ON K7L 3N6, Canada
[34]Dept. of Physics and Astronomy, University of Kansas, Lawrence, KS 66045, USA
[35]Department of Physics and Astronomy, UCLA, Los Angeles, CA 90095, USA
[36]Centre for Cosmology, Particle Physics and Phenomenology - CP3, Université catholique de Louvain, Louvain-la-Neuve, Belgium
[37]Department of Physics, Mercer University, Macon, GA 31207-0001, USA
[38]Dept. of Astronomy, University of Wisconsin–Madison, Madison, WI 53706, USA
[39]Dept. of Physics and Wisconsin IceCube Particle Astrophysics Center, University of Wisconsin–Madison, Madison, WI 53706, USA
[40]Institute of Physics, University of Mainz, Staudinger Weg 7, D-55099 Mainz, Germany
[41]Department of Physics, Marquette University, Milwaukee, WI, 53201, USA





[42]Institut für Kernphysik, Westfälische Wilhelms-Universität Münster, D-48149 Münster, Germany

[43]Bartol Research Institute and Dept. of Physics and Astronomy, University of Delaware, Newark, DE 19716, USA

[44]Dept. of Physics, Yale University, New Haven, CT 06520, USA

[45]Columbia Astrophysics and Nevis Laboratories, Columbia University, New York, NY 10027, USA

[46]Dept. of Physics, University of Oxford, Parks Road, Oxford OX1 3PU, UK

[47]Dipartimento di Fisica e Astronomia Galileo Galilei, Università Degli Studi di Padova, 35122 Padova PD, Italy

[48]Dept. of Physics, Drexel University, 3141 Chestnut Street, Philadelphia, PA 19104, USA

[49]Physics Department, South Dakota School of Mines and Technology, Rapid City, SD 57701, USA

[50]Dept. of Physics, University of Wisconsin, River Falls, WI 54022, USA

[51]Dept. of Physics and Astronomy, University of Rochester, Rochester, NY 14627, USA

[52]Department of Physics and Astronomy, University of Utah, Salt Lake City, UT 84112, USA

[53]Oskar Klein Centre and Dept. of Physics, Stockholm University, SE-10691 Stockholm, Sweden

[54]Dept. of Physics and Astronomy, Stony Brook University, Stony Brook, NY 11794-3800, USA

[55]Dept. of Physics, Sungkyunkwan University, Suwon 16419, Korea

[56]Institute of Basic Science, Sungkyunkwan University, Suwon 16419, Korea

[57]Institute of Physics, Academia Sinica, Taipei, 11529, Taiwan

[58]Dept. of Physics and Astronomy, University of Alabama, Tuscaloosa, AL 35487, USA

[59]Dept. of Astronomy and Astrophysics, Pennsylvania State University, University Park, PA 16802, USA

[60]Dept. of Physics, Pennsylvania State University, University Park, PA 16802, USA

[61]Dept. of Physics and Astronomy, Uppsala University, Box 516, S-75120 Uppsala, Sweden

[62]Dept. of Physics, University of Wuppertal, D-42119 Wuppertal, Germany

[63]DESY, D-15738 Zeuthen, Germany

[a]also at Earthquake Research Institute, University of Tokyo, Bunkyo, Tokyo 113-0032, Japan

[†]deceased, 20 February 2022

[*]These authors contributed equally to this work.

**Correspondence:** analysis@icecube.wisc.edu

**Abstract.** The IceCube Neutrino Observatory instruments about 1 km$^3$ of deep, glacial ice at the geographic South Pole using 5160 photomultipliers to detect Cherenkov light emitted by charged relativistic particles. A unexpected light propagation effect observed by the experiment is an anisotropic attenuation, which is aligned with the local flow direction of the ice. Birefringent light propagation has been examined as a possible explanation for this effect. The predictions of a first-principles

5   birefringence model developed for this purpose, in particular curved light trajectories resulting from asymmetric diffusion, provide a qualitatively good match to the main features of the data. This in turn allows us to deduce ice crystal properties. Since the wavelength of the detected light is short compared to the crystal size, these crystal properties do not only include the crystal orientation fabric, but also the average crystal size and shape, as a function of depth. By adding small empirical corrections to this first-principles model, a quantitatively accurate description of the optical properties of the IceCube glacial ice is obtained.

10   In this paper, we present the experimental signature of ice optical anisotropy observed in IceCube LED calibration data, the theory and parametrization of the birefringence effect, the fitting procedures of these parameterizations to experimental data as well as the inferred crystal properties.



# 1 Introduction

The 2021 IPCC report (Masson-Demotte et al., 2021) highlights the need to understand the dynamics of ice sheets in order to
predict their contribution to sea level rise in a changing climate. Ice flows under its own weight, either through basal sliding
or through plastic deformation, which is mediated by the deformations of individual grains as well as interactions between
grains (Cuffey, 2010). The viscosity of an individual ice crystal strongly depends on the direction of the applied strain. As a
hexagonal crystal it will most readily deform as shear is applied orthogonal to the c-axis (crystal symmetry axis, normal to the
hexagonal basal planes), leading to slip of the individual basal planes (McConnel, 1891). As a result ice crystals in a polycrystal
effectively re-orient themselves to minimize resistance when subjected to stress, resulting in a bulk anisotropic viscosity. This
leads to a preferential c-axis distribution, with c-axes orthogonal to the strain (Faria et al., 2014b).

This is experimentally most commonly observed as a crystal orientation fabric through the use of polarized light microscopy
on thin sections of ice core samples (Alley, 1988; Wilson et al., 2003). The aforementioned scenario where all c-axes lie in
a plane is called a girdle fabric (see A for further information), in distinction to other commonly observed fabric states such
as uniform (random directions) or unimodal (aligned in a single direction). Alternatively the impact of the recrystallization
processes on the average crystal size and elongation can also be quantified directly through microscopy (Fitzpatrick et al.,
2014). While ice core analysis delivers ground-truth information, it is limited by its small sampling volume, and often unable
to resolve the absolute direction of fabric orientation as the core orientation is not preserved in the drilling process (Westhoff
et al., 2021). While volumetric quantities such as grain volume and elongation can be directly accessed through tomographic
imaging such as X-ray CT (Linow et al., 2012), these techniques further restrict the sampling volume. Grain size and elongation
evaluated through the imaging of thin slices in turn often strongly depend on the sample plane.

Ice fabric can not only be imaged in ice cores, but also leads to a directionality in the propagation of sound and electro-
magnetic radiation, in principle allowing for remote-sensing of ice properties. The mechanical anisotropy of ice results in a
fabric-dependent speed of sound, as has for example been measured using a sonic logger in boreholes (Kluskiewicz et al.,
2017). Ice crystals also are a birefringent material, such that any incoming electromagnetic radiation is separated into an or-
dinary and extra-ordinary ray of perpendicular polarizations with respect to the c-axis, and which propagate with different
refractive indices. This is today primarily employed by polarimetric radar systems to infer fabric properties (Fujita et al., 2006;
Matsuoka et al., 2003; Jordan et al., 2019; Young et al., 2020) through periodic power anomalies detected as a result of the
direction and polarization-dependent delay in the propagation of radio waves.

Recently, as part of ice calibration measurements for the IceCube Neutrino Observatory (Aartsen et al., 2017), Chirkin
(2013d) described the observation of an optical anisotropy. At receivers 125 m away from isotropic emitters, about twice as
much light is observed for emitter-receiver-pairs oriented along the glacial flow axis versus orthogonal to the flow axis. The
effect was originally modelled as a direction-dependent modification to Mie scattering quantities, either through a modification
of the scattering function as proposed by Chirkin (2013d) or through the introduction of a direction-dependent absorption as
introduced by Rongen (2019). As also shown by Rongen (2019), both parameterizations lack a thorough theoretical justification
and resulted in an incomplete description of the IceCube data.





The wavelength of ∼400 nm employed in the IceCube calibration studies is significantly smaller than the average grain size. Thus, grain boundary properties must be accounted for in addition to the fabric, such that the effect is challenging to derive from first principles. First attempts have been made by Chirkin and Rongen (2020) by attributing the optical anisotropy to

the cumulative diffusion that a beam of light experiences as it is refracted or reflected on many grain boundary crossings in a birefringent polycrystal with a preferential c-axis distribution.

In this scenario the diffusion is found to be strongest when photons initially propagate along the ice flow axis and smallest when initially propagating orthogonal to the flow axis. In addition photons are, on average, deflected towards the flow axis. The deflection per unit distance increases for stronger girdle fabrics, a larger average crystal elongation or a smaller average

crystal size. For crystal realizations where the deflection outweighs the additional diffusion along the flow axis compared to the diffusion along the orthogonal direction, the photon flux along the flow axis will increase with distance compared to the photon flux along the orthogonal axis. This interplay between diffusion and deflection leaves a unique imprint in the spatial and temporal light signatures recorded by IceCube. Comparing these to simulations assuming different ice crystal realizations in turn allows to partially constrain the crystal fabric, size and elongation.

Work on this model has so far been performed and proceedings published (Chirkin, 2013d; Chirkin and Rongen, 2020; Rongen et al., 2021a) in the context of detector calibration for the measurements performed by IceCube. With this paper we will for the first time summarize the full extent of past and ongoing modeling of the ice optical anisotropy to a geophysical audience. The described measurements may be unique to IceCube and thus not easily adopted as a tool in glaciology. Nevertheless we believe that they yield an interesting complementary view on ice physical properties and through comparison to ice core data,

in particular from SPC14 (Casey et al., 2014) drilled ∼1 km from the IceCube array, will be informative to the modeling of ice dynamics.

This manuscript has the following structure: Section 2 introduces the IceCube Neutrino Observatory and how it employs ice as a detection medium. Section 3 describes the properties of the LED calibration data used in this study, explains the software used to generate simulated data and details the likelihood analysis comparing simulated to experimental data in order to infer

ice properties. The experimental signature of the ice optical anisotropy as well as early modeling attempts are summarized in section 4. Section 5 explains the electromagnetic theory governing the birefringence in polycrystals and introduces a software package to simulate the resulting diffusion patterns. Section 6 explains how these diffusion patterns are applied in the IceCube simulation and how crystal properties have been inferred. Section 7 describes the resulting ice model. Section 8 discusses future measurements in upcoming IceCube extensions and through drill-hole logging.

## 2 The IceCube Neutrino Observatory

### 2.1 Scientific context: Neutrino Astronomy

The IceCube Neutrino Observatory operates within the context of astroparticle physics and multi-messenger astronomy. While astronomy is most commonly associated with the observation of the universe in visible light, today the entire electromagnetic spectrum ranging from radio waves to hard X-rays and ultra-high-energy gamma rays is exploited, with each spectral range





giving a complementary insight. Infrared radiation for example is only weakly attenuated by interstellar dust (Li and Draine, 2001) allowing for the imaging of objects obscured by dust clouds.

In addition to photons, the quanta of light, other stable messenger particles are also observed. Most prominently cosmic rays, primarily protons, have now been found at energies exceeding $5 \cdot 10^{19}$ eV, the equivalent of roughly 8 Joules, per particle (Aab et al., 2020). While these ultra-high-energy cosmic rays offer the promise to probe the highest-energy processes in the universe,

they are deflected by magnetic fields along their journey from source to detection (Aartsen et al., 2015). Thus their arrival directions at Earth cannot be easily traced to their origins, making the identification of the sources of high-energy cosmic rays one of the biggest challenges in astroparticle physics.

Associated with the production of high-energy protons, one expects the production of high-energy neutrinos, or astrophysical neutrinos (Margolis et al., 1978). These are electrically neutral, elementary particles belonging to the family of leptons (as

counterparts to electrons, muons and taus). As they are electrically neutral, they are not deflected in magnetic fields and thus point back to their point of origin. Additionally they only interact through the weak force and as a result can traverse vast astronomical distances without their flux being significantly attenuated. While these properties ensure that neutrinos carry unbiased information about the highest-energy regions of the universe, these same properties also make them exceptionally hard to detect, requiring cubic-kilometer scale detectors to intercept a few dozen astrophysical neutrinos per year (Markov,

1960). Detectors of this scale can only be built into natural media such as ocean water or glacial ice, which need to be characterized in-situ as for example presented here.

## 2.2  The detector

The IceCube Neutrino Observatory (Aartsen et al., 2017) has, among other science goals, been built to explore the cosmos using high-energy astrophysical neutrinos. Located about 1 km from the geographic South Pole, it is logistically supported by

the Amundsen–Scott South Pole Station. IceCube features a surface detector, called IceTop, as well as the deep in-ice array of interest here, which consists of 5160 optical sensors instrumenting a one cubic-kilometer volume of ice at depths of 1450 m to 2450 m. The instrumentation layout is shown in Figure 1. Each sensor, called a Digital Optical Module (DOM, see Figure 2) (Stokstad, 2005; Abbasi and others, 2010; Abbasi et al., 2009), is equipped with a 10-inch photomultiplier tube sensitive to light between approximately 300-600 nm and all required readout electronics to be able to time-stamp the arrival time of

individual photons to within two nanoseconds. It addition each DOM features 12 LEDs which can emit light pulses of known intensity and duration into the ice and which are used to calibrate the optical properties of the instrumented ice, as detailed in this paper. Construction took six years, with 86 holes of 60 cm diameter being drilled using hot water drilling (Benson et al., 2014). Cables called "strings" were instrumented with 60 DOMs each and deployed in the boreholes.

The top 1450 m were left without instrumentation because of the strongly scattering ice that exists here above the bubble-to-

air-hydrate transition, which takes place at roughly 1350 m as determined by a predecessor experiment, AMANDA (Ackermann et al., 2006).

Upon a neutrino interaction in the ice, charged particles with relativistic velocities are created, which emit blue light along their path through a process called Cherenkov radiation (Cherenkov, 1937). A small fraction of this light, after propagating



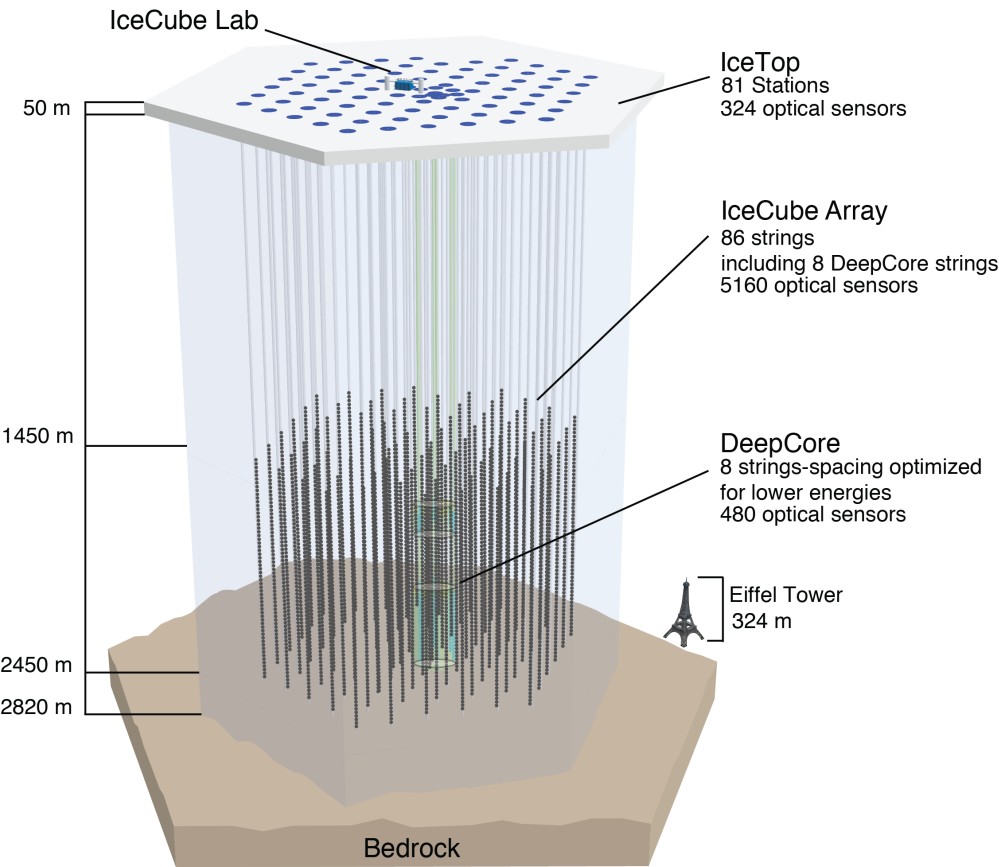

**Figure 1.** Overview of the IceCube detector. The 86 cables of the deep in-ice array, called strings, are indicated as gray lines, with black dots for the 60 DOMs per string. The lateral spacing of the strings is ∼ 125 m with a vertical spacing between DOMs of 17 m. A central part of the array, called DeepCore, is more densely istrumented. The detector is capped by a surface detector, aimed at cosmic ray physics, called IceTop. Figure Credit: IceCube

through the ice, reaches some of the sensors and is detected. Reconstruction of the particle properties, namely energy and

direction, relies on a precise understanding of the optical properties of the instrumented ice. Generally the particle energy is proportional to the amount of detected light, while the arrival direction is inferred from the geometric deposition of the light as well as its timing information (Aartsen et al., 2013b).

     Since its completion in 2010, the IceCube detector has been in continuous operation with an up-time exceeding 99%. On average around 2000 particle events are detected and reconstructed per second, with the vast majority of these being particle

showers induced by cosmic rays striking Earth's atmosphere, and only a vanishing fraction (approximately hundreds per year) being astrophysical neutrinos. Using IceCube data, a wide range of results have been obtained. Those include, among others, the discovery of a high-energy astrophysical neutrino flux (Aartsen et al., 2014) and first associations of high-energy neutrinos





to astrophysical objects (Aartsen et al., 2018b), competitive measurements of neutrino oscillation parameters (Aartsen et al., 2018a) and world-leading limits on possible dark-matter properties (Albert et al., 2020).

## 2.3 Glacial ice as an optical medium

IceCube detects individual photons that are produced through Cherenkov radiation or as emitted by the calibration LEDs. On their way from their source to a potential detection at a DOM these photons are subject to absorption and scattering in the ice, shaping both the intensity pattern in the detector as well the arrival time distributions on every module.

Absorption is characterized by a wavelength $\lambda$ dependent absorption length $\lambda_a(\lambda)$, the propagated distance at which the survival probability of a photon drops to $1/e$. In contrast, scattering does not reduce the photon count, but results in discrete direction changes at an average distance of $\lambda_b(\lambda)$, the geometric scattering length. Scattering is further described by the scattering function, a probability density distribution describing the probability of deflection angles in each scattering process. Neglecting its functional form, the scattering function is described through the average deflection angle or asymmetry parameter $g = \langle \cos\theta \rangle$. The effective scattering length $\lambda_{eff}$, denoting distance at which an initially directional beam becomes diffuse independent of the scattering function, is given as (Aartsen et al., 2013d)

$$\lambda_{eff}(\lambda) = \lambda_b(\lambda)/\left[1 - g(\lambda)\right]. \tag{1}$$

As ice itself is only very weakly absorbing (Warren and Brandt, 2008) (and as we will see later also effectively weakly scattering), the light propagation is dominated by Mie scattering on impurities. In this scenario absorption and scattering strengths are commonly denoted by coefficients ($a = 1/\lambda_a$ and $b_e = 1/\lambda_{eff}$), which are proportional to the impurity concentration. The impurity constituents are believed by He and Price (1998) to be dominated by mineral dust, marine salt and acid droplets as well as (volcanic) soot. These constituents range from nanometer to micrometer in size, with their combined size distribution resulting in a very strong forward scattering with $g \approx 0.95$ at the relevant wavelengths around $400\,\mathrm{nm}$ (He and Price, 1998). The impurities have been deposited with the snow precipitation over the past $100\,\mathrm{ka}$, which was compressed into the ice that is present today at the relevant depths. The impurity composition and concentration, and thus also the optical properties, accordingly trace the global climatological conditions. This stratigraphy was traced at millimeter resolution using a laser dust logger deployed down seven IceCube drill holes as described by Aartsen et al. (2013a).

The detailed stratigraphy associated with the yearly layering cannot be constrained through IceCube data, nor is it needed in order to accurately describe the photon propagation over large distances exceeding tens of meters. Instead, average properties in $10\,\mathrm{m}$ depth increments, here called "ice layers", are being considered. Each layer is described by its dust-induced absorption and scattering coefficients at a wavelength of $400\,\mathrm{nm}$. These are scaled to other wavelengths and the total absorption coefficients including the ice intrinsic absorption in the infrared $A_{IR}$ as described by Aartsen et al. (2013d):

$$a(\lambda) = a_{dust}(\lambda) + A_{IR}e^{-\lambda_0/\lambda} \cdot (1 + 0.01 \cdot \delta\tau) \tag{2}$$

with

$$a_{dust}(\lambda) = a_{dust}(400\,\mathrm{nm}) \cdot \left(\frac{\lambda}{400\,\mathrm{nm}}\right)^{-\kappa} \tag{3}$$





for the absorption coefficients, with tabulated temperature differences $\delta\tau$ compared to the temperature at $1730\,\mathrm{m}$ depth as measured by Price et al. (2002), and

$$b_e(\lambda) = b_e(400\,\mathrm{nm}) \cdot \left(\frac{\lambda}{400\,\mathrm{nm}}\right)^{-\alpha} \tag{4}$$

for the effective scattering coefficients.

While all parameters are in principle depth dependent, e.g. due to changes in the impurity composition, some are deemed constant enough to be described by a single global value or functional parametrization. These are, the coefficients $\alpha$ and $\kappa$ describing the wavelength dependence of scattering and absorption, $A_{IR}$ and $\lambda_0$ in the infrared absorption term as well as the parametrization of the scattering function, achieved through a mixture of the Henyey-Greenstein (Henyey and Greenstein, 1941) and Simplified-Liu (Liu, 1994) approximations of Mie scattering, and its asymmetry parameters $g$. This leaves 6 global parameters (three for the wavelength dependencies ($\lambda_0$, $\kappa$, $\alpha$), one for the ice intrinsic absorption ($A_{IR}$), two for the scattering function ($g$ and the mixing ratio)) and about 100 layers within the instrumented volume, with two parameters ($b_e$ and $a$) each, to be determined.

## 3 IceCube as a laboratory for glaciology

### 3.1 LED calibration data

As will be described in section 3.4 the absorption and effective scattering lengths encountered at IceCube depths range up to 400 meters and 100 meters respectively. The limited volume of the ice cores does thus not allow for a direct measurement of optical properties, even though they are able to inform on the impurity constituents and their size distributions. To enable in-situ calibration of the ice optical properties, each of the 5160 DOMs deployed in ice is equipped with 12 light emitting diodes (LEDs) that are positioned on a "flasher" board and can emit light one at a time or in simultaneous combinations. The LEDs are placed in pairs at $60°$ increments in azimuth, with one LED at a $48°$ elevation angle and the other pointing horizontally into the ice. Most of the LEDs emit light centered at $405\,\mathrm{nm}$ wavelength in a cone of about $9.7°$ width (RMS). The duration and intensity of the light flashes can be configured and range between $6\,\mathrm{ns}$ and $70\,\mathrm{ns}$ (FWHM) and up to $1.2 \cdot 10^{10}$ photons per flash.

For this study data with all available LEDs flashing individually and at the highest possible intensity have been used. Upon an LED flash the arrival times of photons received in all other DOMs are recorded. An example view of the overall light deposition in the detector is shown in Figure 3. An example light curve, histograming the measured arrival times, for one emitter-receiver pair is shown in Figure 4.

### 3.2 Photon propagation simulation

From the recorded LED data, ice properties are inferred by comparing the data to an expectation given different ice realizations. For a point-like emitter in the far field ($d \gg \lambda_{eff}$) and given a weak absorption coefficient compared to the scattering





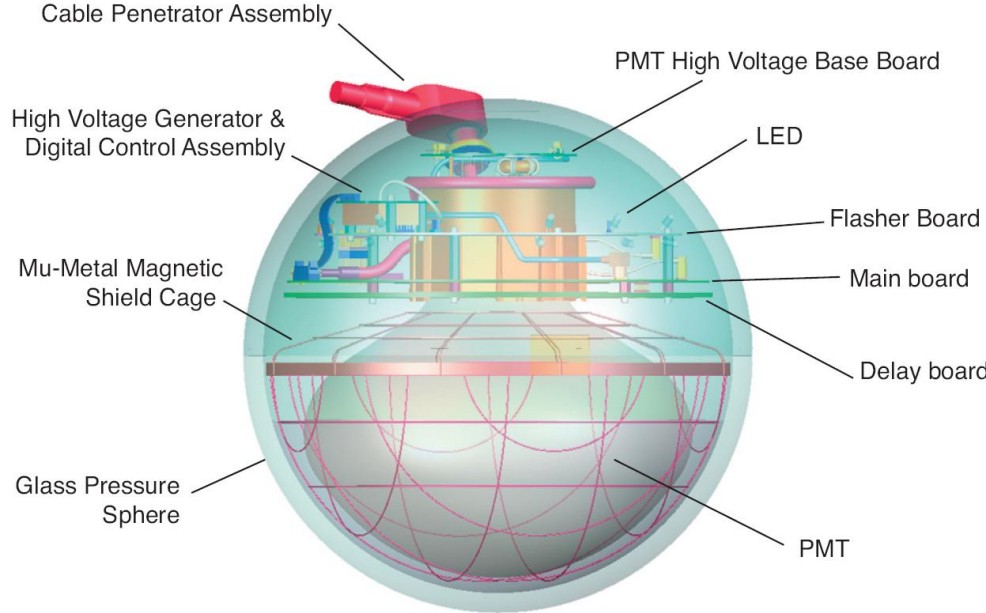

**Figure 2.** Sketch of a Digital Optical Module (DOM), the fundamental building block of IceCube, showing all major sub-assemblies. Figure Credit: IceCube

coefficient, the arrival time distribution $u(t)$, that is the density function belonging to the light curves, at a distance $d$ from an isotropic source is described by a Green's function (Ackermann et al., 2006) as

$$u(d,t) = \frac{1}{(4\pi \cdot Dt)^{3/2}} \cdot \exp\left(-\frac{d^2}{4Dt}\right) \cdot \exp\left(-\frac{tc_{ice}}{\lambda_a}\right), \tag{5}$$

where $D = c_{ice}\lambda_{eff}/3$ is the diffusion constant. As evident from this equation, the time of the rising edge is generally sensitive to the scattering coefficient, while the slope of the tail is determined by the absorption coefficient. While this behaviour is

generally also observed outside the far field, the Green's function is inaccurate in the semi-diffuse regime given by the clean, layered ice and at the sensor spacings used in IceCube. Thus, the photon propagation needs to be fully modeled in simulation. This is achieved through the use of photon propagation software, namely the "photon propagation code" (PPC) (Chirkin, 2013a).

PPC aims to be a full first-principle simulation, tracking each photon individually and as accurately as possible. For every

created photon the total lifetime, or absorption weight in multiples of absorption lengths, is sampled from an exponential distribution with unity scale. Next the distance to the next scattering process is determined in the same fashion and the photon is moved through a depth layered ice model along its current propagation direction towards the next scattering center. For each layer traversed, the length multiplied with the local absorption/scattering coefficient is subtracted from the current absorption / scattering weight. When the scattering weight reaches zero, the scattering site has been reached and the photon is deflected



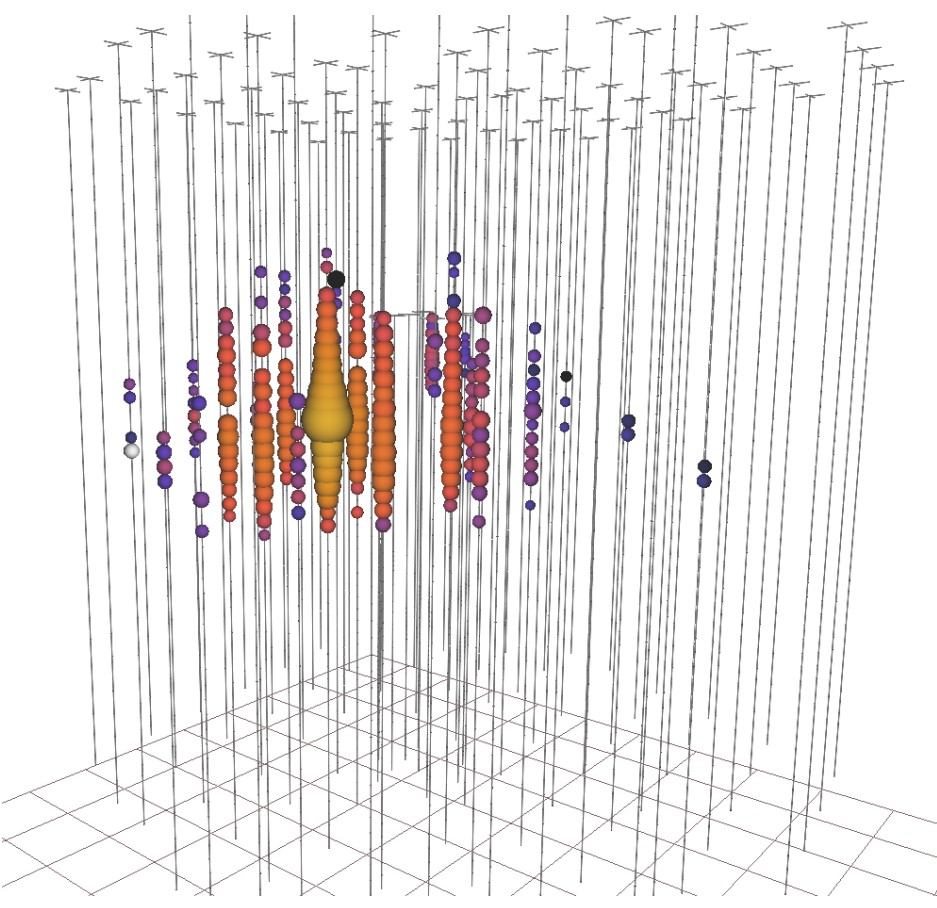

**Figure 3.** Example LED flasher event. DOMs which registered associated light are color coded according to the relative arrival time (yellow being early, purple late). The size denotes the recorded intensity in logarithmic proportions.

according to the modeled scattering function. The scattering transport process is repeated until the photon is either absorbed, as the absorption weight reaches an epsilon cut-off value, or the photon is incident on a DOM and stored for later processing.

PPC has been in active development and use since 2009. As photons propagate independently of each other, their simulation is an ideal use-case for parallelization using Graphics Processing Units (GPUs). Using a single GPU, the full paths of $\sim 10^8$ photons can be simulated per second, corresponding to simulating one full LED flash in 100 seconds. Computational resources

are still the limiting factor in these studies, in particular when it comes to evaluating systematic uncertainties through repeated analysis under slightly perturbed assumptions. For this study simulations amounting to roughly 400'000 GPU hours have been performed on the IceCube computing cluster.



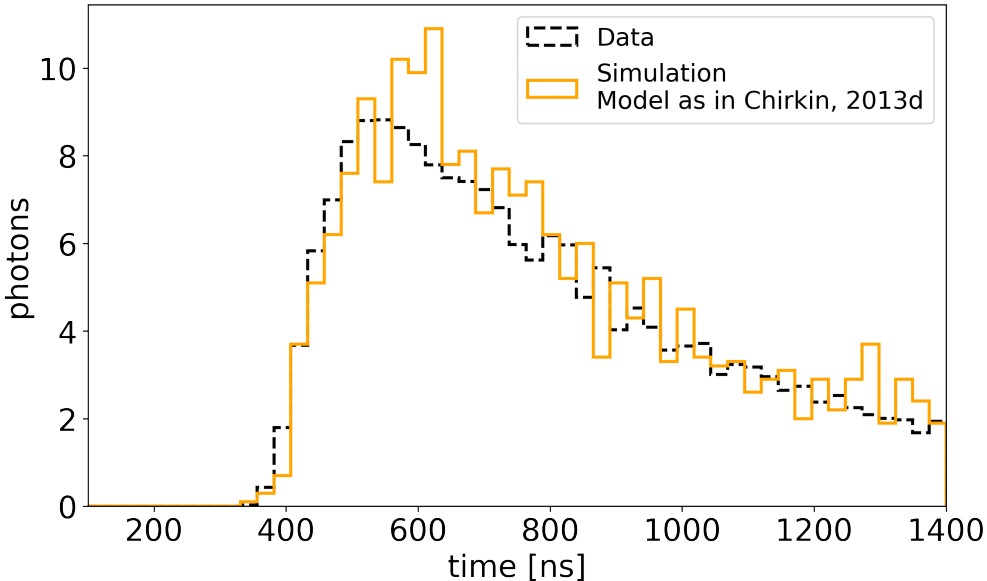

**Figure 4.** Example flasher light curve in 25 ns binning. DOM 50 on string 1 emits light which is detected by DOM 55 on string 8 about 150 m away. The data is averaged over 240 repetitions. The simulation is averaged over 10 repetitions.

### 3.3 Likelihood analysis

The photon propagation described in the previous section enables reproducing (LED) events in simulation, given a set of model parameters including a realization of the ice properties. Most ice calibration studies perform an optimization of the ice assumptions by minimizing the discrepancies between simulated and measured LED events. In practice, best estimators for the ice properties are obtained through a log-likelihood minimization, where a single likelihood value is computed for every pair of emitter and receiver DOMs. For this purpose, the experimental and the simulated events are averaged over the number of repetitions in this LED configuration (usually around 200 in data and 10 in simulation). The light curve of each receiving DOM is then binned in time using a Baysian Blocking (Scargle, 1998) algorithm, where each bin is multiples of 25 ns long and balances maximizing photon statistics per bin and accurately describing the rate of change of photon counts at the rising and trailing edges.

The per-event average expectation in each bin is a function of the sampled ice properties and nuisance parameters, such as a per-LED light yield, a timing offset of the light emission with regard to the LED trigger and the absolute LED orientations. The likelihood function used for comparing this expectation to data is given by

$$-\ln\mathcal{L} = \sum_i \left[ s_i \ln \frac{s_i/n_s}{\mu_s^i} + d_i \ln \frac{d_i/n_d}{\mu_d^i} + \frac{1}{2\sigma^2} \left( \ln \frac{\mu_d^i}{\mu_s^i} \right)^2 \right] \tag{6}$$





where $i$ denotes a receiver DOM and time-bin of its light curve, $s_i$ and $d_i$ the photon count in simulation and data for this bin, respectively, $n_s$ and $n_d$ the simulation repetitions and number of data events, $\sigma$ the model error and $\mu_s$ and $\mu_d$ the simulation and data expectation values. $-\ln\mathcal{L}$ is abbreviated as LLH in the following.

The model error takes into account potential discrepancies in reproducing data with a simulation that may be incomplete or may use non-ideal parametrizations. Using the model error, it is assumed that a difference between the expectation values of simulation and data can exist even at the best fit point, $\mu_s \neq \mu_d \neq (s_i + d_i)/(n_s + n_d)$. This is modeled through the penalty term in the likelihood  (Chirkin, 2013b). This extension also requires an optimization of the now in-principle independent expectation values within the likelihood calculation and is performed as described by Chirkin (2013b).

This likelihood (Chirkin, 2013b) improves on a common Poisson likelihood by taking into account the uncertainty of the expectation caused by the small statistics of the simulated data compared to the experimental data. Therefore, the expectation is optimized including the knowledge of the limited statistics of both the simulated and experimental data. In the limit of infinite statistics of simulated data this likelihood converges to a saturated Poisson likelihood.

DOM noise rates are added to the likelihood as a sum with $s_i$, where a constant noise rate of $500\,\mathrm{Hz}$ is assumed. DOMs with
a total per-event photon count exceeding 500 are excluded from the likelihood calculation to avoid biases from saturated photomultipliers. In addition, only bins within $-500\,\mathrm{ns}$ to $1000\,\mathrm{ns}$ around the peak of each light curve are included in the likelihood calculation, thus minimizing the effect of noise or Cherenkov photons from particle interactions accidentally overlapping with the LED events. This also minimizes the impact of afterpulses, a photomultiplier effect which leads to artificial pulses some time after a bright illumination and which is not modeled as part of the LED simulation.

The parameters of the ice model are generally obtained through likelihood scans, where each scan point is one realization of the ice model parameters tested against flasher data. The timing offset and LED intensity nuisance parameters are optimized for each realization analytically and through a number of low statistics iterations.

The likelihood method described above does not, in general, fulfill Wilks' Theorem (Wilks, 1938), which would, under certain conditions, allow one to approximate the distribution of the likelihood ratio between the best-fit and null hypotheses with
a chi-squared distribution. As such, the log-likelihood contour of a one dimensional likelihood scan enclosing the minimum by a $\Delta LLH$ of 1 does not represent a $1\sigma$ statistical uncertainty. Instead, the spread in LLH values equivalent to the $1\sigma$ uncertainty is obtained by re-simulating a realization close to the optimum a number of times and computing the standard-deviation of the resulting $LLH$ values.

Fitting the flasher data, the statistical errors on the ice properties, in particular the layered absorption and scattering co-
efficients, are entirely due to the limited simulation statistics, but generally remain below 1%. Thus the statistical error is subdominant compared to systematic biases introduced through incomplete modeling. This bias is hard to quantify, in particular due to the enormous computational cost. Taking into account the limited knowledge of the relative detection efficiencies of the DOMs, the discrepancy between fitted values using only horizontal or only tilted LEDs and different realizations of the modeled scattering function, the systematic uncertainty on the scale of absorption and scattering coefficients is estimated to be
around 5%.





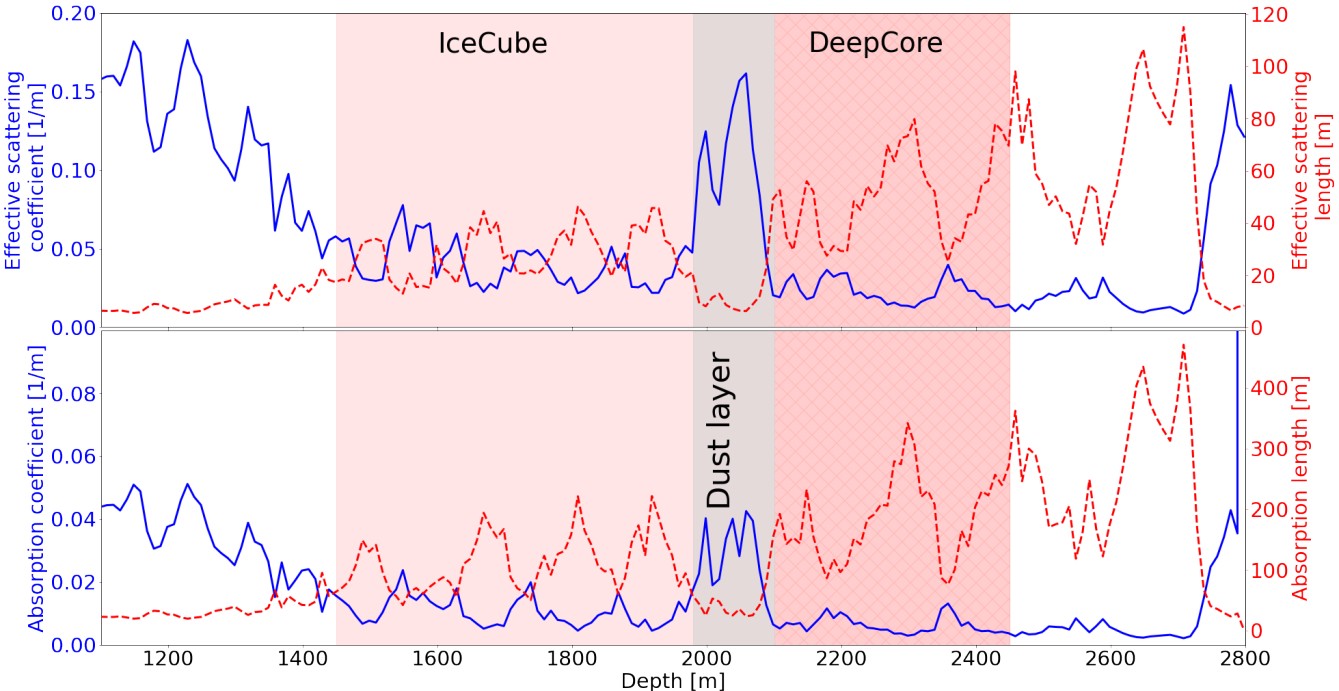

**Figure 5.** Stratigraphy of fitted absorption and scattering strength. Properties above the detector (<1450 m] are taken from AMANDA measurements (Ackermann et al., 2006), or are extrapolated from dust logger data. Properties below (>2450 m] are extrapolated using the stratigraphy as obtained from the EDML ice core (Bay et al., 2010) and ice age vs. depth curve from Price et al. (2000).

### 3.4 The South Pole Ice Model (SPICE)

Employing the experimental and analysis methods described above, absolute absorption and scattering coefficients and their wavelength scaling have been measured for all instrumented depths as described in detail by Ackermann et al. (2006) and Aartsen et al. (2013d). The resulting model, called the "South Pole Ice Model" (SPICE), continues to be updated and refined

as new aspects of the instrumentation such as the properties of the refrozen drill columns (Chirkin et al., 2021) as well as previously unconsidered features in the ice begin to be modeled. The stratigraphy used as the starting point for this study is shown in Figure 5. At the instrumented depths, absorption lengths mostly exceed 100 m, with the most significant exception being a region at around 2000 m, in IceCube commonly referred to as "the dust layer". This has been associated with a period of continuously elevated dust concentrations during stadial around 65,000 years ago (Ackermann et al., 2006).

While primarily developed and employed for the simulation of particle interactions, the deduced model parameters are also informative of ice properties in general. Most prominently the lowest measured absorption coefficients now serve as reference for a upper limit on ice intrinsic absorption as compiled by Warren and Brandt (2008). The technique of time-resolved photon counting has recently also been adopted by Allgaier et al. (2022) to deduce impurity concentrations in firn.




### 3.4.1 Layer undulation

One relevant complication is the layer undulation over the footprint of the array. As established from ground penetrating radar sounding (Fujita et al., 1999) ice isochrons can be traced over thousands of kilometers. While the ice surface is generally flat, deeper layers tend to gradually follow the topography of the underlying bedrock, with additional features such as upwarping and folds in basal ice (Cooper et al., 2019; Dow et al., 2018; MacGregor et al., 2015).

Available radar data generally does not have the spatial resolution required to map features within the footprint of IceCube. Instead the depth offset of characteristic features as observed in the dust logger data from seven different IceCube holes has been used to interpolate the depth-dependent layer undulations assuming laminar flow as described by Aartsen et al. (2013a). Layers with roughly constant scattering and absorption change in depth by as much as 60 m as one moves across the ~ 1 km detector. This gradient is mainly found along the SW direction, orthogonal to the flow direction. At the location of IceCube, the ice flows in the direction grid NW at a rate of about 10 m/year (Lilien et al., 2018), slowly draining into the Weddell sea after flowing through the Pensacola-Pole Basin (Paxman et al., 2019).

Within the context of the ice model, the depth offset at which a given ice layer is encountered relative to the stratigraphy as defined in the center of the detector is generally referred to as "tilt". Work is currently ongoing to further refine the modeling of layer undulation by incorporating data from ground penetrating radar data as well as fits to LED data.

## 4 The ice optical anisotropy

### 4.1 Experimental signature

Naively, the amount of light received from an isotropic source should not depend on the direction of the receiver with respect to the emitter. However, if we consider many DOMs, each with their 12 calibration LEDs and at a random azimuthal orientations in the refrozen drill holes, as isotropic emitters, and we average observations along different directions of emitter-receiver pairs of DOMs, we find a significant directional dependence. About twice as much light is observed along the direction of the ice flow compared to the orthogonal ice tilt direction when measured at distances of ~125 m, as seen in Figure 6. This *ice optical anistropy* was first seen in 2013 (Chirkin, 2013d), and is here called the *ice optical anisotropy*. The experimental arrival time distributions are nearly unchanged compared to a simulation expectation without anisotropy (as will be evident in Figure 8).

### 4.2 The anisotropy axis

A determination of the axis of the ice optical anisotropy can be achieved independent of any model assumption, by fitting the phase of the sinusoidal intensity modulation as shown in Figure 6. To obtain spatial resolution, the data is binned in emitting DOMs, either within a tilt-corrected depth range or by string number. Thus the data is dominated by propagation in a given depth range or in the vicinity of a given string. Figure 7 shows the resulting anisotropy axes.



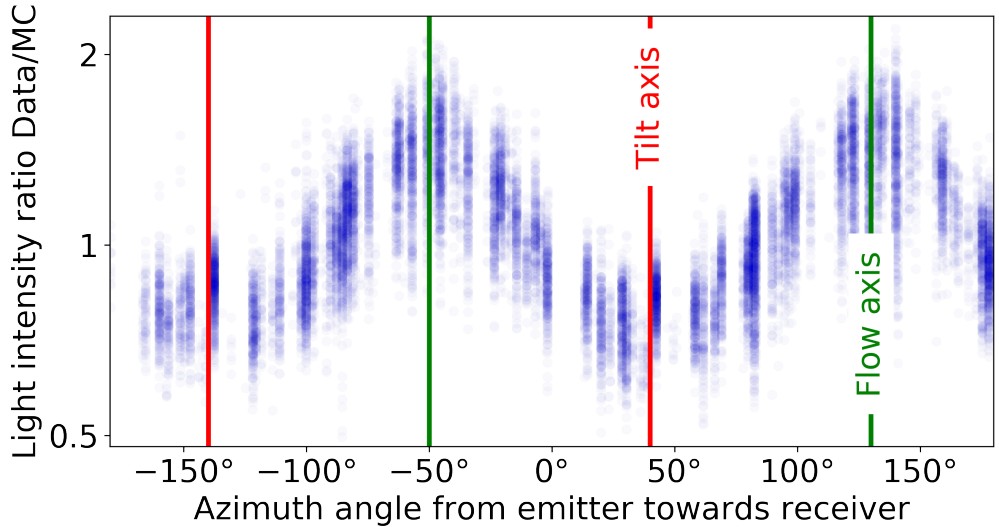

**Figure 6.** Ice optical anisotropy seen as azimuth dependent intensity excess in flasher data. Each dot is the observed intensity ratio for one pair of light emitting and light detecting DOMs comparing data to a simulation with no anisotropy modeling enabled. The tilt and flow directions are shown for reference.

The anisotropy axis is seen to have constant direction throughout the entire detector and is considered constant for all following investigations. The resolution is around $1°$ everywhere, except in the strongly scattering and absorbing dust layer. Edge strings are also disregarded as the lack of symmetric neighbors potentially leads to biased results.

The absolute direction is $130°$ in the IceCube coordinate system (azimuth of $0°$ is defined with respect to the positive x-axis in Figure 7 and runs counterclockwise), equivalent to the $40°$W meridian in the universal polar stereographic coordinate system, and is in excellent agreement with present day flow direction as measured using a GPS stake field by Lilien et al. (2018).

As a part of the models described in sections 4 and 5 a possible elevation angle to the anisotropy axis has been considered. In both cases a near constant elevation angle of on average $5°$ has been fitted. However, this fit is difficult to completely disentangle from effects that may arise as a result of mis-modeling of the layer undulations or the optical properties of the refrozen drill holes (Chirkin et al., 2021). As the resulting improvement in data-simulation agreement was seen to be small, this additional complication is not further considered here. As will be explained later, this elevation angle would directly relate to an elevation angle of the crystal orientation fabric. As this is a parameter which is hard to accurately obtain from ice core data, it may be further investigated in the future.

### 4.3 Early empirical modeling

Following the paradigm that ice optical properties are driven by Mie scattering on impurities, early attempts tried to model the anisotropy through directional modifications of absorption and scattering. In the original parameterization presented by




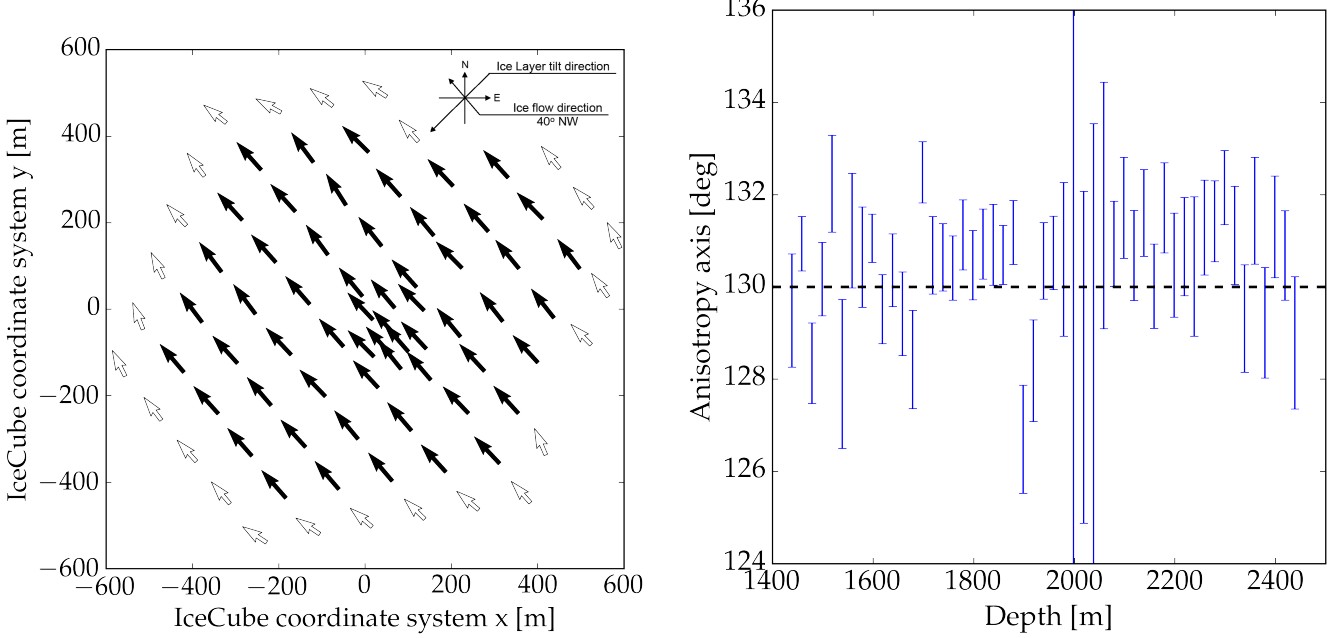

**Figure 7.** Measured anisotropy axes as a function of lateral position averaging over all depth (top) and as a function of depth averaging over all strings (bottom). Strings on the perimeter of the detector have been excluded, as the lack of symmetric neighbors leads to potentially biased results. The sensitivity is greatly reduced in the region of strongest scattering around 2000 m. The dashed line indicates the anisotropy angle averaged over all strings.

Chirkin (2013d), it was argued that due to time and space reversal symmetries the absorption length and geometric scattering length cannot be direction dependent. Therefore the anisotropy was implemented as a modification to the scattering function, the only remaining Mie scattering parameter.

Instead of evaluating the scattering function $f(\cos(\theta))$ with respect to the true orientation of the in-going and out-going photon directions $\boldsymbol{n}_{in}$ and $\boldsymbol{n}_{out}$, the scattering function is evaluated in terms of a stretched coordinate system:

$$f(\boldsymbol{n}_{in} \cdot \boldsymbol{n}_{out}) \rightarrow f(\boldsymbol{k}_{in} \cdot \boldsymbol{k}_{out}), \tag{7}$$

where the transformation is introduced through a matrix $A$:

$$\boldsymbol{k}_{in,out} = \frac{A \cdot \boldsymbol{n}_{in,out}}{|A \cdot \boldsymbol{n}_{in,out}|}. \tag{8}$$

When written in terms of a basis with the flow direction along the x-axis and the z-axis along the true zenith, the matrix is diagonal and can be expressed as:

$$A = \begin{pmatrix} \alpha & 0 & 0 \\ 0 & \beta & 0 \\ 0 & 0 & \gamma \end{pmatrix} = \exp \begin{pmatrix} \kappa_1 & 0 & 0 \\ 0 & \kappa_2 & 0 \\ 0 & 0 & \kappa_3 \end{pmatrix}. \tag{9}$$





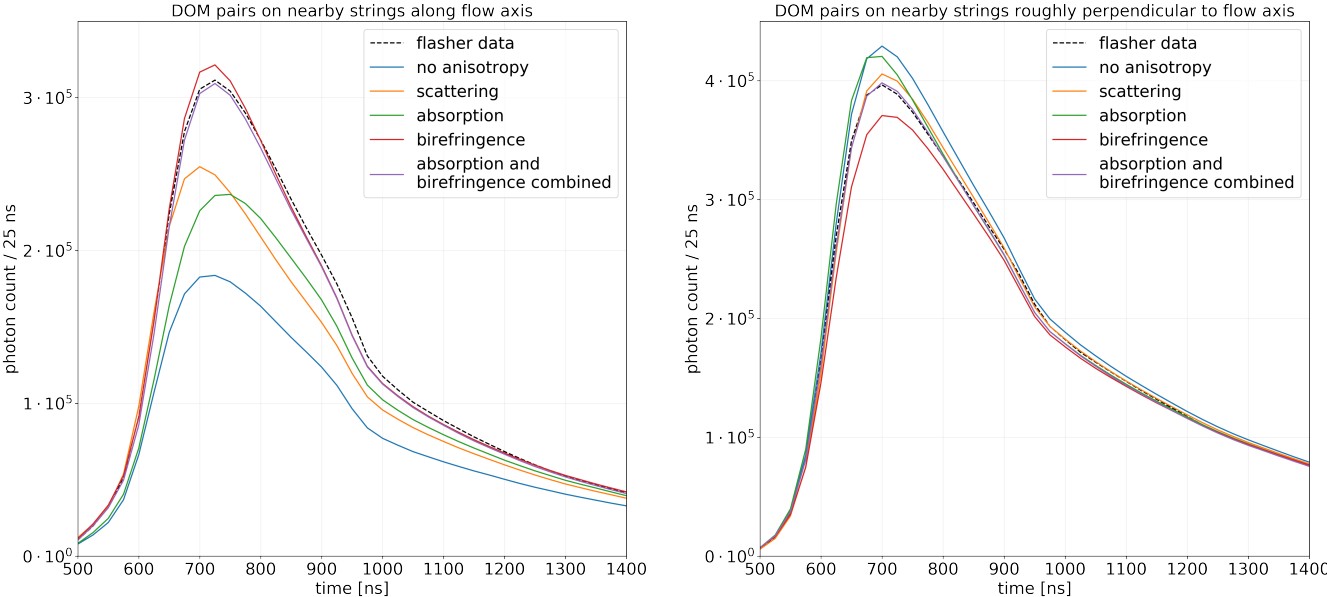

**Figure 8.** Comparison of fit quality achieved with different models for the ice optical anisotropy. Shown are photon arrival time distributions (summed counts in 25 ns time bins) for all nearest-pair emitters and receivers, roughly aligned along and perpendicular the ice flow. While the array geometry is well aligned with the flow axis, the nearest inter-module propagation direction perpendicular to the flow is roughly 30° off. The "absorption" and "scattering" models represent an ad-hoc, directional modifications to Mie scattering and absorption, but are unable to describe timing and intensity simultaneously. "Birefringence" refers to the microstructure based effect introduced in this paper. A combination of the "absorption" and "birefringence" model yields the closest match to data to date.

In order to conserve the direction-averaged effective scattering coefficients, which have been fitted to a high accuracy prior to the discovery of the anisotropy, it is further required that:

$$\alpha \cdot \beta \cdot \gamma = 1 \quad \text{or equally} \quad \kappa_1 + \kappa_2 + \kappa_3 = 0 \tag{10}$$

While not derived from first-principle Mie calculations, the parametrization was justified to be a plausible result of elongated impurities becoming preferentially aligned by the flow and thus introducing a direction dependence to the scattering function. While several glaciological studies (Potenza et al., 2016; Simonsen et al., 2018; Gebhart, 1991) explore the shapes of impurities, elongations for different impurities are not well established, nor is there to our knowledge any evidence for elongated impurities becoming oriented with the flow.

    In order to better understand the effect, it is helpful to derive a small-angle approximation of the modification to the average 335 scattering angle:

$$\frac{b_{eff,new}}{b_{eff}} = \frac{1 - \langle \cos\theta \rangle}{1 - g} = \frac{1}{2} \cdot (B_{in}B_{in} - n^i B_{in} n^j B_{jn}) \cdot |A\boldsymbol{n}|^2 \tag{11}$$





with $B = A^{-1}$ and $g$ being the direction averaged $\langle \cos\theta \rangle$. It is evident that the parametrization changes the effective scattering coefficient as a function of the propagation direction. Photons propagating along the flow axis experience less scattering than photons propagating along the tilt axis or inclined from the horizontal.

Using this parametrization, the anisotropy strength averaged over the entire detector was fitted to $\kappa_1 = -8\%$ and $\kappa_2 = \kappa_3 = 4\%$ (Chirkin, 2013d). An evaluation of the data-simulation agreement is shown in Figure 8. It shows summed photon arrival time distributions for all nearest emitter-receiver pairs, roughly aligned along and perpendicular to the ice flow for a variety of anisotropy models and the employed flasher data. This model results in more intensity being observed along the flow axis. However, there remains substantial disagreement between the model and the observed data. As scattering is reduced in the flow

direction light arrives earlier on average. The resulting change in the rising edge position is strongly penalized in the fit and limits the amount of intensity that can be recovered.

To reduce the shift of the rising edge, a directional modification to Mie absorption was considered as an alternative. This can be realized using an ellipsoidal scaling of the coordinate system similar to that described by Equation 9, but applied to the depth dependent Mie absorption coefficients at the time of propagation. This modulates the absorption coefficient by a factor

$\exp(\kappa_1 - \kappa_2)$ between the flow and tilt directions. A factor of nearly 11 was required to fit the data, which seems unphysical. As evident from Figure 8, this model results in a delayed rising edge for propagation along the flow direction as desired, and did result in an improved data description compared to the scattering based model described earlier, but is also unable to fully match the intensity difference to data.

To conclude, while resulting in partially successful effective descriptions, directional modifications to Mie scattering or

absorption cannot reproduce observations nor are such modifications well motivated on first principles.

## 5   Light diffusion in birefringent polycrystals

Departing from the paradigm that optical properties are purely driven by impurities, let us consider the impact of the microstructure of the ice itself on light propagation.

Light diffusion in birefringent, polycrystalline materials has been discussed as early as 1955 by Raman and Viswanathan

(1955). While the literature agrees that the combined effect of ray splitting on many crystal interfaces will lead to a continuous beam diffusion, the resulting diffusion patterns remained largely unexplored. Price and Bergström (1997) already considered this average overall diffusion in the context of Cherenkov Neutrino telescopes, but disregarded it as subdominant compared to scattering on impurities.

In a homogeneous, transparent and non-magnetic medium the relation between the electric field and the displacement field

as well as the magnetic fields is given as  (Landau and Lifshitz, 1960):

$$\boldsymbol{B} = \boldsymbol{H}, \quad \boldsymbol{D} = \epsilon \boldsymbol{E} \tag{12}$$

As the dielectric tensor $\epsilon$ is symmetric, one can always find a coordinate system where it is diagonal $\epsilon = diag(n_x^2, n_y^2, n_z^2)$, with $n_i$ being the refractive index along the given axis. Uniaxial crystals, such as ice in glacial environments, have two distinct





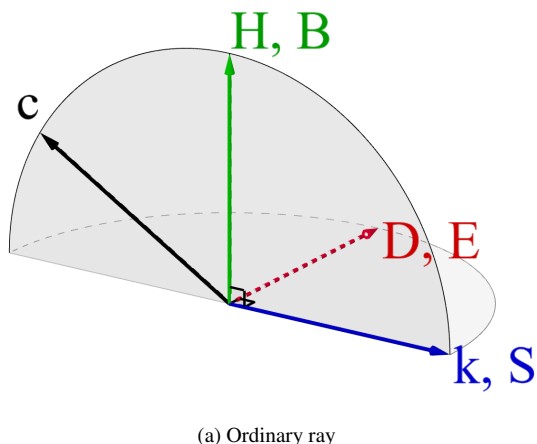
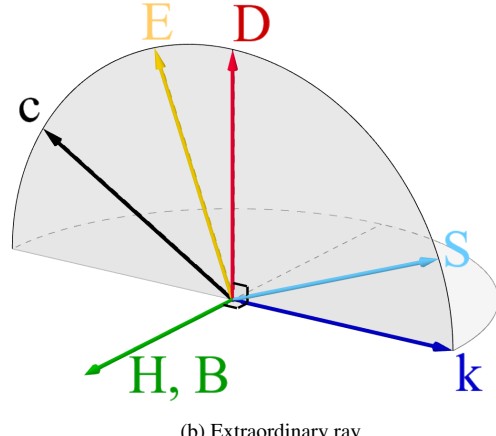

(a) Ordinary ray
(b) Extraordinary ray

**Figure 9.** Orientation of all electromagnetic vectors for the ordinary and extraordinary ray with respect to the crystal axis (c-axis). See text for detailed explanation of this figure.

refractive indices: $n_x = n_y \equiv n_o \neq n_z \equiv n_e$. The axis of the refractive index $n_e$ defines the optical axis and coincides with the

c-axis.

A light ray entering a uniaxial crystal is split into an ordinary wave and an extraordinary wave of orthogonal polarizations. Figure 9 visualizes the orientations of all electromagnetic vectors, the plane spanned by the optical axis $c$ and the wave vector $k$ is highlighted in grey. The electric field vector $E$ and the displacement vector $D$ for the ordinary wave are always co-linear to each other and perpendicular to both the optical axis of the crystal and the parallel propagation vectors $k$ and $S$. However,

the electric field $E$ for the extraordinary wave is not, in general, perpendicular to the propagation vector $k$. It lies in the plane formed by the propagation vector and the displacement vector. The electric field vectors of these waves are mutually orthogonal (Zhang and Caulfield, 1996). The energy flow is given by the Poynting vector $S = \frac{c}{4\pi} E \times H$. For the extraordinary wave, the Poynting vector $S$ is not parallel to $k$.

While the ordinary ray always propagates with the ordinary refractive index $n_o$, the refractive index of the extraordinary ray

depends on the opening angle $\theta$ between the optical axis and the wave vector $k$ (as described in a later section with Equation 15). The difference to $n_o$ is largest when the optical axis and the wave vector are perpendicular. In this case the extraordinary ray propagates with the refractive index $n_e$.

The birefringence strength can be expressed as:

$$\beta = \left( \frac{n_e}{n_o} \right)^2 - 1. \tag{13}$$

For ice $\beta$ is $\approx 2 \cdot 10^{-3}$ across the entire visible wavelength spectrum. Refractive indices at specific wavelengths can be found in Table 1.



| wavelength $\lambda$ (nm) | $n_o$ | $n_e$ | $\beta$ |
|---|---|---|---|
| 405 | 1.3185 | 1.3200 | $2.3 \cdot 10^{-3}$ |
| 436 | 1.3161 | 1.3176 | $2.3 \cdot 10^{-3}$ |
| 492 | 1.3128 | 1.3143 | $2.3 \cdot 10^{-3}$ |
| 546 | 1.3105 | 1.3119 | $2.1 \cdot 10^{-3}$ |
| 624 | 1.3091 | 1.3105 | $2.1 \cdot 10^{-3}$ |
| 691 | 1.3067 | 1.3081 | $2.1 \cdot 10^{-3}$ |

**Table 1.** Refractive indices of ice taken from Petrenko and Whitworth (2002)

## 5.1 Analytic calculation of a single grain boundary transition

Assuming an arbitrary ray incident on a plane interface, we first calculate the four possible wave vectors, the ordinary and extraordinary refracted rays and the ordinary and extraordinary reflected rays. Given the wave vectors, the four associated
Poynting vectors are calculated from the boundary conditions, yielding the energy flow and as such probable photon directions.

### 5.1.1 Wave vectors

Figure 10 shows the situation at hand: An incoming wave vector $\boldsymbol{k}$ intersects the interface and is split into four outgoing wave vectors $\boldsymbol{r}$. The coordinate system can always be chosen such that the surface normal $\boldsymbol{n}$ is along the y-axis and that the surface components of $\boldsymbol{k}$, and as such $\boldsymbol{r}$, are along the x-axis. Here we implicitly assume, as an approximation, that the boundary
surface is a perfect plane infinite in its extension, and, without a loss of generality, that the incoming and outgoing waves are all plane waves.

Because of translational symmetry of the interface surface, the surface components of all wave vectors are identical (Landau and Lifshitz, 1960): $k_x = r_x$. As the wave number is given by $k = \frac{2\pi}{\lambda}$, we can define a vector $\boldsymbol{n}$ such that $\boldsymbol{k} = \omega \boldsymbol{n}/c$, whose magnitude $n$ is the direction-dependent refractive index $n = \sqrt{\epsilon(\theta)}$. As such the magnitude of the wave vector is proportional
to the refractive index and we shall simplify $|\boldsymbol{k}| = n$ in the following.

**Outgoing ordinary rays**

Given the magnitude $n_o$ and surface component $k_x$ of the wave vector the y-component is:

$$r_y = \pm\sqrt{n_o^2 - k_x^2} \tag{14}$$

The outgoing ordinary ray of an inbound ordinary ray is not deflected, as it does not see a change in refractive index. In the
case of no birefringence, one obtains Snell's law for refraction and the usual law for reflection ($r_y = -k_y$).





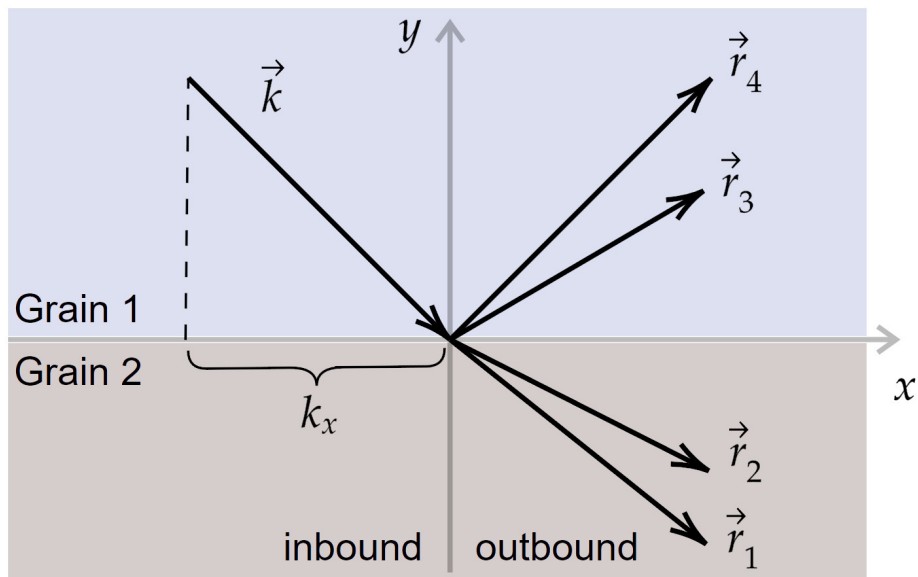

**Figure 10.** Sketch of wave vectors for the incident, reflected and refracted rays. The surface component is identical for all rays.

**Outgoing extraordinary ray**

Determining $r_y$ for the extraordinary rays follows the same logic, only with a refractive index which depends on the opening angle $\theta$ between the outgoing wave vector $\boldsymbol{r} = (r_x, r_y)$ and the optical axis $\boldsymbol{a} = (a_x, a_y, a_z)$:

$$\frac{1}{n^2} = \frac{1}{n_e^2} + \left(\frac{1}{n_0^2} - \frac{1}{n_e^2}\right) \cdot \cos^2\theta \tag{15}$$

The optical axis is given by the optical axis of medium 1 for the reflected and of medium 2 for the refracted ray. Rewriting $\cos(\theta)$ as scalar product between the wave vector and the optical axis gives:

$$\frac{1}{n_e^2} + \left(\frac{1}{n_o^2} - \frac{1}{n_e^2}\right) \cdot \frac{(a_x r_x + a_y r_y)^2}{n^2} - \frac{1}{n^2} = 0. \tag{16}$$

Here $n^2 = \boldsymbol{r}^2 = r_x^2 + r_y^2$. The solution is:

$$r_y = \frac{-\beta a_x a_y r_x \pm \sqrt{D}}{1 + \beta a_y^2} \tag{17}$$

with:

$$D = (\beta a_x a_y r_x)^2 - (1 + \beta a_y^2)(r_x^2 \cdot (1 + \beta a_x^2) - n_e^2) \tag{18}$$

$$= n_e^2 \cdot (1 + \beta a_y^2) - r_x^2 \cdot (1 + \beta \cdot (a_x^2 + a_y^2)) \tag{19}$$

Of the two solutions the direction appropriate for the reflected/refracted ray is chosen and the other discarded. In the case of no birefringence ($\beta = 0$) we again obtain the solution for the ordinary ray.



### 5.1.2 Poynting vectors

Once the wave vector directions are determined, the boundary continuity conditions can be written for normal components of $D$ and $B$, and for tangential components of $E$ and $H$. If $n$ is a normal vector perpendicular to the interface surface, we have:

$$n \cdot D_1 = n \cdot D_2, \quad n \cdot B_1 = n \cdot B_2, \tag{20}$$

$$n \times E_1 = n \times E_2, \quad n \times H_1 = n \times H_2 \tag{21}$$

Here the subscript 1 indicates the total sum of fields for incident and reflected waves, and the subscript 2 indicates the fields of the refracted waves propagating away from the boundary surface in the second medium. Since $B = H$ two of the equations above simply imply that $B_1 = B_2$ and $H_1 = H_2$. Together with the boundary conditions for $D$ and $E$, this is a system of 6 linear equations. These equations are sufficient to determine the amplitudes of 4 outgoing waves: two reflected (ordinary and extraordinary), and two refracted (also, ordinary and extraordinary). Since we only have 4 unknowns, 2 of these equations are necessarily co-linear to the rest, if the wave vectors were determined correctly.

From the solution to the linear equation system the Poynting vectors and as such the photon directions of the (up to) four outgoing rays are calculated. The relative intensity of these rays, as usually denoted in Fresnel coefficients, is derived from the Poynting theorem, which for our case (no moving charges, no temporal change in total energy) is given as

$$\oiint_{\partial V} S \cdot dA = 0 \tag{22}$$

where $\partial V$ is the boundary of a volume $V$ surrounding the interface. The choice of volume is arbitrary. A simple choice is a box around the interface. In the limit of an infinitely thin but wide box, it is evident that the sum of Poynting vector components normal to the interface plane is conserved.

Evanescent waves, i.e. waves with a complex wave vector, which decay away from the boundary surface and arise when the discriminant in the wave vector equation (Eq. 18) is negative, will necessarily yield vanishing contributions to such sum. As the photon interacts with a boundary there is a brief flow of energy along the surface boundary within evanescent solutions (if any), but no energy flows away from the boundary within such solutions. The evanescent waves need to be considered when solving the boundary conditions as given in Equation 21.

After deriving the solution presented here, we learned of the paper by Zhang and Caulfield (1996) and found that our approach is similar to the one they described.

### 5.2 Simulating diffusion patterns

Based on the calculations above, a photon propagation simulation for birefringent polycrystals was implemented in C++. At each grain transition, the outgoing photon is then chosen randomly, with probabilities proportional to the (up to) four normal components of the non-evanescent Poynting vectors, so to account for the relative intensities.

The resulting diffusion patterns, defined as the distribution of photon directions after crossing a given number of grains, depends on two factors related to the polycrystal configuration.





The assumed probability density distribution of c-axis orientations, that is the crystal orientation fabric, determines the refractive indices a photon will encounter. As measured c-axis distributions offer limited statistics and are restricted to the encountered fabric states, it is here necessary to statistically sample generic c-axis distributions. Appendix A briefly summarizes the different kinds of fabric and describes the approach developed to sample an arbitrary number of c-axes based on Woodcock

parameters, the usually published statistical moments associated to the fabric orientation tensor. Woodcock parameters for the ice at the South Pole are available from the South Pole Ice Core, SPC14 (Casey et al. (2014), drilled by the SPICEcore project in 2014–2016 at a location ∼1 km from the IceCube array using the Intermediate Depth Drill designed and deployed by the U.S. Ice Drilling Program (IDP) (Johnson et al., 2014)). It reached a final depth of 1751 m (Winski et al., 2019)), which corresponds to an ice layer depth of ∼1820 m in the IceCube coordinate system accounting for the layer undulation between

the two reference points. The c-axis distributions have been measured by Voigt (2017) at all depths and show an exceptionally clean girdle fabric at the overlapping depth.

As evident from Snell's law, in addition to the change in refractive index, the slope of the interface surface also dictates the refraction angle at a grain boundary transition. Thus the distribution of grain boundary plane orientations, resulting from a given grain shape, needs to be modelled in addition to the crystal orientation fabric. Appendix B shows that the surface

orientation density of an ensemble of ice crystals, simulated as a polyhedral tessellation of a volume, can be approximated using a tri-axial ellipsoid that represents the average shape. For a generalized ellipsoid the diffusion patterns are thus not only a function of the opening angle between the initial photon direction and the flow (as expected from the crystal orientation fabric), but depend on the absolute zenith and azimuth orientation of the propagation direction with respect to the flow. Employing an alternative parametrization, developed prior to the one introduced in section 6.1, it was determined early on that fully tri-axial

ellipsoids offer no advantage to describing the flasher data compared to prolate spheroids, where the major axis is aligned with the flow and the horizontal and vertical minor axes are identical. These spheroids, described by the size of the major axis and an elongation, are what we restrict ourselves to here. Grain size and shape distributions have not yet been fully published by the SPICEcore collaboration, but are expected from preliminary material shown at conferences (Alley et al., 2021) as well as other cores (Weikusat et al., 2017; Lipenkov et al., 1989; Stoll et al., 2021; Faria et al., 2014a) to be on the mm-scale with

elongations of at most a factor of two.

Simulated diffusion patterns after 1000 boundary crossings for four initial propagation directions relative to the flow axis and assuming on average spherical grains as well as a perfect girdle fabric are shown in Figure 11. The overall diffusion is largest when propagating along the flow direction and becomes continuously smaller towards the tilt direction. For intermediate angles the distribution is slightly asymmetric, resulting in a mean deflection towards the ice flow axis. The diffusion being

largest along the flow axis results in a reduction of intensity in this direction, which is contrary to observations. The deflection however slowly diverts intensity from the tilt direction and overpopulates the flow (see Figure 12) direction. Thus a good fit to the data should be obtainable by finding the right combination of crystal orientation fabric, shape and crystal size as it changes the number of crystals per distance (see section 6.1).

To validate our calculations and implementation, a polycrystal was realized in Zemax, a commercial optics simulation

program, using a polycrystal tesselation simulated using *Neper: Polycrystal Generation and Meshing* (Quey et al., 2011) and




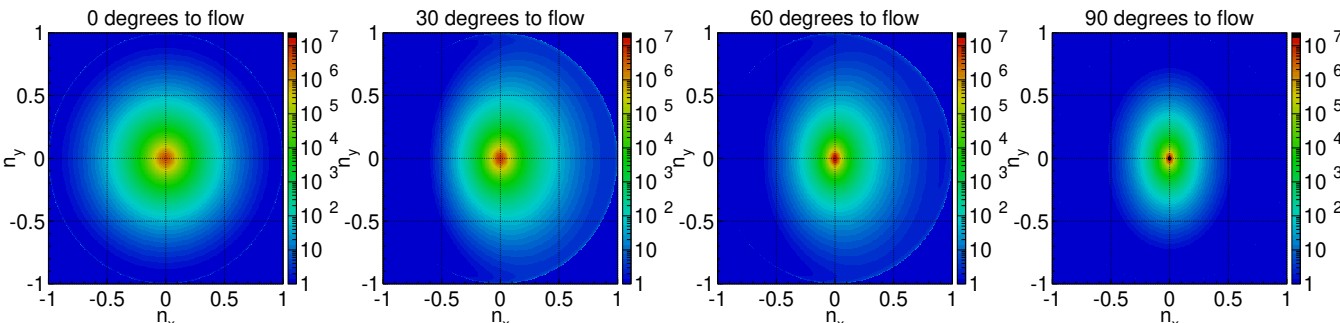

**Figure 11.** Example diffusion patterns after photon propagation through 1000 crystals (roughly equivalent to 1 m) with a perfect girdle distribution of c-axis orientations. The initial direction is perpendicular to the picture. The figures show histograms of the x and y components of the final propagation directions ($n_x$ and $n_y$). The subtle effect of photon scattering towards the ice flow (with components $(\sin(\alpha), 0, \cos(\alpha))$ in this example, $\alpha =$ angle between initial photon direction and ice flow) can be seen. Left: Propagating along the flow. Right: Propagating along the tilt direction. The change in diffusion as well as the slight asymmetry for intermediate angles are visible.

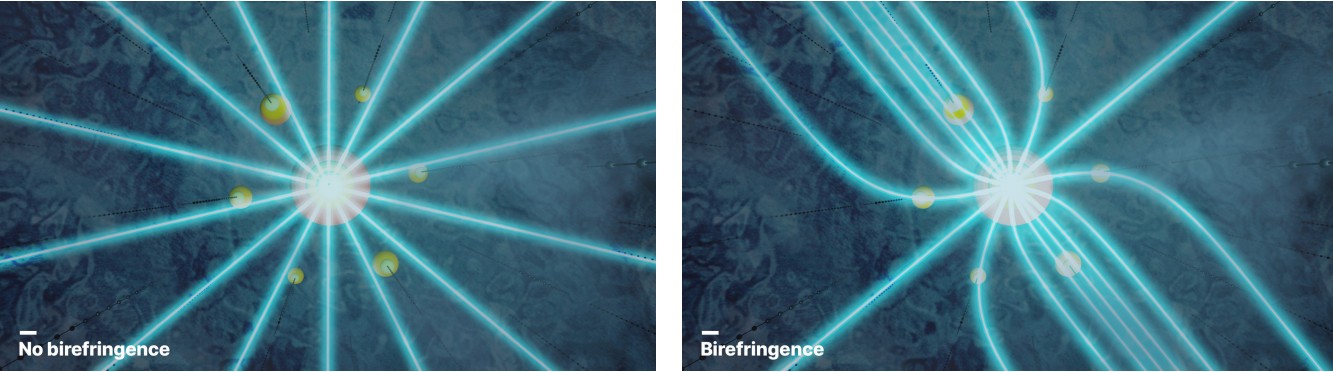

**Figure 12.** Artist illustration visualizing the deflection concept. Without birefringence light streams out radially from an isotropic light source. With birefringence rays get slowly deflected towards the flow axis. (The effects of scattering and diffusion are not shown. The hexagonal pattern of the IceCube array around the light source is shown; DOMs which detected photons are indicated similarly as in Fig. 3.)





exporting each interlocking monocrystal as a CAD object. The same quantitative behavior as described above is reproduced, however this approach does not allow for a flexible configuration and is slow to simulate a reasonable photon statistics

## 5.3 Comparison to fabric-induced anisotropies in radar measurements

Before incorporating the diffusion patterns into the overall IceCube simulation and fitting new ice parameters, we will discuss
some conceptual differences of the birefringence-induced optical anisotropy in comparison to birefringence effects in radar measurements, which many readers may be more familiar with.

When probing the ice with radio waves the employed wavelength is orders of magnitude larger then the crystal size. Thus the waves do not interact with individual grains, and propagation is only influenced by the bulk dielectric tensor, weighting the per-crystal dielectric tensor with their relative occurrence and gradually changing as function of location. Since the birefringence
strength $\beta = (n_e/n_o)^2 - 1$ is an order of magnitude larger in radio ($\beta \sim 1\%$) compared to optical ($\beta \sim 0.2\%$), the available observables are primarily direction-dependent timing delays (either of the entire pulse, or measured as a phase difference) and–for polarimetric systems–changes in the received polarization with respect to the emitted polarization.

Given the timing precision of IceCube, and given the low birefringence strength in the optical regime, the effect of birefringence on timing will not be relevant here. Even assuming the unrealistic case where one ray propagates purely with the
ordinary and another with the extraordinary refractive index, the propagation delay over 250 m would only amount to $\sim 1$ ns, which is undetectable with IceCube. Polarization is also not an available observable using IceCube data. Since each crystal effectively acts as a polarization analyzer and a large number of these is randomly sequenced, the diffusion patterns also do not depend on the initial polarization.

Instead, since the wavelength is small compared to the crystal size, light rays experience the individual grains as distinct
objects and slowly diffuse through the continued refractions and reflections at the grain boundaries. Given a mean elongation or equivalently a preferential c-axis distribution in addition to the diffusion, rays get on average slowly deflected towards the elongation axis. To our knowledge this is a newly discovered optical effect, not described in the literature before.

## 6 The birefringence ice model

### 6.1 Parametrizing diffusion patterns

Diffusion patterns have been simulated for a wide range of spheroid elongations (1-3) and fabric parameters (spanning the plane of Woodcock parameters between 0.1 and 4 in both dimensions). As evident from the example in Figure 11, these diffusion patterns have a strong central core with a broad large-angle tail. The tail is dominated by single large angle reflections and as such scales linearly with the number of crystal crossings. We found that the precise simulation of the tail is unimportant, in particular as shape uncertainties of the Mie scattering function far outweigh the errors introduced by a simple parametrization.
Therefore the distribution is modeled as a 2d-Gaussian on a sphere, lending itself to usual scaling (with distance) relationships for mean displacement and width. The distributions are very slightly skewed towards the flow axis, and are slightly better de-





scribed by a skewed Gaussian. A number of more complicated functions were also fit with good success in precisely describing the underlying distribution. Figure 11 in fact uses a function with ten parameters to illustrate all features of the distribution without statistical fluctuations. These were however abandoned, as no simple distance scaling could be established.

The three parameters of the diffusion pattern modeled with the 2d-Gaussian on a sphere are the two widths (in the directions towards the flow, $\sigma_x$, and perpendicular to it, $\sigma_y$), and a single mean deflection towards the flow, $m_x$. The mean deflection in the perpendicular direction was zero for all cases that we chose to include into the final model (i.e., single-axis ellipsoids for particle shape and selected crystal fabric configurations). Because we mainly simulate small deflections (ignoring the long tails), we simulated the 2d Gaussian in Cartesian coordinates, and then projected that to the sphere with an inverse stereographic

projection. The three quantities were fitted to the following functions of angle $\eta$ of the initial photon direction with respect to the ice flow, for simulations with a fixed number of 1000 crystal crossings:

$$m_x = \alpha \cdot \arctan(\delta \cdot \sin\eta\cos\eta) \cdot \exp(-\beta\sin\eta + \gamma\cos\eta) \tag{23}$$

$$\sigma_{x,y} = A_{x,y} \cdot \exp(-B_{x,y} \cdot (\arctan(D_{x,y}\sin\eta))^{C_{x,y}}). \tag{24}$$

These functions were found to describe all considered crystal realizations with only 12 free parameters ($A_{x,y}..D_{x,y}$ and $\alpha..\delta$).

Figure 13 shows the mean deflection for nine crystal configurations. Note that increasing elongation has a stronger effect compared to a strengthening fabric, i.e. increasing the value of the Woodcock parameter $\ln(S2/S3)$.

## 6.2   Applying diffusion patterns in photon propagation

During normal photon propagation directions are only updated upon scattering. To minimize the additional computational burden, the new birefringence anisotropy is discretized and also evaluated only at the scattering sites. This requires scaling the

diffusion, deflection and displacement derived from simulation through 1000 grains to the number of traversed grains between two scattering sites. This introduces a new model parameter, the average grain size, and also requires taking into account the different average crystal chord lengths as a function of propagation direction (as described in Rongen (2019)), further increasing the importance of elongation over fabric.

    The grain size distribution, that is the size distribution of ice mono-crystals, defines the distance between interface crossings.

As would be expected from a diffusion process, and was confirmed in simulation, the deflection scales linearly and the diffusion scales with the square root of the number of traversed grains $n$ ($\sigma_{x,y} \propto \sqrt{n}$ and $m_x \propto n$). The overall ice diffusion strength, including both Mie scattering as well as the birefringence-induced diffusion, has previously been measured to great accuracy. To decouple the fitting of anisotropy properties from this overall ice, the effective scattering Mie coefficient was reduced by the amount resulting from the birefringence-induced light diffusion assuming on average isotropic photon directions.

Updating not only a photon's direction with deflection due to birefringence, but also the photon coordinates (as it shifts transversely with respect to straight-path expectation) at the next Mie scattering site, improves the agreement with data in the final fit. Due to the simple physics of cumulative photon deflections, the effect can be simulated at a small additional computational cost and with no additional parameters. Assuming without loss of generality that all birefringence deflections happen at constant distance interval $\Delta l$ and that these can be sampled from the same distribution (which depends on the initial

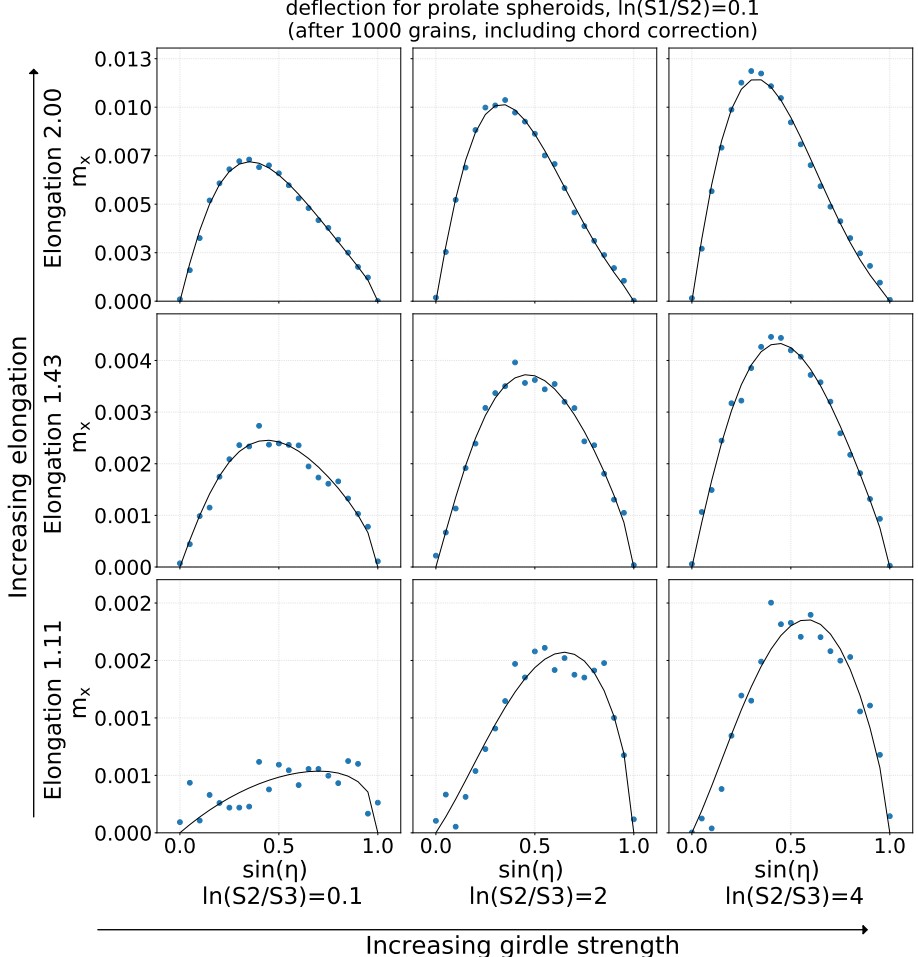

**Figure 13.** Deflection $m_x$ as described in the text as a function of opening angle to the flow for a number of crystal configurations. The black curves were fitted through the blue, simulated points using the functional form introduced in Equation 24.

photon direction), as the individual and even final calculated deflections are very small, we can express the new photon direction $\boldsymbol{n}$ and coordinates $\boldsymbol{r}$ after $N$ deflections as:

$$\boldsymbol{n} = \boldsymbol{n}_0 + \sum_{i=1}^{N} \Delta \boldsymbol{n}_i, \tag{25}$$

$$\boldsymbol{r} = \sum_{i=1}^{N} \boldsymbol{n}_i \cdot \Delta l = \Delta l \cdot N \cdot \boldsymbol{n_0} \cdot + \Delta l \cdot \sum_{i=1}^{N} \sum_{j=1}^{i} \Delta \boldsymbol{n}_j \tag{26}$$

     The second term in each of the two expressions above describes a cumulative direction change $\delta \boldsymbol{n}$ and relative coordinate

update $\delta \boldsymbol{r}$ respectively (we note that the total distance traveled is $L = \Delta l \cdot N$). We can now calculate, that in the limit of large





$N$ we get

$$\langle \delta \boldsymbol{r} \rangle = \langle \delta \boldsymbol{n} \rangle \frac{L}{2}, \tag{27}$$

$$\langle \Delta (\delta \boldsymbol{r} - \delta \boldsymbol{n} \frac{L}{2})^2 \rangle = \langle \Delta (\delta \boldsymbol{n})^2 \rangle \cdot \frac{L^2}{12}, \tag{28}$$

$$\langle \Delta (\delta \boldsymbol{r} - \delta \boldsymbol{n} \frac{L}{2}) \cdot \Delta (\delta \boldsymbol{n}) \rangle = 0. \tag{29}$$

$\Delta$ in the equations above is the variation (difference) from the mean of the quantity immediately following in brackets. These equations indicate that the coordinate update $\delta \boldsymbol{r}$ can be sampled from a distribution with a mean given by the first equation (which could be approximated by propagating the photon half the distance with initial direction vector and the other half with the final direction vector), and variance given by the second equation. Because there is no correlation between the residual in the variance and the deflection vector, as shown by the third equation, the variance can be sampled using the already tabulated

birefringence parameters independently from sampling the variance of the deflection vector.

## 6.3 Fitting to flasher data

Besides the anisotropy direction already discussed in section 4.2, the model described above requires four parameters to specify a birefringence anisotropy realization; crystal size and elongation and the two Woodcock parameters $\ln S1/S2$ and $\ln S2/S3$. Additionally allowing for a correction to the previously established total absorption and scattering coefficients adds two more

parameters. As minimizing all six parameters for all 100 depth layers in the ice model is not computationally feasible, we need to simplify the model by identifying some parameters which are either depth independent or have a small effect on the data-simulation agreement.

The required pre-fits, as well as the final depth evaluation, were performed following the method described in Aartsen et al. (2013d) by minimizing the summed LLH comparing the *single-LED* data set (where all 12 LEDs were flashed one at

a time on all in-ice DOMs ) with the full photon propagation simulation of these events taking into account precisely known DOM orientations as measured in Chirkin et al. (2021). Fits for individual layers were carried out by only including LEDs situated within the considered (tilt corrected) ice layer into the LLH summation. This method offers a reduced depth resolution compared to Aartsen et al. (2013d), but reduces computation time while making use of the full data. An example LLH space at a depth of ∼1500 m is shown in Figure 14. During the pre-fits the following behavior was noted: Given a girdle fabric

$(\ln(S_1/S_2) >> \ln(S_2/S_3))$, the actual fabric strength has a small effect and cannot be distinguished by the data. Accordingly the fabric has been fixed to values as measured in the deepest sections of the South Pole Ice Core, SPC14, Voigt (2017) $(\ln(S_1/S_2) = 0.1$ and $\ln(S_2/S_3) = 4)$. The fit is largely degenerate in crystal elongation and size, with small, near spherical crystals yielding similar results to larger, more elongated realizations. Thus, the elongation was fixed to 1.4, which is a good fit at all layers and similar to the values as measured in the deepest parts of SPC14 (Alley et al., 2021).

Fitting the remaining parameters (absorption and scattering corrections and crystal size) for all layers yields a significant improvement as seen for example in the average light curves in Figure 8 (birefringence-only line). The best-fit still features clearly visible discrepancies, such as an elevated intensity in the peak region in the case of propagation along the flow direction





**Figure 14.** LLH space for one ice layer and a subset of parameters. Each panel shows a marginalized 2D space, each point being a simulated ice realization, color coded by its LLH distance from the best fit. In this example, the absorption anisotropy is allowed to float (corresponding to the final model). This example is particularly detailed and was used to understand the behavior of the pre-fits. In particular note the strong degeneracy in crystal elongation and size (parameterized as the scale of the major axis). Near spherical crystals yield similar results to larger, more elongated realizations. The final fit for size, scattering and absorption correction as performed for all layers generally contains around 100 tested realizations per layer.





and too little intensity in the peak region in the case of propagation perpendicular to the flow direction. Problematically, the crystal sizes required to obtain this result are on the order of 0.1 mm and as such far smaller than expected from the overlapping SPC14 depths (Alley et al., 2021).

After thoroughly checking both the assumptions and implementation of the birefringence model, it was decided to reintroduce scattering as well as absorption anisotropy, both following the formalism as described in section 4.3, into the fit. As would be expected from the timing behavior, the fit does not make use of the scattering anisotropy, but surprisingly the absorption anistropy is mixed into the birefringence model with a significant non-zero contribution. The fitted strength of the absorption anisotropy is nearly depth independent with an average of $\kappa_1 = 0.6$ and $\kappa_2 = \kappa_3 = -0.3$, resulting in a directional modulation of the absorption coefficients by a factor 2.45. This means a departure from a first-principle model, but was adopted for its improvement in data-simulation agreement. After including the absorption anisotropy, absorption and scattering corrections and the crystal size were again fitted for all layers.

## 7    Resulting ice model

Figure 15 depicts the best-fit stratigraphy of grain sizes. The overall grain size of $\sim$1 mm as well as the increase in size where the ice is older and cleaner at larger depths are as generally expected and measured in glaciology (Laurent et al., 2004; Alley et al., 2021). As noted previously, the fit is largely degenerate in elongation and size. As a result the overall size scale is somewhat unconstrained. Repeating the fit under the assumption of an elongation of 1.7 instead of 1.4 for example results in on average 26% larger circle equivalent diameters.

As shown in Figure 8, the new model significantly improves in matching the flasher data light curves both in terms of timing and total intensity with regards to older models and overall achieves an excellent data-simulation agreement. While these light curves only represent part of the full data, the average relative deviation of each model from the data in the plots as shown is 8.5%, 3.3% and 2.4% for the scattering function based model, the birefringence-only model and after including the ad-hoc absorption anisotropy respectively.

Wide-spread application in physics analyses requires large-scale simulations and is still in preparation. Nevertheless, first tests employing the ice model in direct-fit reconstructions (Chirkin, 2013c) of high-energy events  (Aartsen et al., 2013c) indicate that the improved data-simulation agreement seen in flasher data also translates to more accurate descriptions of neutrino events.

## 8    Outlook

The model presented here is the first time that the ice microstructure is included in the modeling of ice optical properties at macroscopic scales. Due to the need to include absorption anisotropy, for which no first-principle explanation is known, there appear to remain additional physical effects not fully accounted for by the first-principles model. At this point it remains unclear whether the anisotropic Mie absorption is real or if it is an artifact from an incomplete modeling of birefringence effects.



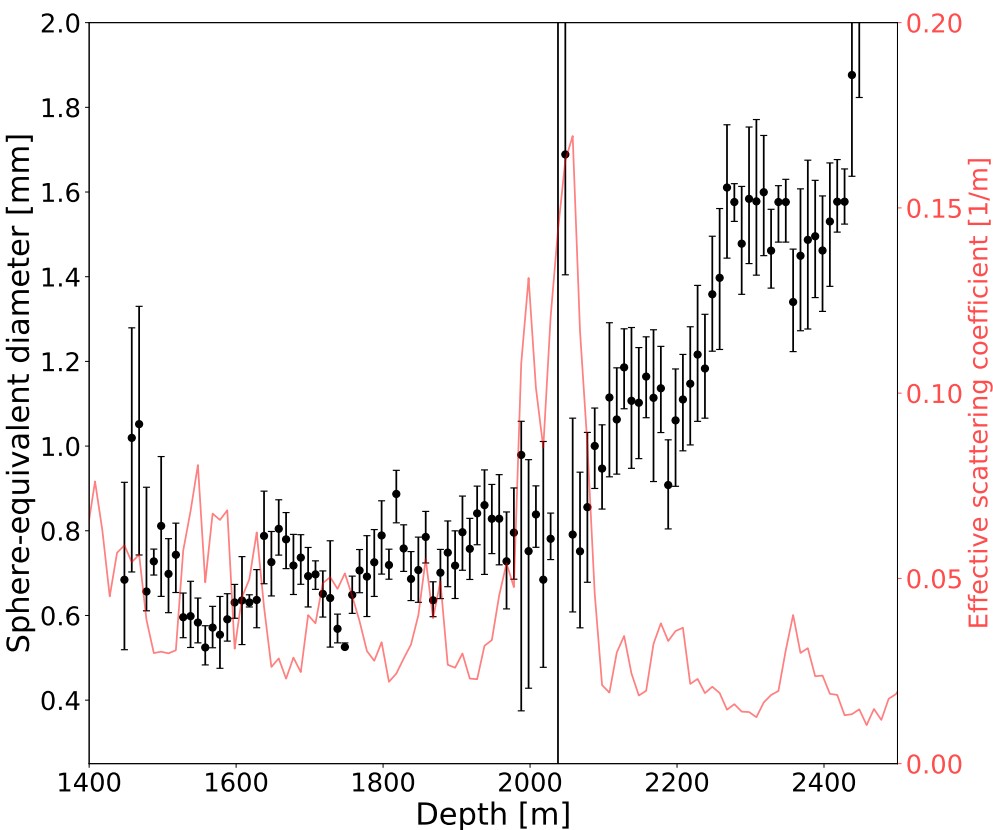

**Figure 15.** Best-fit crystal sizes as deduced in this analysis. The sphere-equivalent diameter denotes the diameter of a sphere with equivalent volume to the fitted spheroid describing the average crystal size and elongation at each depth. Error bars denote the statistical uncertainty only.

Inclusion of ice-intrinsic attenuation in the electromagnetic calculations in section 5 may already change the overall diffusion
patterns. In addition, birefringent materials also exhibit diattenuation where the imaginary index of refraction is polarization-
and direction-dependent (Grechushnikov and Konstantinova, 1988). The overall imaginary refractive index of ice is largely
unknown (with upper limits derived from IceCube/AMANDA measurements (Ackermann et al., 2006) as mentioned earlier)
and diattenuation of ice in the optical has, to our knowledge, not been studied at all. A first step in exploring these options
will be to include per crystal (di-)attenuation into the electromagnetic modeling, with the complex refractive indices as free
parameters and fitting required values given different assumptions on the crystal orientation fabric.

The only other known and currently neglected birefringence effect is photoelasticity. Photoelasticity describes the change in
refractive index due to applied stresses and is a property of all dielectric media, including ice. Ice is anecdotally known (Hobbs,
2010) to exhibit strong photoelasticity compared to its intrinsic birefringence strength. Yet, the stress optical parameters have
so far only been measured by Ravi-Chandar et al. (1994), for light of an unspecified wavelength, at an unspecified temperature

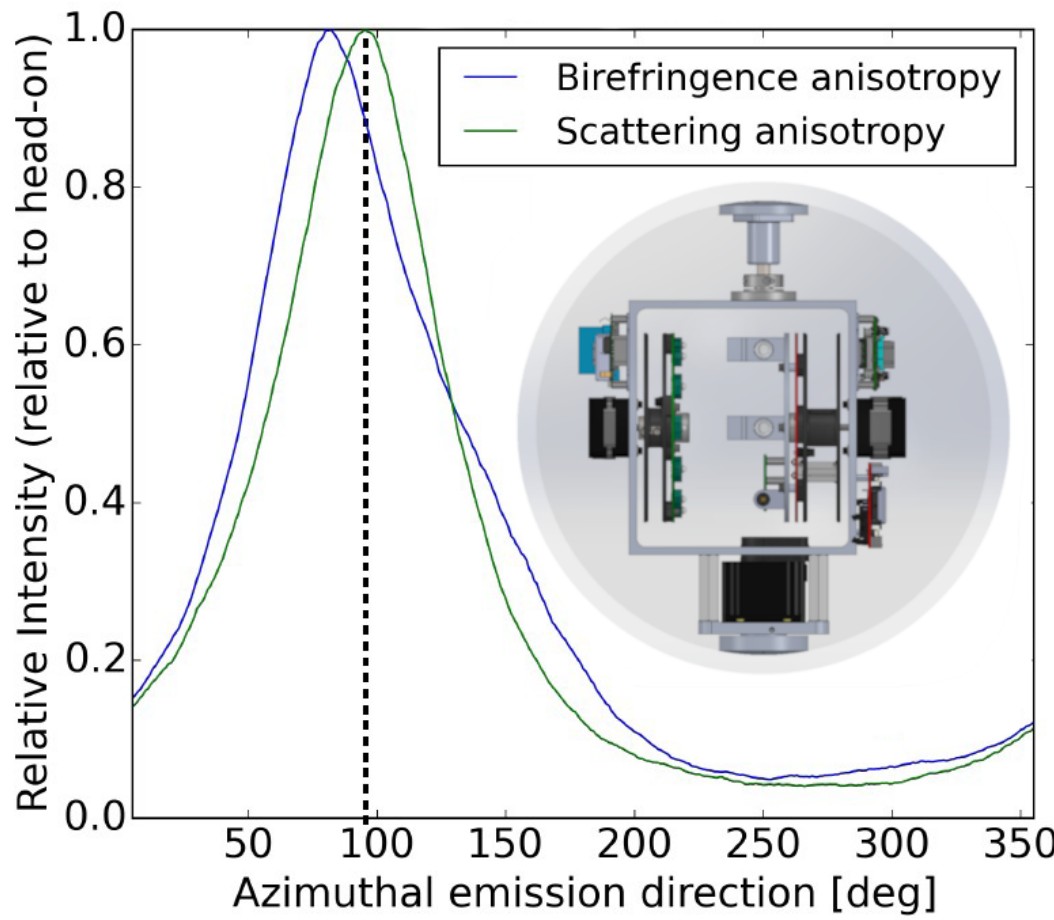

**Figure 16.** Simulated intensity profiles as the Pencil Beam sweeps over a receiver given two different anisotropy models. A design sketch of the Pencil Beam is shown superimposed.

and only for light propagating along the c-axis. Ravi-Chandar et al. (1994) arrived at a material fringe value of $\sim 67\,\mathrm{kN/m}$. Taking this measurement at face value, unrealistically large internal stress of roughly $200\,\mathrm{MPa}$ would be required to match the unstressed difference in refractive index.

To investigate the potential relevance of photoelasticity for light diffusion in deep glacial ice, the first step will be to repeat the Ravi-Chandar et al. (1994) measurement and extend it to light propagating orthogonal to the c-axis. If photoelasticity adds a 635 significant contribution, it would allow the presented measurement to also probe the stress state of the sampled ice, in addition to the already studied microstructure.





## 8.1 IceCube Upgrade

The IceCube Upgrade (Ishihara, 2021), planned to be deployed in 2025/26, marks the first extension of the IceCube Detector. Over 700 additional modules, including a number of stand-alone calibration devices (Henningsen et al., 2020; Rongen et al.,
2021b), will be deployed on seven additional strings. Of particular interest for the anisotropy are eleven so called Pencil Beam devices, as depicted in Figure 16. They allow for a laser-like beam to be directed in arbitrary directions, enabling in particular sweeps over receiver directions. The birefringence-induced deflection yields a unique signature, where the emission direction of maximum received intensity is offset from the geometric direction to the receiver. Measuring sweeping profiles for several emitter-receiver pairs at different orientations will allow to disentangle absorption and birefringence contributions
to the anisotropy with high precision.

## 8.2 Borehole logging

The described measurement is particularly tailored to the IceCube experiment. Nevertheless the optical anisotropy effect may still prove to be a useful tool for glaciology. As decribed by Rongen et al. (2020) a, most likely fabric induced, azimuthal anisotropy was also observed in the back-scattered intensity recorded by an optical dust logger deployed down the SPC14 drill
hole.

To date, the measurement has only been described qualitatively. An accurate simulation of the back-scattering scenario would need to include a good model for the large-angle tail of the Mie scattering function, which is currently poorly constrained from IceCube data. Given a better understanding of back-scattering processes, for example derived using the Pencil beam described above, optical logging of drill holes could become a complementary tool for fabric, crystal size and elongation studies and find
wider application in glaciology.

## 9   Conclusions

Measurements of ice optical properties in the context of the calibration of the IceCube Neutrino Observatory and its predecessor AMANDA offer unique insights into the properties of glacial ice. In the past, modeling and measurements focused on the impact of air-borne impurities as deposited with the original snow accumulation on absorption and scattering and their stratig-
raphy. This in particular yielded the most stringent upper limit as compiled by Warren and Brandt (2008) on the absorption coefficient of pure ice, as measured in the deepest parts of the detector.

Here we have described the observation of an ice optical anisotropy, a direction dependent intensity modulation aligned with the local ice flow axis. The effect has been identified to largely result from diffusion within the polycrystaline ice microstructure, resulting in a previously unknown optical effect: a slow but continuous deflection towards the normal vector of
the girdle plane of the crystal orientation fabric. Combining prior knowledge about the crystal orientation fabric and average grain elongation as obtained from SPC14, the depth-dependent average crystal size has been fitted to IceCube LED calibration



data. The resulting depth evolution conforms to the expectation of larger crystals at greater depth and an inverse correlation to impurity concentrations.

The first-principle birefringence explanation was not able to fully describe the experimental data. This could be improved

upon by including an ad-hoc Mie absorption anisotropy, for which no first principle explanation is known. The origin of this remaining discrepancy will hopefully be resolved using upcoming instrumentation in the IceCube Upgrade, modeling of ice intrinsic diattenuation, as well as future lab measurements regarding the photoelasticity of ice.

Overall the large variety of measurements performed in close vicinity to the Amundson-South Pole Station (optical data from IceCube and its upcoming detector upgrade, the SPC14 ice core, ground penetrating radar data from PolarGap (Forsberg

et al., 2015) and others, GPS stake field fields such as (Lilien et al., 2018), among many) make the geographic South Pole a unique laboratory for comparative measurements. Yet to date, the overlap in sampled depth between SPC14 and IceCube is unfortunately too small to allow for quantitative comparison. This may be resolved by future drilling projects such as a potential deployment of the RAID drill (Goodge et al., 2021).

*Code and data availability.* The photon propagator software (PPC), compatible ice model configurations including the model derived in

this work, as well as the electromagnetics code used to generate the diffusion patterns are available from https://github.com/icecube/ppc. IceCube raw data, including the LED calibration data, is generally not publicly available. For specific inquiries please contact analysis@icecube.wisc.edu.





## Appendix A: Sampling c-axes distributions from the eigenvalues of ice fabric orientation tensors

One can describe the crystal orientation fabric of $N$ c-axes, measured in an ice sample, by $N$ unit vectors $\boldsymbol{n}_i$, with components $n_{ix}, n_{iy}, n_{iz}$. Note that $\boldsymbol{n}_i$ is equivalent to $-\boldsymbol{n}_i$ as the vector can be chosen to point along either direction of the axis. By convention $\boldsymbol{n}_i$ are chosen to point upward. Scheidegger (1965) suggested to represent this ensemble of vectors via the matrix:

$$
a = \begin{bmatrix} \sum n_{ix}^2 & \sum n_{ix} \cdot n_{iy} & \sum n_{ix} \cdot n_{iz} \\ \sum n_{iy} \cdot n_{ix} & \sum n_{iy}^2 & \sum n_{iy} \cdot n_{iz} \\ \sum n_{iz} \cdot n_{ix} & \sum n_{iz} \cdot n_{iy} & \sum n_{iz}^2 \end{bmatrix} \tag{A1}
$$

The normalized form $A = a/N$ is called the second-order orientation tensor. It was introduced in glaciology through Gödert and Hutter (1998). $A$ has three eigenvectors and three corresponding eigenvalues $S_1$, $S_2$ and $S_3$, with $S_1 + S_2 + S_3 = 1$.

The axes of the coordinate system in which the c-axes are evaluated can be chosen such that the x-axis points along the mean c-axis direction $\sum \boldsymbol{n}_i/N$, that the z-axis points along the pole to the best-fit girdle to the distribution (see Woodcock (1977)) and that the y-axis is orthogonal to the other two. In this case the coordinate axes are the eigenvectors, $S_j = \sum_i n_{ij}^2$ and the eigenvalues follow a strict ordering such that $S_1 \geq S_2 \geq S_3$.

A perfectly uniform, girdle or unimodal fabric features the following relations between the eigenvalues:

- uniform:     $S_1 \approx S_2 \approx S_3 \approx 1/3$
- unimodal:    $S_1 \approx 1; \quad S_2 \approx S_3 \approx 0$
- girdle:       $S_1 \approx S_2 \approx 0.5; \quad S_3 \approx 0$

Woodcock (1977) realized that all possible fabric states can be visualized in a 2D plot, as only two of the three eigenvalues are independent. He suggested the representation where the abscissa is given as $\ln(S_2/S_3)$ and the ordinate is given as $\ln(S_1/S_2)$. In this representation uniform c-axis distributions are found at the origin of the plot. The distance from the origin $C = \ln(S_1/S_3)$ is called the strength parameter. Girdle fabrics are found to the lower-right, while unimodal fabrics reside to the upper-left. The type of fabric can also be quantified by the so called Woodcock shape parameter $K = \ln(S_1/S_2)/\ln(S_2/S_3)$. Large $K$ values denote a unimodal fabric. $K$ values smaller than 1 denote a girdle fabric.

Neither the orientation tensor nor its eigenvalues retain the full information on the ensemble of underlying c-axes. Thus, an assumption on the functional form of the fabric has to be made when trying to sample a distribution. Here we focus on describing random, girdle and unimodal distributions, as well as combinations of these, as those are the types most commonly encountered in ice fabric measurements, with SPC14 in particular featuring very strong girdle fabric.

The book "Statistical analysis of spherical data" by Fisher et al. (1987) gives a good overview of commonly used probability density functions (PDFs) for directional data. Of the presented PDFs the Watson (1965) distribution seems most applicable for our case as:

1. It can represent both unimodal as well as rotational symmetric girdle data.
2. There exists an (approximate) parameter estimation based on eigenvalues alone.



In its standardized form the PDF, evaluated on a spherical coordinate system with the polar angle $\theta$ and the azimuth angle $\phi$, has only one free parameter $\kappa$ and is given as:

$$f(\theta,\phi) = C_w \exp(\kappa \cdot \cos^2\theta)\sin\theta \qquad (A2)$$

with the normalization constant

$$C_w = 1 / \left(4\pi \int_0^1 \exp(k \cdot u^2) du\right). \qquad (A3)$$

In the following the Python package available at https://github.com/duncandc/watson_distribution is used to sample from the Watson distribution. Alternatively the sampling approach described in Fisher et al. (1987) may be used.

At $\kappa = 0$ the direction distribution is perfectly uniform. For positive $\kappa$ the distribution is bimodal in vector space which is equivalent to a unimodal distribution in axis space and has the highest probability at the poles. For negative $\kappa$ values the distribution is girdle with the directions equally distributed around the equator.

Best and Fisher (1986) showed that for a purely unimodal distribution the $\kappa$-parameter can be estimated from the eigenvalues as

$$\kappa = \begin{cases} 3.75 \cdot (3 \cdot S_1 - 1), & \frac{1}{3} \le S_1 \le 0.34 \\ -5.95 + 14.9 S_1 + \frac{1.48}{1 - S_1} - \frac{11.05}{S_1^2}, & 0.34 < S_1 \le 0.64 \\ -7.96 + 21.5 \cdot S_1 + \frac{1}{1 - S_1} - 13.25 \cdot S_1^2, & S_1 > 0.64, \end{cases} \qquad (A4)$$

while for a purely girdle fabric the $\kappa$-parameter can be estimated as

$$\kappa = \begin{cases} \frac{1}{2 \cdot S_3}, & 0 \le S_3 \le 0.06 \\ 0.961 - 7.08 \cdot S_3 + \frac{0.466}{S_3}, & 0.06 < S_3 \le 0.32 \\ 3.75 \cdot (1 - 3 \cdot S_3), & 0.32 < S_3 \le \frac{1}{3}. \end{cases} \qquad (A5)$$

For ice fabrics, the plane of girdle c-axes shall intersect the poles, where also the c-axes of a unimodal distribution are found. As such the directions sampled from girdle Watson distributions are rotated by $90°$. Due to the underlying rotational symmetry 730 the eigenvalues of the resulting Watson distributions follow a strict relation:

$$S_1 = S_2 \qquad \& \quad S_3 = 1 - 2 \cdot S_1 \quad \text{for a girdle Watson}$$
$$S_2 = S_3 \qquad \& \quad S_1 = 1 - 2 \cdot S_2 \quad \text{for a unimodal Watson.} \qquad (A6)$$

Obviously no single Watson distribution can describe an arbitrary set of eigenvalues with $S_1 \ne S_2 \ne S_3$. This is achieved by combining directions sampled from a girdle and a unimodal Watson distribution.

Given a sample of c-axes from a girdle Watson distribution with eigenvalues $S_{ig}$ and a sample of c-axes from a unimodal Watson distribution with eigenvalues $S_{iu}$, as well as a relative fractional contribution of the girdle sample $f_g$ to the total sample, the eigenvalues of the combined sample $S_i$ are given by: $S_i = f_g \cdot S_{ig} + (1 - f_g) \cdot S_{iu}$ The combination of $f_g$, $S_{1g}$ and $S_{2u}$





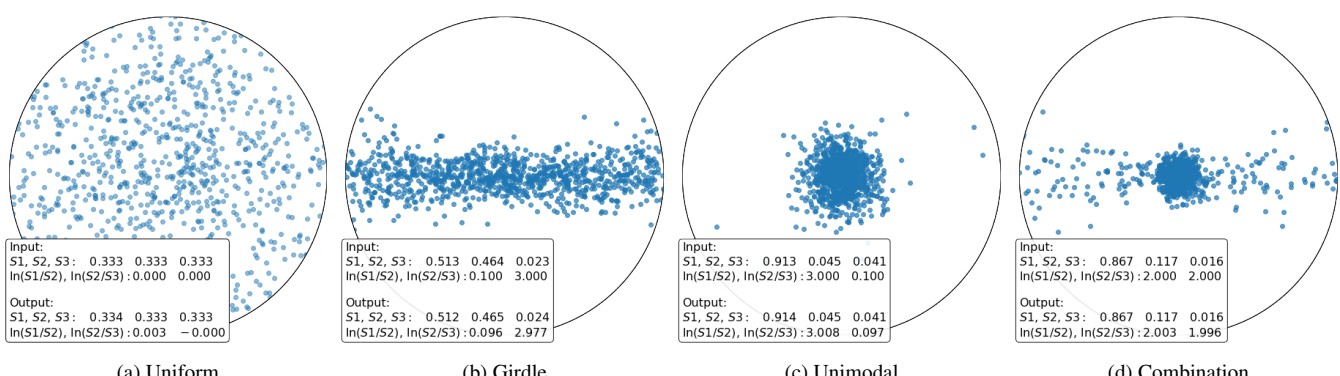

| (a) Uniform | (b) Girdle | (c) Unimodal | (d) Combination |

**Figure A1.** Example c-axis distributions in the usual Schmidt equal-area projection generated using the described method.

which yields the desired eigenvalues $S_1$, $S_2$ and $S_3$ is found by solving the equation system $S_i$, which has been simplified using the relations in Equation A6:

$$S_1 = f_g \cdot S_{1g} + (1 - f_g) \cdot (1 - 2 \cdot S_{2u}),$$

$$S_2 = f_g \cdot S_{1g} + (1 - f_g) \cdot S_{2u},$$

$$S_3 = f_g \cdot (1 - 2 \cdot S_{1g}) + (1 - f_g) \cdot S_{2u} \tag{A7}$$

The third equation is not independent since $S_1 + S_2 + S_3 = 1$. Thus further information is needed to be able to constrain the variables. To fulfill the assumption that $S_u$ is unimodal and $S_g$ is girdle one can further constrain $1/3 < S_{1g} < 0.5$ and $0 < S_{2u} < 1/3$. For cases where the system is still underconstrained one can for example further demand that $S_{2g} = S_{2u} = S_{1g}$ (equivalent to $S_{1u} + S_{3g} = 1$), so that both distributions have an equal spread around the girdle plane. The solution is then given as:

$$f_g = 0.5 \cdot (\epsilon - 4S_1 - 2S_2 + 3)$$

$$S_{1g} = \frac{2S_1}{-\epsilon + 4S_1 + 2S_2 + 1}$$

$$S_{2u} = \frac{\epsilon - 2S_2 - 1}{2 \cdot (\epsilon - 4S_1 - 2S_2 - 1)} \tag{A8}$$

with: $\epsilon = \sqrt{16 \cdot S_1^2 + 16 S_1 \cdot (S_2 - 1) + (2S_2 + 1)^2}$. From these one can derive the Watson parameters $\kappa$ using the approximations as given in Equations A4 and A5.

To verify and visualize the success of the presented sampling approach, c-axes distributions according to a number of combinations of $\ln(S_1/S_2)$ and $\ln(S_2/S_3)$ have been generated as shown in Figure A1. The sampled c-axes distributions yield eigenvalues which are accurate to within the approximation of the parameter estimation for the Watson distributions and well sufficient for most applications.

Note that by design the c-axes distributions for intermediate fabric states do not contain a single elliptical distribution but a rotationally symmetric girdle and a circular unimodal. This seems suitable for our application to ice fabrics. In very deep





glacial ice where the fabric slowly evolves from girdle to unimodal, experimental distributions such as published by Weikusat
et al. (2017) indeed show the described superposition and not an elliptical distribution usually sketched for these eigenvalues.

### Appendix B:  Sampling surface orientations from an ellipsoid

As the average grain shape deviates from a sphere, the encountered distribution of face orientations depends on the photon
direction. Assuming that the face orientation of a solid, tessellated into elongated polyhedra, to be described by the surface
orientation density of an ellipsoid describing the average grain shape, one can sample the distribution as follows.

The surface of an ellipsoid is defined by the equation,

$$f(x', y', z') = \frac{x'^2}{a^2} + \frac{y'^2}{b^2} + \frac{z'^2}{c^2} = 1 \tag{B1}$$

where $a$, $b$ and $c$ are the dimensions of the major and minor axes. The normal vector on any point of the surface is given by the
gradient

$$\nabla f = [2 \cdot \frac{x'}{a^2}, 2 \cdot \frac{y'}{b^2}, 2 \cdot \frac{z'}{c^2}]. \tag{B2}$$

For a given set of azimuth and zenith angles, the coordinates on a unit sphere $(x, y, z)$ and on the ellipsoid $(x', y', z')$ are given
as:

$$x = \sin\theta \cdot \cos\phi \quad \text{and} \quad x' = a \cdot x \tag{B3}$$

$$y = \sin\theta \cdot \sin\phi \quad \text{and} \quad y' = b \cdot y \tag{B4}$$

$$z = \cos\theta \quad \text{and} \quad z' = c \cdot z \tag{B5}$$

Substituting the ellipsoid surface position into Equation B2 the surface normal at this position is then:

$$\boldsymbol{n} = [\frac{2}{a} \cdot \sin\theta \cdot \cos\phi, \frac{2}{b} \cdot \sin\theta \cdot \sin\phi, \frac{2}{c} \cdot \cos\theta]. \tag{B6}$$

One can now sample these gradients with angles chosen to be uniform on a sphere. As the surface density per solid angle of an
ellipsoid is different from a sphere, the relative surface density

$$\mu_{(x,y,z)} = ||dS'||/||dS|| = \sqrt{(ac \cdot y)^2 + (ab \cdot z)^2 + (bc \cdot x)^2} \tag{B7}$$

has to be applied as a weighting factor, where the maximum weighting factor is given as

$$\mu_{\text{max}} = \max(ac, ab, bc). \tag{B8}$$

Instead of weighting, one can also employ a rejection sampling with an acceptance probability of $\mu/\mu_{\text{max}}$.

    In addition to the distribution of face orientations, the distribution of face orientations actually encountered by a photon
can be obtained by weighting the distribution of face orientations with the scalar product of the photon's propagation vector





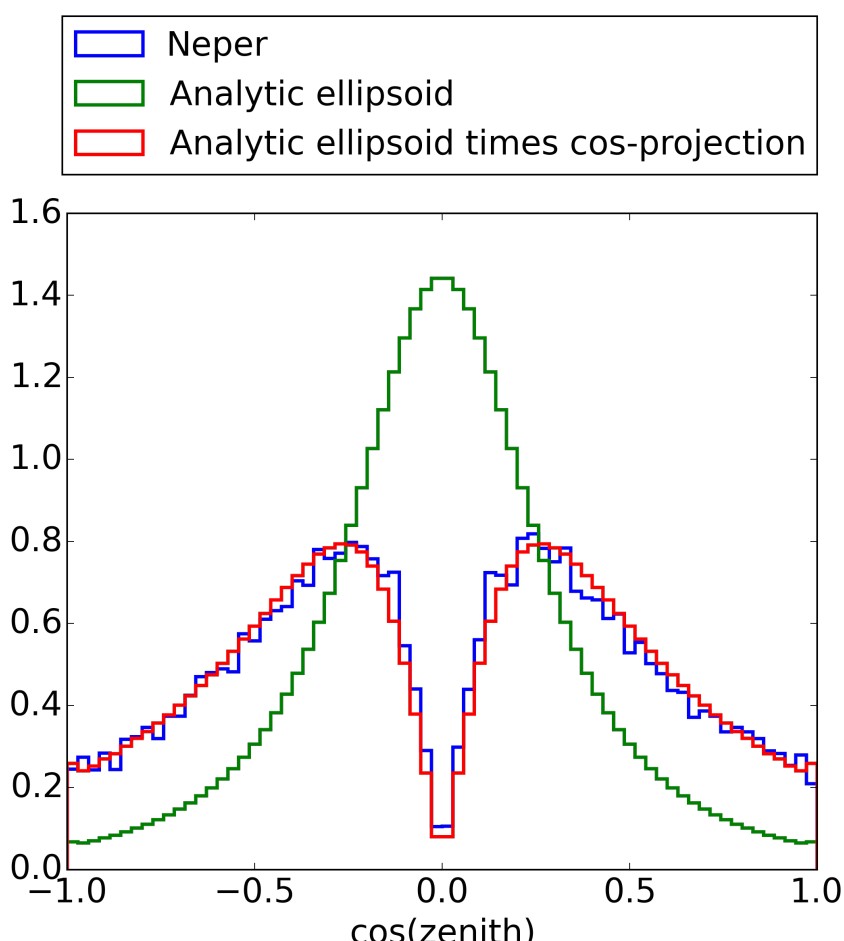

**Figure B1.** Ellipsoid surface sampling for an ellipsoid with unity minor axes and a major axes of two along the z-axis. Green: Analytic $\cos(\theta)$ distribution of face normal vectors. Red: Analytic $\cos(\theta)$ distribution weighted by the encounter probability, given by the scalar product with a photon propagating along z. Blue: Encounter probability as found in a Neper crystal tessellation simulation when tracing photons along vertical lines.



and each face normal vector. The probability to encounter a given plane is therefore simply the projected area relative to the incident light.

Figure B1 shows the $\cos(\theta)$ distribution of (encountered) face normal vectors for a spheroid with elongation two, which has the major axis aligned with the z-axis. The distribution is compared to a crystal-like Voronoi tessellation generated with Neper and assuming the same mean elongation. Lines have been traced through the tessellation, identifying grain boundary encounters and computing their incidence angles. The distributions are found to be indistinguishable, confirming that the ensemble of polyhedra faces follows the average ellipsoid.

*Competing interests.* The authors declare that they have no competing interests (both financial or non-financial).

*Author contributions.* The IceCube Collaboration designed, constructed and now operates the IceCube Neutrino Observatory. Data processing and calibration, Monte Carlo simulations of the detector and of theoretical models, and data analyses were performed by a large number of collaboration members, who also discussed and approved the scientific results presented here. The IceCube collaboration acknowledges the substantial contributions to this manuscript from Martin Rongen and Dmitry Chirkin. The manuscript was reviewed by the entire collaboration before publication, and all authors approved the final version.

*Acknowledgements.* The authors gratefully acknowledge the support from the following agencies and institutions: USA – U.S. National Science Foundation-Office of Polar Programs, U.S. National Science Foundation-Physics Division, U.S. National Science Foundation-EPSCoR, Wisconsin Alumni Research Foundation, Center for High Throughput Computing (CHTC) at the University of Wisconsin–Madison, Open Science Grid (OSG), Extreme Science and Engineering Discovery Environment (XSEDE), Frontera computing project at the Texas Advanced Computing Center, U.S. Department of Energy-National Energy Research Scientific Computing Center, Particle astrophysics research computing center at the University of Maryland, Institute for Cyber-Enabled Research at Michigan State University, and Astroparticle physics computational facility at Marquette University; Belgium – Funds for Scientific Research (FRS-FNRS and FWO), FWO Odysseus and Big Science programmes, and Belgian Federal Science Policy Office (Belspo); Germany – Bundesministerium für Bildung und Forschung (BMBF), Deutsche Forschungsgemeinschaft (DFG), Helmholtz Alliance for Astroparticle Physics (HAP), Initiative and Networking Fund of the Helmholtz Association, Deutsches Elektronen Synchrotron (DESY), and High Performance Computing cluster of the RWTH Aachen; Sweden – Swedish Research Council, Swedish Polar Research Secretariat, Swedish National Infrastructure for Computing (SNIC), and Knut and Alice Wallenberg Foundation; Australia – Australian Research Council; Canada – Natural Sciences and Engineering Research Council of Canada, Calcul Québec, Compute Ontario, Canada Foundation for Innovation, WestGrid, and Compute Canada; Denmark – Villum Fonden, Carlsberg Foundation, and European Commission; New Zealand – Marsden Fund; Japan – Japan Society for Promotion of Science (JSPS) and Institute for Global Prominent Research (IGPR) of Chiba University; Korea – National Research Foundation of Korea (NRF); Switzerland – Swiss National Science Foundation (SNSF); United Kingdom – Department of Physics, University of Oxford.





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
