# Peer review of "In-situ estimation of ice crystal properties at the South Pole using LED calibration data from the IceCube Neutrino Observatory"

_The Cryosphere, 2022_

## Author Comment (AC1)

**Reply to review report by Anonymous Referee #1 submitted on the 10th of December 2022**

The IceCube Collaboration

March 8, 2023

Review of the manuscript entitled, "In-situ estimation of ice crystal properties at the South Pole using LED calibration data from the IceCube Neutrino Observatory" submitted to The Cryosphere.

This study discusses propagation patterns of the $\sim$400 ns wavelength light wave (or photons) within a thick ice sheet at South Pole, Antarctica, with a local condition of the crystal preferred orientation (CPO), grain shape and size of polycrystalline ice. The authors (a team of very many people) used 5000 photomultipliers installed within the ice sheet, as IceCube Neutrino Observatory instruments. Through procedures for calibration of the light emitters and the receivers, the authors discovered unexpected light propagation effect within the ice sheet. The authors described it as "an anisotropic attenuation". It is orientation-dependent variations of the received light, which has directional dependence of wave propagation along the flow direction of the ice sheet. The authors examined birefringence of ice crystal grains, that is, anisotropic properties of reflections and refractions at ice grain boundaries, birefringent propagation within each crystal grains, as well as additional effects from grain shapes and sizes as possible main causes for the observed effect. The authors examined both orientation dependent scattering effects due to impurity-related inclusions and various effects related optical birefringence of ice. They claimed that they made a model that is quantitatively accurate for optical properties of the IceCube glacial ice. They found an explanation as to why curved light trajectories occur resulting from asymmetric diffusion. To harmonize between simulated effects using their model and the observational data in the field laboratory, the authors raised a possibility that ice crystal may have anisotropic absorption properties and stress-strain induced changes in optical properties, which is still unknown in the research field of ice physics.

We would like to sincerely thank the reviewer for the significant time investment in reviewing the manuscript and the constructive and encouraging feedback.

**General comments**

[G1] This is an interesting and informative paper in which the authors examined propagation of light wave (photons) within the thick polar ice sheet, using a local area ( 1 km3) of the ice sheet as a field laboratory. Ice has special crystal orientation fabric and undulation of internal layers in the vicinity. To my knowledge, there seems no prior studies who found or explained propagation of light wave within a condition of a particular crystal orientation fabric. Basic investigations that the authors performed seem sound, such as CPO, grain shape and size, and propagation of light wave within polycrystalline ice.

I must confess that for a few sub-sections of the manuscript, I found difficulty to understand.

The study also clarified that there were still many major unknowns, such as anisotropy in imaginary part of the refractive index of ice single crystals (in crystal lattice) or stress/strain induced properties. I must note that there is major uncertainty in the discussions. Possibilities that the authors raised – ice crystal lattice should have both anisotropic absorption properties and stress-strain induced changes in optical properties – may be either correct or incorrect. If it is correct, we must find yet-unexplored physical properties of ice. If the hypothesis is incorrect, there can be major points to reconsider, repair or modify in their models. Questions are open now. My concern is that in this study there are accumulated errors in model simulations, and then possibly false reasons are examined attempting to explain them, without knowing they are errors.

We fully acknowledge and hope to have candidly outlined in the manuscript that while we believe the newly discovered optical effect to be well motivated, rigorously derived and tested (using for example two independent simulation approaches), the necessity to include the ad-hoc absorption anisotropy in order to arrive at the best data-simulation agreement and reasonable grain sizes indicates that we are still missing a vital component. This could be an entirely new, yet unaccounted-for effect or an incompleteness in the modeling of the discussed birefringence effect. Such has been our experience extracting the optical properties of ice surrounding our detector, on more than one occasion, that a newly gained insight often leaves one with more questions to be answered.

We generally acknowledge the concern of accumulated errors given the multitude of components involved in this analysis. Yet without concrete pointers where our work may be insufficient, we can only present the methodology and results achieved so far. It is also worth mentioning that we went through a large number of iterations regarding the parametrization and application of diffusion patterns in photon propagation simulation (sections 6.1 and 6.2), which are not discussed in the manuscript. The presented parametrization was chosen for chosen for its simplicity, ease of scaling to arbitrary distances and speed of application during photon propagation simulation. Yet the other approaches did not result in vastly different results.

[G2] This paper is useful and important for people who analyze Cherenkov light in the IceCube project. They need to know accurately how light wave have nature of light propagation in glacial ice. In terms of cryospheric sciences, it may provide insights into crystal properties in the ice sheet. What seems robust and new is light propagation within preferred orientation of ice lattice, grain shape and size and grain boundary network. In terms of cryospheric sciences, immediately useful practical applications are not clear at least for me.

We concur with the reviewers assessment that in the cryospheric context the presented work is as of now primarily of academic interest. Our hope in attempting to publish this work in a cryospheric journal is to draw attention to this unusual aspect of ice study and hopefully inspire practical followup work in the future.

[G3] In this paper, many factors that can affect propagation of light wave were discussed. However, I felt that they are not well listed or summarized to be understood comprehensively. The authors tend to list them one by one in different sections within the manuscript. Sometimes they were mentioned only conceptually and vaguely such as "the first-order principles". I suggest that the authors should provide a table (or tables) as supplementary information, listing items, shape and size of the items, state of presence (or distribution), possible effects in terms of light wave propagation, reference papers, and notes (such as unknown, hypothesized by this work, and so on). Such tables will help better understanding. Possible items that are useful for readers are, for example, as follows.

1. Ice matrix items that can cause effects of reflection, refraction, scattering or absorption.
   - Anisotropic refractive index (real part) of ice in each crystal grain
   - Crystal Preferred Orientation
   - Grain shape and size
   - Distribution network of grain boundaries in terms of ice fabric, grain shape and size
   - Distribution of triple junctions of grains in terms of ice fabric, grain shape and size (The authors did not mention it. But it is one of major locations for presence of impurities. See Stoll et al. (2021) given in [D19] below.)
   - Anisotropic refractive index (imaginary part) of ice in each crystal grain

2. Clathrate hydrate inclusions: Number density, size distribution and possible localization within ice matrix.

3. Various inclusions
   - Dust (Number density, size distribution and possible localization within ice matrix).
   - Salt particles (Number density, size distribution and possible localization within ice matrix).
   - Acids (State of presence and possible localization at triple junctions)
   - Soot (Number density, size distribution and possible localization within ice matrix)
   - Volcanic ash (Number density, size distribution and possible localization within ice matrix)

Possible alignment of these items along some orientations.
Possible effects from stress/strains or pressures.
Also, the authors cited an old textbook by Hobbs published in 1960s. Rather than citing such old very thick textbook, I suggest the authors to cite responsible original papers which really addressed points. For readers it is hard to find out points of discussions in thick textbooks. Also, similar situation occurs when citation is PhD thesis.

To shortly comment on the categories and some additional items raised:

- Category I is the topic of this paper. The refractive index, CPO and grain size & shape are all discussed in section 5. But, in contrast to the different impurity constituents, they can not be considered individually as all three are needed to obtain a diffusion pattern. An attempt at visualizing the relative impact on the diffusion is for example given in Figure 13.

- The existence of an anisotropic imaginary refractive index is at this point speculative and should thus not be included in the suggested table. While it is a natural extension to the model, and as such mentioned, it is an unlikely explanation as the ice crystal intrinsic absorption is vanishingly

small compared to absorption on impurities. We will add an according statement to section 8 (Outlook).

- As a follow-up to D19 (continued), we will quickly mention air-hydrate in section 2.3 (glacial ice as an optical medium) and explain why it is not quantitatively considered.

- The impurity constituents and their impact on the optical properties were essentially the topic of most ice calibration-related papers by IceCube prior to this manuscript (most notably Ackermann 2006 and Aartsen 2013). So not to further expand the already long manuscript, we do not wish to provide more detail than already present in sections 2.3 and 3.4. It shall here be noted though that while the size distributions and imaginary refractive indices of the four constituents are qualitatively different enough to categorize them as either Mie or Rayleigh scatterers and weakly or strongly absorbing, their number density and size distribution (with particular interest around 400 nm) is not sufficiently known (at the South Pole) to base optical modeling on these input parameters. Instead, the ice optical properties is obtained by comparing LED calibration data to an phenomenological model which is based on the physical expectation resulting from these impurity types.

- Please also note that prior to investigating birefringence, the distribution of impurity (preferentially along grain boundaries and in triple junctions) was in fact studied as a possible cause for the observed optical anisotropy. With no consensus information about the orientation of elongated impurities available, Rongen 2019 restricted themselves to Mie scattering scenarios and was able to conclude that no inhomogeneous impurity distribution can lead to a large-scale anisotropy. To inform the reader, that this natural seeming approach was attempted but rejected, we suggest adding the following sentence to section 4.3 (Early empirical modeling):

  Alternatively, a directionality of Mie scattering may be believed to be the result of inhomogeneous impurity distributions, with the different impurity types known to aggregate on the grain boundaries with different probabilities and through different mechanisms (Stoll et al., 2021b; Durand et al., 2006). Yet the derivation of Mie scattering properties only depends on the volumetric particle densities and is independent of homogeneity. In the context of studying the ice optical anisotropy, Rongen (2019) explicitly tested this in a number of simulated toy experiments and verified that inhomogeneous impurity distributions can not lead to a large-scale anisotropy.

The suggested summary table, as part of the appendix, seems helpful and will be included in the next revision, with a preview given below.

| Type | Absorption | Scattering | | How modeled |
|---|---|---|---|---|
| Impurities | | | Total | |
| Soot | strong | Rayleigh (isotropic) | | combined absorption and |
| Mineral dust | strong | Mie (forward) | | scattering coefficients in |
| Salts | weak | Mie (forward) | >90% | 10 m tilt corrected layers |
| Acids | weak | Mie (forward | | (see sections 2 and 3) |
| Polycrystaline micro-structure | none | Asymmetric diffusion | <10% | scattering and deflection (see sections 5 and 6) |

Table 1: Conceptual overview of different constituents considered as part of the ice optical modeling. For details on the behavior of different impurities please see Ackermann, 2006 and He and Price 1998. The polycrystalline micro-structure leading to asymmetric diffusion is newly considered in this work.

Hobbs is only cited with regard to the *'anecdotally known photoelasticity of ice'*, with the only measurement known to us (Ravi-Chandar (1994)) being cited in the following sentence. The field of ice optical properties in deep glacial ice is rather small and we aim to cite original work where possible (Aartsen 2013, Ackermann 2006, Chirkin 2013, Chirkin 2020, He 1998, Price 1997, Price 2000, Rongen 2020, Rongen 2021, Warren 2008). So not to hide already publicly available material on the topic and to allow the reader to follow up on details, we think it pertinent to cite PHD theses and conference proceedings, where no other citable material exists yet. To some extent this paper also serves to carry forward the work started in the cited theses and proceedings to a full publication.

We wish to stay with our phrasing of 'first-principles', which is supposed to indicate that the birefringence model has been derived from the ground up (starting with Maxwell's equations and the refraction on a single grain boundary transition). This, to us, is an important distinction from previous empirical parametrization attempts (the absorption or scattering anisotropy).

> [G4] As for citations for the glaciology-based information, there are points that seems necessary to be updated, or more proper papers should be cited. In detailed comments, I will comment one by one.

Being primarily of a different field of research, navigating the glacilogical literature is indeed at times a struggle and we would like to thank the reviewer for the suggestions made in the detailed comments, many of which we propose to adopt as will be detailed below.

> [G5] My concern as for the structure of the paper is sequential order of Section 4.3 and Section 5. Section 4.3 discusses early empirical modelling, given just after the observational results (Sections 4.1 and 4.2). When we think about glacial ice as a media of light propagation, it seems natural that light diffusion in birefringent polycrystals should be taken as one of substantial bases. This phenomenon should have priority to be discussed. And then, empirical modelling related to absorption should be provided as the item with secondary priority because in this aspect modelling attempt does not seem very successful to explain observations. In addition, assumptions for directional dependence for absorption seem to be suffered from lack of observational evidence.
>
> For readers, they need to be informed, with an order of importance. Present order will tend to confuse readers, as I felt so.

As introduced / reviewed in section 2.3 (*Glacial ice as an optical medium*) ice optical properties are largely driven by impurities and the direction averaged optical properties are well described using just these. We understand that given experience in radar measurements or ice microstructure the birefringent nature of ice may seem like the natural starting point, but for the ice optical properties it is not.

We hope that the introductory sentence to section 4.3 (*Following the paradigm that ice optical properties are driven by Mie scattering on impurities, early attempts tried to model the anisotropy through directional modifications of absorption and scattering.*) ensures that the reader knows this section to be merely a stepping stone with not too much emphasis given. In addition we propose to significantly reduce detail in this section (see G6), such to further de-emphasize it. Swapping the section order to us will not provide any benefit, as the parametrizations described therein do not prove helpful in understanding the physical origin of the effect as described by the bulk of the following sections. The modifications to absorption and scattering are here primarily introduced so to be available when resorting to an additional absorption anisotropy at the end of section 6.3

> [G6] Length of the manuscript
> Main text alone has 14,000 words. The paper is very long. For a better readability, this can be more concise, by sending some parts to supplementary information.

We acknowledge that the manuscript is very long, maybe even to the point of hindering accessibility as also pointed out by the other reviewer. Yet given we rarely publish in glaciological journals (the last instance being Ackermann2006) and that a large fraction of the target audience is likely neither familiar with ice optical properties nor the IceCube detector, we find a short summary of all aspects required to perform this kind of measurement is pertinent. Otherwise the reader is left to explore a large body of work in an unfamiliar field before being able to study the novel aspects of this manuscript.

We propose the following cuts:

- Equations 2, 3, 4 and 5 shall be cut along with the accompanying text and replaced with more conceptual descriptions.

- Figures 2 and 16 shall be cut without replacement.

- Lines 234 to 240 shall be cut without replacement.

- Section 4.3 (Early empirical modeling) shall be replaced with the following suggestion, removing most details regarding the mathematical modeling:

  Following the paradigm that ice optical properties are driven by Mie scattering on impurities, early attempts tried to model the anisotropy through directional modifications of absorption and scattering. In the original parameterization presented by Chirkin(2013d), it was argued that due to time and space reversal symmetries the absorption length and geometric scattering length cannot be direction dependent. Therefore the anisotropy was implemented as a modification to the scattering function, the only remaining Mie scattering parameter. This effectively results in a change of the effective scattering coefficient as a function of the propagation direction. Photons propagating along the flow axis experience less scattering than photons propagating along the tilt axis or inclined from the horizontal.

  While not derived from first-principle Mie calculations, the parametrization was justified to be a plausible result of elongated impurities becoming preferentially aligned by the flow and thus introducing a direction dependence to the scattering function. While several glaciological studies (Potenza et al., 2016; Simonsen et al., 2018; Gebhart, 1991) explore the shapes of impurities, elongations for different impurities are not well established, nor is there to our knowledge any evidence for elongated impurities becoming oriented with the flow.

  An evaluation of the data-simulation agreement is shown in Figure 8. It shows summed photon arrival time distributions for all nearest emitter-receiver pairs, roughly aligned along and perpendicular to the ice flow for a variety of anisotropy models and the employed flasher data. The scattering-based anisotropy model results in more intensity being observed along the flow axis. However, there remains substantial disagreement between the model and the observed data. As scattering is reduced in the flow direction light arrives earlier on average. The resulting change in the rising edge position is strongly penalized in the fit and limits the amount of intensity that can be recovered. To reduce the shift of the rising edge, a directional modification to Mie absorption was considered as an alternative by Rongen (2019). A factor 11 modulation of the absorption coefficient was required to fit the data, which seems unphysical. As evident from Figure 8, this model results in a delayed rising edge for propagation along the flow direction as desired, and did result in an improved data description compared to the scattering based model described earlier, but is also unable to fully match the intensity difference to data.

To conclude, while resulting in partially successful effective descriptions, directional modifications to Mie scattering or absorption cannot reproduce observations nor are such modifications well motivated on first principles.

In addition, we propose to expand lines 67-74 (describing the structure of the manuscripts) as follows:

This manuscript has the following structure: Section 2 introduces the IceCube Neutrino Observatory (Sections 2.1 and 2.2) and how it employs ice as a detection medium (Section 2.3). Section 3 describes the properties of the LED calibration data used in this study (Section 3.1), explains the photon propagation software used to generate simulated data (Section 3.2) and details the likelihood analysis comparing simulated to experimental data in order to infer ice properties (Section 3.3). Section 3.4 briefly reviews the state of the isotropic, layered model used to describe the ice optical properties prior to this work. The experimental signature of the ice optical anisotropy (Section 4.1) as well as early modeling attempts (Section 4.3) are summarized in section 4. The newly developed model to account for the ice optical anisotropy based on the ice-intrinsic birefringence is described starting with Section 5. Sections 5.1 and 5.2 explain the electromagnetic theory governing the birefringence in polycrystals, while Section 5.3 introduces a software package to simulate the resulting diffusion patterns. Section 5.4 compares the experimental signatures and conceptual understanding of the underlying optics to birefringence observations in radar sounding, a field most readers are probably more familiar with. Section 6 explains how the diffusion patterns are applied in the IceCube photon propagation simulation (Sections 6.1 and 6.2) and how crystal properties have been inferred (Section 6.3). Section 7 describes the resulting ice optical model. Section 8 discusses shortcomings of the model as well as future measurements in upcoming IceCube extensions and through drill-hole logging.

While this adds some additional text, we hope that this additional context allows the reader to judge which parts of the manuscript are relevant for their particular interests/expertise.

The different gaps in explanations, alternative aspects to be added and areas of excessive details identified by the two reviewers highlight that arriving at a fully concise manuscript is futile given the diametral demands, but we hope to strike a balance here.

**Detailed comments**

[D1] Abstract, Lines 4-5:
I suggest that "Birefringent light propagation has been examined" can be modified as "Birefringent light propagation through networks of ice grain boundaries has been examined" (or something like this) to stress substantial points, medium of propagation, in this study. It seems that birefringence is one of components in the model. But it seems that grain boundary network closely related to ice fabric is also one of essential conditions. The authors termed as "birefringence model". My concern is adequacy of this vague wording. Please choose more concrete terms.

Regarding the first sentence we concur and propose the following rewording:

Birefringent light propagation through the polycrystalline ice microstructure has been examined as a possible explanation for this effect.

Regarding the term "birefringence model" in line 5, we propose to simply drop the qualifier "birefringence" in this case as it is not needed.

[D2] Abstract in general:

Readers of this paper will wonder if this paper discusses polycrystalline properties, ice lattice properties within single crystal, or both. Indeed, this paper discusses both. Please consider a possibility that the authors already mention these key points in the abstract. It seems fairer then.

While single-crystal properties are discussed as part of the derivation, the presented model (and so the new insight presented here) takes single-crystal properties (the refractive indices for example) solely as input parameters and is only sensitive to the listed polycrystal properties.
We think that the rewording resulting from D1 clarifies the scope of the paper at the earliest possible point in the abstract.

[D3] Abstract, Lines 6-8: Only polycrystalline properties are given. How about lattice properties?

Please see the response to D2.

[D4] Citations in general:

It seems to me that several citations require "e.g.," because they are not unique choice of possible citations. In the introduction, Cuffey, McConnel, Faria, and Alley papers (books) are such examples. Cuffey citation is a textbook where established knowledge is reviewed. McConnel is a very old paper. Anisotropy in plastic deformation of ice was reviewed in many textbooks of ice, such as Hobbs, Petrenko&Whitworth etc. Rather than giving only one, the most original paper, it is beneficial for readers to find recent textbooks as well. Alley 1988 is one of papers in which the measurement method was applied. I suggest that useful method papers for readers include, as follows.

- *Langway, C. C. J. (1958), Ice fabrics and the universal stage., SIPRE Tech. Rep., 62.*

- *Wilen, L., C. L. DiPrinzio, R. B. Alley, and N. Azuma (2003), Development, principles and applications of automated ice fabric analyzers, Microscopy Research and Technique.*

We are happy to mark exemplary citations with 'e.g' and will also ask the editor if this is the preferred citation style for this journal. The two suggested additional references are helpful and will be added.

[D5] Lines 20-21:

The authors mention only a case of growth of vertical girdle fabric here. At this stage of this paper, the authors should assume that readers do not know how special the cases of the vertical girdle fabric are. The authors need to specify type of strain, compression, extension, or shear. If the authors express c-axes orthogonal to the strain, it is response to the extensional strain or convergent ice flow.

[D6] Lines 22-23

This statement is wrong: vertical girdle fabric is typical only at ice divides and in convergent flow. They are rather limited zone in the ice sheet. It will not occur in divergent flow or simple laminar flow.

[D7] Line 23 "aforementioned scenario"

Despite these words, it was not mentioned before in this manuscript. It seems that this manuscript takes the girdle fabric as a basis. Please inform non-specialist readers of more general aspects, such as relations between type of strains and consequent preferred orientations of the c axes.

The short introductory summary addressed by these three comments was indeed very much constructed to fit the girdle fabric scenario as encountered in the ice instrumented by IceCube. To avoid generally incorrect statements we propose the following alternative introduction and would be grateful for feedback:

As a hexagonal crystal it will most readily deform as shear is applied orthogonal to the c-axis (crystal symmetry axis, normal to the hexagonal basal planes), leading to slip of the individual basal planes (McConnel, 1891). In polycrystalline ice the crystals effectively re-organize themselves to minimize the stored strain energy, resulting in non-isotropic / preferential c-axes distributions and a bulk anisotropic viscosity (Faria et al., 2014b). The effects of recrystallization are experimentally most commonly observed as a crystal orientation fabric through the use of polarized light microscopy on thin sections of ice core samples (Alley, 1988; Wilson et al., 2003). In this work we only consider scenarios where c-axes are distributed isotropically (uniform fabric), are aligned in a single direction (unimodal fabric) or lie in a plane (girdle fabric). The latter is of primary importance for the studied ice.

[D8] Lines 29-30, citation of Linow

Linow et al. (2012) discuss a very special case of firn, and not ice-depths below pore close-off. In ordinary ice cores using X-ray CT, we will not observe presence of grain boundaries, grain volume or grain elongation. Please rewrite and repair this part of description. Please do not mislead readers. In addition, X-ray CT is available even for whole cylinder of ice cores. Please look at Freitag 2013 paper indicated below for example. Thus, "these techniques further restrict the sampling volume" is not correct as a general statement.

- *Freitag, J., S. Kipfstuhl, and T. Laepple (2013), Core-scale radioscopic imaging: a new method reveals density–calcium link in Antarctic firn, J. Glaciol., 59(218), 1009 - 1014, doi:10.3189/2013JoG13J028.*

Thank you. We were not aware of the capability to CT-scan entire ice cores and acknowledge that the firn case is special. In response we propose to rewrite the sentence as follows:

Volumetric quantities such as grain volumes and shapes are generally not accessible through the commonly employed techniques, with exceptions like tomographic X-ray imaging in firn where

grains are commonly separated by air pockets (Linow et al., 2012).

> [D9] Lines 30-31:
> It is sudden that the authors mention thin slices here. It is nothing to do with previous sentences.

While we do not want to elaborate on the technique here, we propose to clarify the context by rewording to 'the microscopy of thin slices cut from ice cores'. This should make the connection to line 26 more evident.

> [D10] Around lines 32-33:
> I suggest that a method described in papers below is available for detection of ice fabric using thick volume of ice core using radio-wave birefringent nature of polycrystalline glacial ice. Because your context is on limitations of earlier methods, these seem necessary citations.
>
> - *Saruya, T., S. Fujita, and R. Inoue (2022), Dielectric anisotropy as indicator of crystal orientation fabric in Dome Fuji ice core: method and initial results, J. Glaciol., 68(267), 65-76, doi:10.1017/jog.2021.73.*
>
> - *Saruya, T., Fujita, S., Iizuka, Y., Miyamoto, A., Ohno, H., Hori, A., Shigeyama, W., Hirabayashi, M., and Goto-Azuma, K.: Development of crystal orientation fabric in the Dome Fuji ice core in East Antarctica: implications for the deformation regime in ice sheets, The Cryosphere, 16, 2985–3003, https://doi.org/10.5194/tc-16-2985-2022, 2022.*

The sentence in line 32-33 only serves to introduce remote sensing through reflected radiation. The radar technique is elaborated on in lines 35-39 with references given there in (Fujita et al., 2006; Matsuoka et al., 2003; Jordan et al., 2019; Young et al., 2020).

> [D11] Lines 33-34:
> I did not understand a relation between "can not only be imaged in ice cores" and the rest of this sentence. As for meaning, I did not find good link.

We propose the simplify the sentence to:

Ice fabric can not only be studied through the microscopy of ice cores. It also leads to a directionality in the propagation of sound and electromagnetic radiation.

> [D12] Line 41: Optical anisotropy
> Term is vague. Please specify as optical anisotropy of polycrystalline glacial ice, single crystal, or both.

At this point we are introducing a short hand term describing the observed effect which is used throughout the rest of the manuscript. To clarify this we propose to style it in italic. The observed effect is elaborated on in the following paragraph. At this point the underlying physical effect is not explored so adding an additional qualifier like 'polycrystalline' seems unwarranted here.

> [D13] Lines 42-46: "The effect was originally modelled as a direction-dependent modification to Mie scattering quantities, either through a modification of the scattering function as proposed by Chirkin (2013d) or through the introduction of a direction-dependent absorption as introduced by Rongen (2019). As also shown by Rongen (2019), both parameterizations lack a thorough theoretical justification and resulted in an incomplete description of the IceCube data."
>
> Please specify, at least, how these authors assumed sources of scattering or absorption. Otherwise, readers do not easily understand what kinds of studies they have done before unless they visit these papers and read closely. Also, it is hard for readers to explore someone's PhD thesis.

While Mie scattering already implies the interaction with impurities, we agree that explicitly mentioning impurities at this point is useful. We agree that citing PHD theses is best avoided if other publications are available. This is sadly not the case here. We propose the following updated sentence:

The effect was originally modelled as a direction-dependent modification to impurity induced Mie scattering quantities, either through a modification of the scattering function as proposed by Chirkin (2013d) or through the introduction of a direction-dependent absorption as introduced by Rongen (2019).

> [D14] Line 47: grain size
> Please specify size range, to provide not only concept but also range of quantity.

Good point. We propose to add: *expected to be on the millimeter scale*

> [D15] Line 48: grain boundary properties
> What kind of properties? Please specify to readers, to provide not only concept but also concrete physical basis.

We propose to replace *properties* with *spacings and orientations*.

> [D16] Lines 53-54:
> Please explain to readers about physical mechanisms responsible for the scenario of the diffusion. Things are explained rather conceptually around this part of the paper.

Yes, going over this section again an introduction is indeed missing. We propose to change the structure of the paragraph starting on line 47 as follows.:

First attempts to attribute the observed effect not to Mie scattering but to the ice intrinsic birefringence have been made by Chirkin and Rongen (2020). Here the optical anisotropy results from the cumulative diffusion that a beam of light experiences as it is refracted or reflected on many grain boundary crossings in a birefringent polycrystal with a preferential c-axis distribution. The wavelength of $\sim 400\,\text{nm}$ employed in the IceCube calibration studies is significantly smaller than the average grain size, which is expected to be on the millimeter scale. Thus, grain boundary spacings and orientations must be accounted for in addition to the fabric.

> [D17] Line 110
> Bubble – clathrate hydrate transition occurs as a thick zone ranging several hundred meters. Please let readers know it.

Agreed. We propose the following rewording of this sentence:

> The top 1450 m were left without instrumentation because of the strongly scattering ice that exists above. The depth where all bubbles have converted to air hydrates was determined to be ~1350 m by the predecessor experiment AMANDA (Ackermann et al., 2006).

> [D18] Lines 137 – 138
> When you express as "ice", please specify whether it is polycrystalline glacial ice or intrinsic nature of single crystal of ice. Otherwise, it will be one of sources of confusion for readers. When you discuss nature of absorption or scattering, we need to know, if nature under discussion is polycrystalline ice with grain boundary network or not. Also, we need to know if nature is for ice that include various kinds of impurities/inclusions or not. Please be careful on this point throughout this manuscript.
> Specifically at lines 137-138, please clarify if Warren and Brandt assessed nature of ice sheet ice or not.

To aid distinction we propose to add the qualifier *pure*, contrasting with the second half of the sentence ('*..., the light propagation is dominated by Mie scattering on impurities.*'). Whether the considered ice is a mono-crystal or polycrystalline is inconsequential at this point as the grain boundaries themself do not absorb light and the microscropic distribution of impurities does not impact the macroscopic absorption length (see response to G3).

Given the second half of the sentence ('*..., the light propagation is dominated by Mie scattering on impurities.*') and the following discussion if impurity types, we think the manuscript concise in describing that the primary drivers of ice optical properties are impurities.

Warren and Brandt review the complex refractive index of impurity free ice, but in the optical regime where lab-grown ice to date under-performs compared to deep glacial ice, have to resort to in-situ measurements carried out by the IceCube predecessor experiment AMANDA (Ackermann 2006). This additional information is evident when following the reference and we think it unnecessary to include at this point in the manuscript.

[D19] Lines 139-141, "The impurity constituents are believed by He and Price (1998) to be dominated by mineral dust, marine salt and acid droplets as well as (volcanic) soot."

I understand that He and Price (1998) paper summarized possible materials that can interact with light, with their knowledge in 1998. However, there are advancement of science in cryospheric sciences.

I suggest that the authors to consider providing updated knowledge, rather than drawing attentions of readers to old belief of 24 years ago.

Just as an example, I provide one of possible statements.

The impurity constituents are dominated by insoluble mineral dust, salt components, liquid phase acids, soot and volcanic glass (e.g., Arienzo et al., 2017; Barnes et al., 2003; Narcisi et al., 2005; Sakurai et al., 2011; Stoll et al., 2021).

Points: Belief by He and Price seems old to cite here now in 2022. Chemical reactions related to salts are much more understood nowadays. Various chemical reactions occur in the atmosphere during transport of aerosols and in snow and firn to general salts and acids in ice. In addition, soot is not related to volcanic eruptions (though there may be exceptions). Droplet does not seem proper wording. Acids sometimes exist at grain boundaries as liquid depending on components, temperature, and chemical reactions. Particles that come from volcano is glass shards.

Here, possible citations are as follows. There are much more choices.

- *Arienzo, M. M., McConnell, J. R., Murphy, L. N., Chellman, N., Das, S., Kipfstuhl, S., and Mulvaney, R. (2017), Holocene black carbon in Antarctica paralleled Southern Hemisphere climate, J. Geophys. Res. Atmos., 122, 6713– 6728, doi:10.1002/2017JD026599.*

- *Barnes, P. R. F., E. W. Wolff, H. M. Mader, R. Udisti, E. Castellano, and R. Röthlisberger (2003), Evolution of chemical peak shapes in the Dome C, Antarctica, ice core, Journal of Geophysical Research-Atmospheres, 108(D3), doi:412610.1029/2002jd002538.*

- *Narcisi, B., J. R. Petit, B. Delmonte, I. Basile-Doelsch, and V. Maggi (2005), Characteristics and sources of tephra layers in the EPICA-Dome C ice record (East Antarctica): Implications for past atmospheric circulation and ice core stratigraphic correlations, Earth and Planetary Science Letters, 239(3-4), 253-265, doi:10.1016/j.epsl.2005.09.005.*

- *Sakurai, T., Ohno, H., Horikawa, S., Iizuka, Y., Uchida, T., Hirakawa, K., & Hondoh, T. (2011). The chemical forms of water-soluble microparticles preserved in the Antarctic ice sheet during Termination I. Journal of Glaciology, 57(206), 1027-1032. doi:10.3189/002214311798843403*

- *Stoll, N., J. Eichler, M. Hörhold, W. Shigeyama, and I. Weikusat (2021), A Review of the Microstructural Location of Impurities in Polar Ice and Their Impacts on Deformation, 8, doi:10.3389/feart.2020.615613.*

We very much agree that the knowledge on which He and Price (1998) is based is outdated, in particular with regard to acids. Yet, to our knowledge He and Price (1998) is still the only paper which attempts to calculate bulk absorption and scattering coefficients from microscopic properties (refractive indices, size distributions, number distributions, etc.). While avoiding more details on the individual impurity constituents, we propose to correct the sentence as follows:

The primary impurity constituents contributing to absorption and scattering were identified by He and Price (1998) to be mineral dust, marine salt and acid inclusions as well as (volcanic) soot.

The Stoll et al. (2021) reference has been picked up in response to G3.

[D19 continued] In addition, as for clathrate hydrate crystals, a paper below seems informative for examination of light wave propagation, even if the authors evaluate possible effects are negligibly small. In the present paper, you are discussing weak changes in refractive index. There is huge amount of clathrate hydrate crystals in the ice sheet. Thus, readers need to know this presence is properly assessed by the authors.

- *Uchida, T., A. Miyamoto, A. Shin'yama, and T. Hondoh (2011), Crystal growth of air hydrates over 720 ka in Dome Fuji (Antarctica) ice cores: microscopic observations of morphological changes below 2000 m depth, Journal of Glaciology, 57(206), 1017-1026, doi:10.3189/002214311798843296.*

While not relevant to the overall scattering coefficient or the ice optical anisotropy, the topic of air-hydrates seems to naturally come to mind here. So we agree that it should quickly be discussed. We propose to add the following paragraph following line 146:

While not contributing to absorption, air-hydrates also contribute to scattering. While their number density is large, the small difference in refractive index (Uchida, 1995) and their large size (Uchida, 2011), compared to the typical wavelengths considered, result in isotropic scattering and they contribute at most a few percent to the overall scattering coefficient (He and Price, 1998). Thus, scattering on air hydrates was thus far not modeled separately and the effect was simply absorbed in the overall scattering coefficients. Diffusion through scattering on grain boundaries was also already quantitatively estimated by He and Price (1998) to contribute about as much as air-hydrates to the overall scattering coefficient. At the time the average deflection process described in this work was not known and thus its large importance not realized. The quantitative contribution of diffusion in the polycrystal to the overall scattering coefficient as derived in this work is given in section 7.

[D20] Line 145:
I suggest that "climatological conditions" can be "climatological conditions such as dusts and aerosols in the atmosphere in the past" to be more concrete.

We are happy to adopt this suggestion.

[D21] Lines 170-171:
The authors wrote as "The limited volume of the ice cores does thus not allow for a direct measurement of optical properties, even though they are able to inform on the impurity constituents and their size distributions."
It seems a vague and subjective statement. There should be many methods to directly measure "optical properties" of ice. You state generally as optical properties; it seems impossible to provide a statement like this. You mention propagation through distance of 100-400m. If we can prepare proper experimental setting, we may be able to detect it.
Later at lines 290, you showed directional dependence of the signal was double for propagation of 125 m. If assume that directional dependence of the signal was 3dB / 100m as approximation, for diameter of an ice core (0.1m), it is 0.0003dB/0.1m. It is far better if the authors provide size of numbers that is necessary for a scale of ice core measurements.

The distance scales involved here do not make this a viable measurement to be conduced on ice core samples. Consider for example a full 1 m ice core segment with a representative absorption length of 200 m. Over this distance only $1 - \exp(-1/200) = 0.5\%$ of the light is absorbed, which is very

challenging to resolve, taking into account experimental uncertainties (primarily coupling of the light source and sensor to the ice core). To our knowledge no such measurement has ever (successfully) been performed. This is supported by the comparatively far weaker limits set by lab measurements as given in the review by Warren 2008 and his decision to base the stated value for the optical absorptivity on in-situ measurements by AMANDA (Ackermann 2006). Things get even worse when trying to measure a directionality. Over the assumed $10\,\text{cm}$ distance and following the linear extrapolation as suggested by the reviewer a maximum intensity difference of 0.16% is expected.

> [D22] Line 184:
> Please clarify meaning of "ice realizations" to the readers of TC. I did not understand what was meant. My concern is that the same problem happens to many readers of TC.

This was also noted by the second reviewer and we would like to follow his suggestion and instead talk of *'hypothesized optical properties and ice-crystal orientations'*.

> [D23] Line 187:
> $c_{ice}$ is not defined anywhere. I imagine it is speed of light in ice. I wonder if it is an expression commonly used in physics.

It is indeed the speed of light in ice and a notation commonly employed in physics. Yet, this equation was alread cut as part of the response to G6.

> [D24] Lines 187-189:
> I did not find definition of scattering coefficient in the equation. Did you simply rephrase "diffusion coefficient" as "scattering coefficient"? If it is so, please make it clear to readers.

This part of the manuscript has been shortened in response to G6 and the equation and this particular sentence no longer appears.

In general we would like to note that we tried to consistently denote only the light diffusion caused by impurities (through Mie-scattering) as scattering.

> [D25] Lines 190-191:
> I was confused at multiple points. Please let us understand why this is inaccurate. Please let us know why an assumption of clear and layered ice causes problem? What do you mean with a word "layered"? Do you mean layers caused by deposition layering? Alternatively, do you assume presence of layered propagation paths?
> From here, please note that my understanding after section 3.2 was bad, even after reading the paper many times. I ask the editor to find a reviewer who can fairly evaluate these sections.

This part of the manuscript was streamlined as part of the response to G6, where you please find the newly proposed text.
Regarding the stated questions:

- Equation 5 is only correct if the distance between the emitter and the receiver is at least multiples of the diffusion (can be scattering) length. This is simply not the case with sensors spaced $125\,\text{m}$ apart laterally and within the deep ice instrumented.

- This indeed refers to chronologically layered impurity depositions, as discussed in line 147. On this topic please also see the response to the third general comment by the other reviewer.

- Layered propagation paths are not assumed. The propagation paths are simulated from first-principles by PPC and the resulting paths are not further altered.

> [D26] Section 3.2 in general
>     I did not understand how your Photon Propagation Code (PPC) was used. Please let readers know how you assumed many physical properties of ice and inclusions, in your PPC calculations.

While the optical properties described in section 2.3 (with more details in the references cited therein) fully describe the light diffusion (without anisotropy), the equations themselves (even with known optical properties) do not allow to generate predictions that can be compared to data. PPC as described in section 3.2, probabilistically applies the equations put forward in section 2.3 to describe the propagation paths of adequately initialized photons (for example coming from LEDs). The resulting detection events at the deployed sensors with travel times from the point of emission make up the simulation data to be compared to the experimental data. No further assumptions than those made in section 2.3 enter the simulation.

In response to G6 the beginning of section 3.2 has changed significantly (please see there) and we hope the relevance is clearer now.

> [D27] Sections 3.2 and 3.3 in general
>     It was hard for me to understand this part of the paper. Possibly, some scientists can understand these sections without difficulties.

> [D28] Figure 5:
>     I did not understand the authors' purpose of showing this figure. I wonder why mixtures of data from various origins were given here.

While not directly related to the ice optical anisotropy effect the stratigraphy of absorption and scattering length provides important contextual information, both regarding the clarity of the instrumented ice and the methology of analyzing discrete 10 m layers which are the same here as in sections 6-8.
Ice properties can only be directly calibrated for depth contained within the detector. For context and to describe the light propagation of particle physics events on the edges of the instrumented ice, extrapolations employing input from other sources are necessary.

> [D29] Sections 3.4
>     I did not understand the model "South Pole Ice Model". It seems again vague words. What kind of model to analyze what? What are the parameters? If you call it as Ice model, again it seems vague as a term. Can it be ice flow model, light wave propagation model in ice, or ice sheet model related to absorption and scattering? Do you mean profiles in Figure 5 as layering model?

The introductory two sentences to 3.4 (*Employing the experimental and analysis methods described above, absolute absorption and scattering coefficients and their wavelength scaling have been measured for all instrumented depths as described in detail by Ackermann et al. (2006) and Aartsen et al. (2013d). The resulting model, called the "South Pole Ice Model" (SPICE),..*) define what the term "South Pole Ice Model" refers to. It is the name given to the combination of parameters required to

describe the optical properties of the ice instrumented by IceCube.

The name has been long established (with uses in publications such as Aartsen (2013d) and Chirkin (2013d)) and we would like to keep using it for this manuscript.

> [D30] Line 274:
> I did not understand meaning of "footprint of IceCube". Is it related to footprint of the radar beam pattern in IceCube experiment area? Is it area and depth ranges where you covered by IceCube experiment? If true, why was a term footprint used?

To us, the term footprint refers to the surface projection of the detector geometry. So the area at the surface under which the detector is located. To avoid confusion, we propose to simply replace *footprint* with *surface extent*. (The same applies to line 270.)

> [D31] Line 276 "laminar flow as described by Aartsen et al. (2013a)"
> Are you describing flow regime of the ice sheet? Please state more in detail to make it understandable to readers. Also, the vertical girdle fabric should develop under conditions of convergent ice flow. If it is simple laminar flow, presence of the vertical girdle fabric should not be explained. Please provide a brief statement as to how this ice fabric developed within the ice sheet with laminar flow dominated by simple shear strains. Simple shear will give single pole fabric.

The term was in this instance simply meant to imply that the chronological layering is assumed to be intact (so no layer folding, turbulent flow, etc) at all locations in the detector. We will abandon the term 'laminar flow' here and propose the following updated sentence:

> Instead, the depth offsets of characteristic features as observed in the dust logger data from seven different IceCube holes has been used to interpolate the depth-dependent layer undulations assuming an undisturbed chronological layering as described by Aartsen et al. (2013a).

The exact strain state of the instrumented ice is not known to us. We simply follow the observation of a very strong girdle fabric as reported by SPC-14.

> [D32] Line 281:
> Please word "ice model" (again vague) as ice sheet flow model or something like this to make it understandable. By choices of terms, I was often confused.

The abbreviated term *ice model* has at this point already been used six times in the manuscript and we would assume familiarity. To avoid confusion regarding the term we have elaborated on the definition of the term in the introduction. For the proposed text please see G6.

> [D33] Lines 284 and 290:
> "Ice optical anisotropy" does not seem proper term in physics because main topic in this paper is for the polycrystalline ice within the ice sheet. If wording is ice optical anisotropy, it is vague; not a few readers will first think about optical properties of single crystal ice. Wording something like "anisotropy in optical properties within the ice sheet" or "optical anisotropy in polycrystalline glacial ice " seem better. Please consider.

At this point we are defining a short-hand term describing the observed effect which is used throughout the rest of the manuscript. At this point the underlying physical effect is not explored so adding an additional qualifier like 'polycrystalline' seems unwarranted here.

> [D34] Lines 305-311:
> I did not understand statements related to elevation angle. My concern is that not a few readers will experience the same. Similarly, I did not understand a situation why this is a parameter which is hard to accurately obtain from ice core data. Please make it understandable.

We propose to change the sentence to the following:

As this is a parameter that is difficult to obtain from ice cores as their in-situ orientation is often not retained, it may be further investigated in the future.

> [D35] Figure 8:
> I was again confused with reasons below.
> Photon counting perpendicular flow axis is larger than that along flow axis. It does not seem in agreement with Figure 6. On the left panel, a peak for the scatter case is at a timing smaller than a peak for the absorption case. I do not understand this timing difference.

This is a very good point. The absolute photon count is not only a function of the ice properties but also of the emitter-receivers pairs in that direction. Comparisons should always be made with regard to the '*flasher data*' curve. We propose to add the following sentence to the manuscript:

As more emitter-receiver pairs are included in the perpendicular case compared to the case along the ice flow, the total photon counts are not directly comparable between the two plots, and should instead be compared to the curve titled "flasher data" within each plot.

The timing behavior of the scattering model is elaborated on in section 4.3. (Line 344: *As scattering is reduced in the flow direction light arrives earlier on average.*)

> [D36] Lines 329-330:
> I felt that the context became unreliable to read two lines here. If the impurity particles are aligned due to stress/strain conditions, it should have been observed by ice core scientists. Is it along grain boundaries, along triple junctions, along dislocations or along crystal lattice? What kind of particles do you assume? Rather, how about alignment of triple junctions of grains along the normal axis of the vertical girdle plane (axis of tensile strain and grain elongation)?

We agree that the reasoning for the physical plausibility of the early scattering-based model is questionable. The types of particles are the four impurity types (mineral dust, soot, salt, acids) as described earlier. No such detailed considerations as to the cause of the orientation as suggested by the reviewer were to our knowledge considered at the time. We can here only report on the previous model as proposed at the time before suggesting what we believe to be a better-motivated model. The following sentence also mentions that such preferred impurity orientations have to our knowledge not been seen by ice core scientists.

> [D37] Lines 357-386:
> There is no subsection title only in this part of the manuscript. Please define what you would like to let readers know by providing proper subsection title.

We propose to add the following subsection title: *The electromagnetics of uniaxial, birefringent crystals*

> [D38] Line 455:
> I was not able to find Woodcock parameters in several reference papers given here. Please clarify.

The information is contained in the data-release referenced by Voigt (2017) in line 460 (while the references in line 455 and 458 only describe the core retrieval itself). The material to be added in response to D39 should suffice to inform the reader. We will include a reference to the appendix here.

> [D39] Line 460:
> I visited data set of Voigt (2017). Data are stored separately folder by folder; it was hard to grasp wide view in terms of depth dependence. The same situation will occur for many readers. For a better understanding, it would be nice that you prepare an appendix in which readers can browse crystal fabric pattern. It would be even more nice that Woodcock parameters are given together. Just examples from several depths will help.

This is a valuable suggestion. We will add such plots as part of Appendix A.

> [D40] A paragraph from line 462 to 475:
> We can observe ice fabric, grain shape and size in Alley et al, 2021. Readers will wonder if your assumption of grain shapes agrees with reality. It is something difficult to evaluate only by reading this manuscript.

Alley et al. in their current presentation of measurements do not include any information to help gauge if the average crystal shape is close to a prolate spheroid or whether a full-triaxial ellipsoid is required for an accurate representation.
Regarding the overall size and elongation a comparison to the SPC-14 measurement only makes sense after the values have been fitted and takes place in section 7.

> [D41] Line 477:
> When you denote orientation as "tilt direction", glacier researchers will be more familiar to a term "transverse direction".

As of now, the term tilt direction is introduced in line 290 when discussing the experimental signature. This is indeed an oversight seeing that the tilt itself is introduced in an earlier section. We will make sure to properly introduce the term in section 3.4.1 and emphasize that it is transverse to the flow.

> [D42] Figure 11:
> Please explain in more detail what asymmetry of the distribution in the second and the 3rd from the left figures mean.

We propose the following updated caption:

> Example diffusion patterns after photon propagation through 1000 crystals (roughly equivalent to 1 m) with a perfect girdle distribution of c-axis orientations. The initially emitted photon direction is perpendicularly out of the picture, with an opening angle to the flow as indicated. The figures histogram the final direction vectors of many photons. The change in diffusion (width of the distributions) as well as the subtle effect of photon scattering towards the ice flow (towards the right) can be seen.

 Figure 11 and Figure 12 right panel:

> I believe that the special case "90 degrees to flow" will attenuate to zero at the end because of scatter at randomly oriented grain boundaries. Perhaps readers should know it if my understanding is correct.

Yes. This case is unstable to small perturbations. Once any ray randomly departs from this direction, deflection kicks in and directs the light towards the flow direction. In realistic ice, scattering on impurities will far outweigh the diffusion resulting from scattering on the grain boundaries. This expedites the effect but also ensures that some intensity always remains on the tilt axis.

We suggest adding the following discussion of relative diffusion strength and intensity on the tilt axis to section 7 (Resulting ice model) (as also requested by the other reviewer):

> The overall grain size of ∼1mm and the increase in size at larger depths, where ice crystals are generally larger, are as generally expected and measured in glaciology (Laurent et al., 2004; Alley et. al., 2021). In addition an anti-correlation between crystal size and impurity concentrations, as mapped by optical properties can be observed. This follows the expectation that impurity related processes such as impurity drag hinder grain growth (Durand, 2006). [...] Averaged over all instrumented depths light diffusion in the birefringent ice polycrystal amounts to an effective scattering coefficient of $2.47 \cdot 10^{-2}\,\mathrm{m}^{-1}$, accounting for on average $\sim 8.5\%$ of the total scattering present in the ice. The comparatively strong isotropizing effect of Mie scattering also explains why the intensity on the tilt axis is never fully depleted.

[D44] Figure 13:

> It would be useful for readers if you add materials as below.
> (i) Elongation 1.0 case, that is, no grain shape effects and only fabric effects
> (ii) In (S2/S3) 0.0 case, that is no fabric effects and only grain shape effects

Below please find the plot with the requested additions. (For the $\ln(S2/S3) = 0$ column $\ln(S1/S2)$ is also zero.) As you'll find the already provided cases $\ln(S2/S3) = 0.1$ and *Elongation 1.11* are close enough to fully isotropic and spherical to essentially be indistinguishable from the added cases. As all other panels assume $\ln(S1/S2) = 0.1$, we would prefer to stay with the current presentation.

[Figure]

deflection for prolate spheroids, ln(S1/S2)=0.1
(after 1000 grains, including chord correction)

[D45] Figure 13:
    Please note to readers that the scales of the ordinate are different depending on rows.

We will add such a statement to the caption. Following this comment, we also pondered if equal ordinate scales for all rows would be helpful, but this would result in a loss of detail for the smaller elongations.

[D46] Lines 583-584:
    Elongation fixed to 1.4 does not seem the same as elongation given in Alley et al. (2021). In their slide at page 10, the maximum value is 1.24.

Alley et al. (2021) are restricted to a maximum depth of 1700 m. Between 1379 m and 1700 m the maximum elongation steadily increases from around 1.14 to around 1.24 with the gridle fabric getting narrower at the same time (slide 13). As the fitted elongation describes the average ice properties between 1500 m and 2500 m a slighltly larger value would be expected following this trend.
To more explicitly reflect this subjective extrapolation, we propose to change the wording to:

Thus, the elongation was fixed to 1.4, which is a good fit at all layers and seems to be a reasonable value given the largest value measured in the deepest parts of SPC14 ($\sim 1.24$) and the observed trend of increasing elongations up to that depth (Alley et al., 2021).

> [D47] Lines 596: absorption anisotropy by a factor of 2.45
> I find no reason to support or not because there are many items of unknown.

We absolutely agree. As stated in the Outlook we do not currently think the introduced absorption anisotropy physical and thus no emphasis should be put on the quantitative result.
Stating the number as resulting from the fit still seems worthwhile for future discussions of possible resolutions. To emphasise this point we actually suggest to update lines 616 as follows:

Due to the need to include an absorption anisotropy **in order to arrive at reasonable grain sizes**, for which no first-principle explanation is known, there appear to remain additional physical effects not fully accounted for in the first-principles model. At this point it remains unclear whether the anisotropic Mie absorption is real or if it is an artifact from incomplete modeling of birefringence effects.

> [D48] Line 601:
> Cleaner seems a strange word. Even in case you intended to mean "more transparent", it is not the case in the ice sheet. Degree of transparency depend on inclusions.

At our particular location the ice below the dust layer ($\sim$1500m) is on average more transparent compared to the ice above (see Figure 5), which is indeed believed to be a result of fewer inclusions. To be more explicit and to expand on the discussion of the depth dependence as also requested by the over reviewer we propose to add a paragraph as detailed in response to D43.

> [D49] Lines 601-602:
> In two citations, both indicate much larger grain size. Thus, quantitative much seem questionable.

Alley et al., 2021, being a measurement of the SPC-14 ice core from the same location as Icecube, is the primarily relevant reference here. Their mean equivalent diameter (center line of the colored bar) in the overlapping depth region (around $1600\,\mathrm{m}$) gives a grain size of $\sim 1.75\,\mathrm{mm}$ compared to the $\sim 0.75\,\mathrm{mm}$ as shown in Figure 15. Both measurements have systematic biases. The inclusion of the yet unmotivated absorption anisotropy and the ambiguity between elongation and size in our fit are examples already discussed in the manuscript. The SPC-14 measurement is biased by only looking at sample planes aligned with the flow direction in this depth range, which is likely to give elevated results compared to the true volumetric mean as elongation increases.

Thus we think the general statement justified. Since this paragraph has already been modified in response to D43, please check there for the newly proposed wording.

---

## Author Comment (AC2)

**Reply to review report by David Lilien submitted on the 9th of January 2023**

The IceCube Collaboration

March 7, 2023

Review of "In-situ estimation of ice crystal properties at the South Pole using LED calibration data from the IceCube Neutrino Observatory" by Abbasi et al.
This paper presents observations of an anisotropic effect in the calibration data from the IceCube detector near South Pole. Photons are deflected from paths that would be expected from scattering and absorption alone. Previous work has shown that this effect can only be approximated poorly using anisotropic absorption and scattering, and these effects are not on firm physical grounds anyway. In the present work, the authors consider the effect of birefringence. They essentially run simulations of the propagation of light (both ordinary and extraordinary waves) through 1000 crystals, and look at the effect of refraction and reflection as a function of the crystal shape and orientation. From this, they parameterize a relatively simple function of how the birefringence affects propagation, and include that function in the original model of absorption and scattering. They find a much better fit to observations, although this fit is still improved with non-physical anisotropic scattering/absorption effects. In addition to describing this work, there is a lot of history and background of IceCube and of other attempts to model the observations.

I must admit that it has been a long time since I had to deal with derivations directly from Maxwell's equations such as those presented in the paper. While I was fully able to follow the arguments and derivation, I do not think I would have been able to spot an error; hopefully other reviewer(s) have that knowledge. For the portions that I can evaluate, I think the work is nearly publishable (with the exception of the second general comment), though the presentation could use significant improvement to be really digestible. I do not object to long papers, but only when it is justified by the content; here I think the paper needs to be shortened so that the point is not lost in all the other material (see first general comment). I think this will be a very nice contribution when these issues are addressed—the observations are fascinating, and I think the explanation is compelling. While there may not be wide applications, this paper describes very basic information about the properties of ice, and thus deserves to be published.

We would like to sincerely thank the reviewer for the significant time investment in reviewing the manuscript and the constructive and encouraging feedback.

**1 General comments:**

> The paper is very well written at the sentence level, but the larger structure leaves gaps at some points and provides overly much detail at others. I think a large part of the issue could be alleviated with a more normal paper structure describing a problem and the work to address it rather than the meandering path of the last 10 years (more on this in the next paragraph).

The paper is certainly not intended as a historical review and a number of proposed cuts to address this appearance will be outlined below. Yet given we rarely publish in glaciological journals (the last instance being Ackermann2006) and that a large fraction of the target audience is likely neither familiar with ice optical properties nor the IceCube detector, we find a short summary of all aspects required to perform this kind of measurement is pertinent. Otherwise the reader is left to explore a large body of work in an unfamiliar field before being able to study the novel aspects of this manuscript.

The different gaps in explanations, alternative aspects to be added and areas of excessive details identified by the two reviewers highlight that arriving at a fully concise manuscript is futile given the diametral demands, but we hope to strike a balance here.

> One example of a gap is how this work fits into the context of known birefringent effects in ice (I'm thinking of birefringence at frequencies used for radar); this is addressed in Section 5.3, but that is an odd place for the reader to get the context—it would be much more at home in the introduction.

We struggle to move the entire section, as it does require knowledge about the diffusion mechanism and its relevant parameters, which is only provided in the previous section. We thus propose to leave the section as is and hope that the changes proposed to the introductory section (see two comments from here) are sufficient to give the added context.

> As another example, Section 6 jumps in with no intro—I assume that it is there because it is computationally necessary not to explicitly model the birefringence effects in individual crystals during the simulations, but it would help if that were stated clearly.

Good catch. The reason is in fact as assumed. Using the exact electromagnetics simulation (section 5.2) directly would slow down the overall photon propagation simulation by orders of magnitude, while the evaluation of the parametrized model only adds an insignificant burden. We will add as much.

> In terms of excess details, this is an extremely long manuscript; it can take a really long time to get to the explanation of how the pieces fit together, by which point the reader is already lost. For example, section (3) describing what is essentially an isotropic model of the optical properties of the detector, is not really motivated and so comes as a distraction/reads as history until much later when I understood how the parameterization from the anisotropic model was then used in the isotropic one.

We propose the following cuts:

- Equations 2, 3, 4 and 5 shall be cut along with the accompanying text and replaced with more conceptual descriptions.

- Figures 2 and 16 shall be cut without replacement.

- Lines 234 to 240 shall be cut without replacement.

- Section 4.3 (Early empirical modeling) shall be replaced, removing most details regarding the mathematical modeling. The suggested new paragraph can be found in response to a detailed comment regarding this section.

In addition, we propose to expand lines 67-74 (describing the structure of the manuscripts) as follows:

This manuscript has the following structure: Section 2 introduces the IceCube Neutrino Observatory (Sections 2.1 and 2.2) and how it employs ice as a detection medium (Section 2.3). Section 3 describes the properties of the LED calibration data used in this study (Section 3.1), explains the photon propagation software used to generate simulated data (Section 3.2) and details the likelihood analysis comparing simulated to experimental data in order to infer ice properties (Section 3.3). Section 3.4 briefly reviews the state of the isotropic, layered model used to describe the ice optical properties prior to this work. The experimental signature of the ice optical anisotropy (Section 4.1) as well as early modeling attempts (Section 4.3) are summarized in section 4. The newly developed model to account for the ice optical anisotropy based on the ice-intrinsic birefringence is described starting with Section 5. Sections 5.1 and 5.2 explain the electromagnetic theory governing the birefringence in polycrystals, while Section 5.3 introduces a software package to simulate the resulting diffusion patterns. Section 5.4 compares the experimental signatures and conceptual understanding of the underlying optics to birefringence observations in radar sounding, a field most readers are probably more familiar with. Section 6 explains how the diffusion patterns are applied in the IceCube photon propagation simulation (Sections 6.1 and 6.2) and how crystal properties have been inferred (Section 6.3). Section 7 describes the resulting ice optical model. Section 8 discusses shortcomings of the model as well as future measurements in upcoming IceCube extensions and through drill-hole logging.

While this adds some additional text, we hope that this additional context allows the reader to judge which parts of the manuscript are relevant for their particular interests/expertise.

> Some of the figures, which appear to be reproduced from elsewhere and are not particularly necessary for the present work, could be cut to streamline things; in my view, Figures 2, 3, 4, and 16 should all be cut.

We are happy to cut Figures 2 and 16. Figure 4, we think necessary to explain the nature of the used data (arrival time distributions of individual photons) and the matching simulation. Figure 3, while not vital, gives a sense of scale and completeness/complexity of the available data which we think is helpful to the unfamiliar reader. With the detector fully embedded in the ice and with each of the 60'000+ LEDs being observed by hundreds of sensors each volume of ice is tested repeatedly and from a multitude of illumination directions.

> At a number of places, avenues that were pursued but proved fruitless are described in great detail—I would suggest cutting these down for readability.

As detailed during the specific comments we propose to significantly shorten section 4.3 (Early empirical modeling).

> The paper would benefit greatly from a more clear description of the observations that imply anisotropy early in the paper (like we get along with figure 8), and a description of what the goal of the paper is. The introduction perhaps attempts to do this, but it reads more like a history of IceCube optical modeling than a problem statement–perhaps this was the goal, but it appears that both the other reviewer and I found this to be challenging to read as-is. Much of the second half of the intro in some way gives what the structure of the paper will be (e.g. describing the isotropic model fitting, etc.), but it was phrased in such a way that it was unclear that these would be expanded upon and would be critical components of the present work. In my view, the abstract does an excellent job of presenting what the paper is about, but the structure of the rest of the paper then does not follow the outline laid out in the abstract. For example, it would be very helpful for the reader to know that you are going to develop a grain-resolving anisotropic optical model, from that parameterize a diffusion function, and then input that into the previous ice model.

This is a valuable comment and we suggest the following modifications to the Introduction section:

- A reference to Figure 6 shall be added to line 42 and a reference to Figure 8 shall be added to line 46.

- We propose to change the structure of the paragraph starting on line 47 as follows:

  First attempts to attribute the observed effect not to Mie scattering but to the ice intrinsic birefringence have been made by Chirkin and Rongen (2020). Here the optical anisotropy results from the cumulative diffusion that a beam of light experiences as it is refracted or reflected on many grain boundary crossings in a birefringent polycrystal with a preferential c-axis distribution. The wavelength of $\sim 400\,\mathrm{nm}$ employed in the IceCube calibration studies is significantly smaller than the average grain size, which is expected to be on the millimeter scale. Thus, grain boundary spacings and orientations must be accounted for in addition to the fabric, making the effect challenging to derive from first principles.

- To clarify on the goal of the paper the following sentence (inspired by the wording of the reviewer) shall be added in line 58.:

  Due to computational limitations, a grain-resolving anisotropic optical model is parametrized using diffusion functions. These function in turn are applied as an extension to the existing, homogenious ice optical simulation. The new simulations, assuming different ice crystal realizations, are then compared to LED flasher data, which allows to partially constrain the crystal fabric, size and elongation.

- As detailed earlier we also propose to expand the second half of the intro, so that the reader is clear about the interplay of the sections and which sections may be most relevant to their interests.

> 2. Overall, I think the consideration of possible fabrics is a bit simplistic — this is fine considering the computational expense, but as of now some statements are simply incorrect. For example in Appendix A, the authors write "Woodcock (1977) realized that all possible fabric states can be visualized in a 2D plot," which is not at all true—only a rotated form of the second order fabric can be visualized in this way. In addition to correcting mistakes like this, we need some consideration of the implications. The question that I am particularly interested in is whether the higher-order moments of the fabric have any effect on the process of interest here; in radar, only the second-order moments control the effects, but I am not sure that the arguments carry over. For example, do circumpolar hoop fabrics and other complicated fabrics which are observed in glacial ice (see, e.g. the Faria paper cited in the text for examples), behave identically to single maxima in terms of this birefringence, or do they produce some other effect? If the latter or if it cannot be determined, then the text should make clear that what was considered was only a subset of the fabrics that have a give pair of Woodcock parameters.

Aside from computational limitations, the simplifications were primarily chosen to coincide with the fabrics to be expected at our study location. The most simple starting point and ice realization at shallow depths is a uniform fabric, below $\sim$1200 m ice at the geographic South Pole features a very strong and clean girdle (https://www.usap-dc.org/view/dataset/601057) and in the deepest ice one would expect this to turn unimodal. These three scenarios can nicely be interpolated between using the plane spanned by the two Woodcock parameters.

We will make sure to include the reasoning above in the introduction and section 5.2. Plain errors such as in the appendix will of course also be fixed (as detailed in the specific comments below).

Regarding more complex fabrics, such as circumpolar hoops, we do share the reviewers suspicion that they will affect the diffusion patterns in yet unexpected/unexplored ways. But we have so-far not simulated such cases. (Partially because that would require a new sampling scheme similar to the one presented in appendix A.) It is also worth pointing out that we found the fabric to be a subdominant contribution to the diffusion pattern, with the elongation being of primary importance. There is of course a physical link between fabric and crystal shape, but without a quantitative model linking them, we are currently left to treating them as independent parameters.

> 3. We need a better description of how the layering fits into the modeling. I think my confusion stems from a difference in how I think of layers (generally packets of ice deposited at the same time or radar reflections depending on context) and how this paper uses layers (as best I can tell, these are packets at specific depths, but I am unclear how they vary spatially and what the "IceCube coordinate system is"). This ambiguity clouds the results to a certain extent—for example, is a girdle in this coordinate system truly a vertical girdle, or is it tilted by the layer slope? While I list this as a major comment, it is only because I think it is important to address, not because it requires a lot of work—just a clear description of layers at line 148 when then first come up would satisfy me (section 3.4.1 comes late). In addition, this paragraph should make clear the extent to which the tilt is included in the model compared to being a source of error.

In this work "layers" also refers to "packets of ice deposited at the same time" and exhibiting equal optical properties. It's just that due to the resolution limitations we only consider ice properties averaged over ten meters, instead of studying for example annual layers. This is similar to the depth resolution in radar measurements. These layers are not assumed to be at the same absolute depth everywhere in the detector. Instead, their absolute depth is given at a reference location, while the absolute depth at any other location requires knowledge about the tilt / layer undulations as detailed in section 3.4.1.

To avoid confusion we propose to change the description at line 148 as follows:

> The detailed stratigraphy associated with the yearly layering cannot be constrained through IceCube data, nor is it needed in order to accurately describe the photon propagation over large distances exceeding tens of meters. Instead, average properties in 10 m depth increments, here called "ice layers", are being considered. The absolute depths of these layers as for example shown in Figure 5 are referenced to a location in the center of the lateral footprint of the detector. At any other location in the detector the same layer are found at slightly different depths following the layer undulations as will be described in section 3.4.1.

We acknowledge that the use of a technical/ slang term such as "IceCube coordinate system" is unfavorable in a paper. The term appears twice, once when converting absolute depth readings in SpiceCore to the IceCube stratigraphy and once when denoting the ice flow / anisotropy direction and primarily serves as internal reference. We propose to leave the appearance in line 302 as is, as an explanation is already present there, and reword the appearance in line 460 to:

> It reached a final depth of 1751m (Winski et al., 2019)), which corresponds to a depth of 1820m in the IceCube ice model (see Figure 5) accounting for the layer undulation between the two reference points.

Regarding the orientation of the girdle, it is worth noting that the diffusion patterns are simulated separate of the photon propagation framework and only their parametrizations are applied during photon propagation. This allows to orient the girdle arbitrarily and independently of the layer undulations. Here the parametrizations are evaluated such that the girdle normal vector is assumed to be perfectly horizontal, with no assumed correlation to the layer undulations.

**2 Specific comments and technical corrections:**

> 20-21: The sentence ending on line 20 and the one beginning there are both incorrect. While ice is indeed mechanically anisotropic, and bulk anisotropy results from anisotropy of the grains, the development of fabric is not related to this anisotropy in such a simple way. Sometimes fabrics orient favorably to the strain direction, but for some of the most common fabrics observed in ice sheets the opposite is true; beneath divides, where the stress state is uniaxial compression in the vertical, vertical single maxima form, but compression is thought to be harder parallel to the c axis than perpendicular to it. This statement perhaps belies a misunderstanding of the multiple processes contributing to fabric development, among which on migration recrystallization is thought to have the effect described here—and migration recrystallization does not dominate fabric development everywhere (or even most places). Moreover, it is unclear what "c-axes orthogonal to the strain" means, given that strain can act in multiple directions—in the case of compression, as already mentioned, the c-axes tend to orient parallel to the direction of maximum compression.

The short introductory summary given in these two sentences was indeed very much constructed to fit the girdle fabric scenario as encountered in the ice instrumented by IceCube. To avoid generally incorrect statements we propose the following alternative introduction and would be grateful for feedback:

> As a hexagonal crystal it will most readily deform as shear is applied orthogonal to the c-axis (crystal symmetry axis, normal to the hexagonal basal planes), leading to slip of the individual basal planes (McConnel, 1891). In polycrystalline ice the crystals effectively re-organize themselves to minimize the stored strain energy, resulting in non-isotropic / preferential

c-axes distributions and a bulk anisotropic viscosity (Faria et al., 2014b). The effects of recrystallization are experimentally most commonly observed as a crystal orientation fabric through the use of polarized light microscopy on thin sections of ice core samples (Alley, 1988; Wilson et al., 2003). In this work we only consider scenarios where c-axes are distributed isotropically (uniform fabric), are aligned in a single direction (unimodal fabric) or lie in a plane (girdle fabric). The later is of primary importance for the studied ice.

> 48: The wavelength belongs in the previous paragraph describing the observation

We would like to keep the quantitative number here (to contrast to the average grain size), but appreciate the point that the wavelength is a vital characteristic of the observation and will also add it in line 41.

> 82-82: "in addition to photons...mostly photons" is very confusing

We don't understand this comment. "Mostly photons" does not appear in this paragraph. Maybe *protons* was misread as *photons*?

> 147: "Ice layers" asks for confusion considering that the ice physically has layers that can exist on similar spatial scales. The terminology should distinguish between these and annual or radar layers. Slices? Packets?

Please see the response to general comment 3.

> 167: This section seems mis-titled, or at a minimum the title is not helpful. As I see it, this section just describes inference of isotropic properties of ice as an optical medium—so why not say that?

The section is meant to introduce the available data (*3.1 LED calibration data*) and means to analyze it (*3.2 Photon propagation simulation* and *3.3 Likelihood analysis*) which are common to the isotropic ice properties summarized in *3.4 The South Pole Ice Model (SPICE)* as well as the inference of anisotropic properties later in the manuscript. "IceCube as a laboratory for glaciology" may still be too sensational of a title and we propose to change it to "Deriving ice optical properties from LED calibration data".

> 170: I do not see why this should be true; e.g. for absorption there is no requirement to measure exactly the e-folding distance, rather than calculating it from the absorption over a shorter distance

While this is generally true, the distance scales involved here do not make this a viable measurement. Consider for example a full $1\,\mathrm{m}$ ice core segment with a representative absorption length of $200\,\mathrm{m}$. Over this distance only $1 - \exp\left(-1/200\right) = 0.5\%$ of the light is absorbed, which is very challenging to resolve, taking into account experimental uncertainties (primarily coupling of the light source and sensor to the ice core). To our knowledge no such measurement has ever (successfully) been performed.

> Figure 2: is this relevant? For this paper, simply saying that you have a photomultplier, LEDs, and associated electronics seems sufficient

Agreed. This Figure will be removed.

> 184: Ice realization could use more explanation. There is more detail below, but here perhaps just "hypothesized optical properties and ice-crystal orientations"

We will adopt this wording.

> 234: I find the idea of an "error rate" to be confusing here. Does this mean there is one incident photon not related to the LED every 2 ms?

The "noise rate" refers to electrical signals at the PMT output which are indistinguishable from the signal of a single photon hitting the PMT although the sensor is in total darkness. 500 Hz, so indeed one on average every 2 ms, may sound devastating, but since we are only considering a signal window of 1.5 us, the probability of a single photon like noise signal occurring within the signal window of each LED flash is smaller than 1/1000.
Since the sentence seems to be causing confusion and the noise modeling isn't actually important to know about here, we propose to just drop the sentence.

> 270: Layers here are still poorly defined. I suggest describing them as annual layers and noting that radar reflections result from contrasts in dielectric properties, which are generally assumed to be isochronous (although not necessarily annual).

We propose to change the sentence from
*One relevant complication is the layer undulation over the footprint of the array.*
to
*One relevant complication are the undulations of layers of equal optical properties over the footprint of the array.*

> 281: This feels incomplete: I am left unclear as to whether any effect of this tilt is included in the modeling. We need a description of whether this fit into anything above and a preview of where it will become relevant below.

The tilt is included in the modeling/simulation used for this work, as it is required to achieve a reasonable data-simulation agreement even before considering the anisotropy.
Tilt as implemented accounts for the depth shifts of isochrons at different lateral locations in the detector, but does not actually account for the layers being sloped. But we did check that this can not induce an anisotropy like effect.

The tilt is here primarily introduced as it is a topographically related effect (the ice flows down the valley, while the layers curve up the hill slopes) and makes referencing directions easier. To avoid confusion we will add a mention that in this work tilt is included as described in Aartsen2013.

There is one technical complication which is mentioned but not elaborated upon. Fitting the depth-dependent crystal size is, for primarily technical reasons, done in tilt-corrected depth instead of absolute depth. This likely has only a small effect and it is (at least to us) not clear which of the two depths dictates the crystal properties to start with. We will add a sentence on this in the outlook section.

> 286: Rather than naively, which is vague, just state the assumption (i.e. that the transmission medium has only isotropic dielectric properties).

Yes this should be specified. Since in the optical regime pure ice was thus far considered optically perfect the dielectric properties were not foremost on our mind. We propose the following wording

instead:
*Given the optical modelling discussed so far, the amount of light...*

> 291: repetitious

Indeed. We propose to simplify the sentence from
*This ice optical anisotropy was first seen in 2013 (Chirkin, 2013d), and is here called the ice optical anisotropy.*
to
*This ice optical anisotropy was first discussed in 2013 (Chirkin, 2013d).*

> 310: The elevation angle is not difficult to obtain from thin sections on an ice core; the full 3d orientation of each individual c-axis is automatically analyzed. Perhaps this should say that the tilt axis is difficult to measure in an ice core, due to the uncertain orientation of the core.

Agreed. This is an important distinction. We accordingly propose to change the sentence from
*As this is a parameter which is hard to accurately obtain from ice core data, it may be further investigated in the future.*
to
*As this is a parameter that is difficult to obtain from ice cores as their in-situ orientation is often not retained, it may be further investigated in the future.*

> Section 4.3: This is almost all repetition of published work, and the level of detail is unecessary—it should be sufficient to show the curves in Figure 8, point out that a factor of 11 is unreasonable for the absorption anisotropy, and say that the fit was mediocre when only modifying the directional scattering and absorption.

Agreed. This is a good place to shorten the manuscript, without loosing context. Please find a suggested update to section 4.3 below (getting rid of all math details and already saving a full page):

Following the paradigm that ice optical properties are driven by Mie scattering on impurities, early attempts tried to model the anisotropy through directional modifications of absorption and scattering. In the original parameterization presented by Chirkin(2013d), it was argued that due to time and space reversal symmetries the absorption length and geometric scattering length cannot be direction dependent. Therefore the anisotropy was implemented as a modification to the scattering function, the only remaining Mie scattering parameter. This effectively results in a change of the effective scattering coefficient as a function of the propagation direction. Photons propagating along the flow axis experience less scattering than photons propagating along the tilt axis or inclined from the horizontal.

While not derived from first-principle Mie calculations, the parametrization was justified to be a plausible result of elongated impurities becoming preferentially aligned by the flow and thus introducing a direction dependence to the scattering function. While several glaciological studies (Potenza et al., 2016; Simonsen et al., 2018; Gebhart, 1991) explore the shapes of impurities, elongations for different impurities are not well established, nor is there to our knowledge any evidence for elongated impurities becoming oriented with the flow.

An evaluation of the data-simulation agreement is shown in Figure 8. It shows summed photon arrival time distributions for all nearest emitter-receiver pairs, roughly aligned along and perpendicular to the ice flow for a variety of anisotropy models and the employed flasher data. The scattering-based anisotropy model results in more intensity being observed along the flow axis. However, there remains substantial disagreement between the model and the observed

data. As scattering is reduced in the flow direction light arrives earlier on average. The resulting change in the rising edge position is strongly penalized in the fit and limits the amount of intensity that can be recovered. To reduce the shift of the rising edge, a directional modification to Mie absorption was considered as an alternative by Rongen (2019). A factor 11 modulation of the absorption coefficient was required to fit the data, which seems unphysical. As evident from Figure 8, this model results in a delayed rising edge for propagation along the flow direction as desired, and did result in an improved data description compared to the scattering based model described earlier, but is also unable to fully match the intensity difference to data.

To conclude, while resulting in partially successful effective descriptions, directional modifications to Mie scattering or absorption cannot reproduce observations nor are such modifications well motivated on first principles.

318: unclear if this applies only to Chirkin 2013d or the present work as well-defined

This modification of the scattering function is unique to the model developed in Chirkin 2013d and not part of the birefringence modelling proposed here. As the new section 4.3 (see above) does not go into this detail, we do not see any further action necessary.

385: Add a reference to Petrenko and Whitworth here

Will do.

454: Usually the glaciological literature refers to a single Woodcock parameter, $\log(S_1/S_2)/\log(S_2/S_3)$; I assume that these parameters are the numerator and denominator of that fraction, but it would be helpful to state that explicitly

Agreed. We propose to change the wording from *based on Woodcock parameters* to *based on the Woodcock parameters* $\log(S_1/S_2)$ *and* $\log(S_2/S_3)$.

463–475: I am unclear as to how this incorporated in the results, given the lack of data. It becomes clear much later, but we need a preview

Would it be sufficient to state after line 475 that *Both fabric and grain shape are not directly taken from ice core data, but left as free parameters in the fit (section 6.4)?*

Figure 11: I am unclear on whether perpendicular to the picture is the same as perpendicular to the picture. i.e., it would be helpful to say that photons get "emitted" initially into/out of the page. I also think the description of the axes is unclear—as best as I can tell, this is a histogram of the normalized components of the direction vector? Why not use something other than n, considering that it has other meaning elsewhere?

We propose the following updated caption:

*Example diffusion patterns after photon propagation through 1000 crystals (roughly equivalent to 1 m) with a perfect girdle distribution of c-axis orientations. The initially emitted photon direction is perpendicularly out of the picture, with an opening angle to the flow as indicated. The figures histogram the final direction vectors of many photons. The change in diffusion (width of the distributions) as well as the subtle effect of photon scattering towards the ice flow (towards the right) can be seen.*

$n$ is indeed ambiguous. Given the new caption, axis labels do not seem necessary.

> Figure 12: Ice flow direction should be indicated. Perhaps this should come before Figure 11, to give the reader some intuition?

The ice flow direction will be added. We are happy to swap the figure order.

> 488: missing punctuation and article/noun mismatch

Thanks. This will be fixed from
*The same quantitative behavior as described above is reproduced, however this approach does not allow for a flexible configuration and is slow to simulate a reasonable photon statistics*
to
*The same quantitative behavior as described above is reproduced. This approach however does not allow for a flexible configuration and is slow to simulate reasonable photon statistics.*

> Section 5.3: Except for the directionality of the effect, the rest of this section could come much earlier (even in the introduction) and give the reader useful context.

We struggle to move the entire section, as it does require knowledge about the diffusion mechanism and its relevant parameters, which is only provided in the previous section. We thus propose to leave the section as is and hope that the changes proposed to the introductory sections are sufficient to give the added context.

> 529: I count 8 parameters—or should Eq 24 be two equations, one for x and one for y

Equation 24 is indeed a short-notation for two independent sets of parameters, one for the x and one for the y diffusion. We will make this clear by giving two independent $\sigma_x$ and $\sigma_y$ equations.

> 533: Language is odd—update sounds computational but the rest of the sentence sounds physical

Since this refers to the simulation, it is indeed a computational aspect/limitation. We propose to reword this from *During normal photon propagation ...* to *During photon propagation simulation ...*

> Figure 14: Top y-axis label is mission. I guess the panels on the diagonal are actually different than the rest? It is pretty incomprehensible as-is—perhaps some separation between those panels and the others, and something clear in the legend, would help.

The diagonal panels show marginalized LLH contours for the parameter in a given column. Since the LLH value is already encoded in the color of the points for which an axis is given and since there is no clean way to show axis labels for the other diagonal panels but the top one, we opted against a label here. Since no quantitative results are actually derived from this exemplary plot, we suggest to simply remove the diagonal panels resulting in an overall less messy plot.

> 573: we need a definition of pre-fits

Yes. We propose to add the following sentence prior to line 573:

This is done through pre-fits, which either vary all parameters for a single exemplary layer or fit the depth dependence of a given parameter while keeping all other parameters fixed.

> Section 7: These results are really nice, and so I was surprised that this section was so short. As mentioned above, I think more of a traditional paper structure (where there is less history and these are the results) would be beneficial.

To us the primary result of this work / message of the paper is the newly gained conceptual understanding about light propagation in the peculiar glacial ice micro-structure, rather then the resulting ice model. Yet as also requested by the over reviewer we propose to expand to the discussion of the depth dependence to:

> The overall grain size of ~1mm and the increase in size at larger depths, where ice crystals are generally larger, are as generally expected and measured in glaciology (Laurent et al., 2004; Alley et. al., 2021). In addition an anti-correlation between crystal size and impurity concentrations, as mapped by optical properties can be observed. This follows the expectation that impurity related processes such as impurity drag hinder grain growth (Durand, 2006). [...] Averaged over all instrumented depths light diffusion in the birefringent ice polycrystal amounts to an effective scattering coefficient of $2.47 \cdot 10^{-2}\,\mathrm{m}^{-1}$, accounting for on average $\sim 8.5\%$ of the total scattering present in the ice. The comparatively strong isotropizing effect of Mie scattering also explains why the intensity on the tilt axis is never fully depleted.

> 686: The use of such tensors dates at least to Love, 1944 "A Treatise on the Mathematical Theory of Elasticity, 4th ed." and presumably earlier (it has a long history in elastics and fibers). In glaciology, it dates at least to Castelnau et al., 1996, "Viscoplastic modeling of texture development in polycrystalline ice with a self-consistent approach: Comparison with bound estimates."

The Scheidegger(1965) paper is 31 years earlier than Castelnau(1996). As he is the usual reference in glaciological literature, we propose to change the sentence to the following:
*This ensemble of vectors can be represented via the matrix [Scheidegger(1965)].....*

> 698: This is not true, as implied below by the acknowledgment that you cannot obtain a true c-axis distribution from these two parameters

Yes "all" is too strong a statement here. We suggest to change to the following:
*Woodcock(1977) realized that many commonly encountered fabric states can be visualized...*

---

## Author Response (AR2)

**Reply to the re-submission report for the manuscript titled "In-situ estimation of ice crystal properties at the South Pole using LED calibration data from the IceCube Neutrino Observatory" received on the 14th of September 2023**

The IceCube Collaboration

September 30, 2023

**Comments from the editor**

> Dear authors,
>
> Both reviewers and I agree that the manuscript has improved considerably and that it should be accepted for publication in The Cryosphere. We also agree that it will be a pity that such a interesting paper may not be as accessible as it could have been.
>
> There are obvious reasons for that. The paper deals with a topic not common in glaciology, and it is long and technical. However both reviewers are pointing at making even more accessible the introductory sections of the paper by improving terminology and language.
>
> I ask the authors to step back for a second: no new runs, figures or equations, but edit the introductory sections of the manuscript ( Abstract and Sections 1 and 2) so that they are more readable. The reviewers have given several suggestions in that direction.
>
> Thank you again for your efforts, Carlos Martin

We would like to thank the editor for the timely evaluation of the reviewers comments, his patience in the sometimes slow editing process and the overall encouraging words regarding the evolution of the manuscript.

The introductory sections have been reworked as detailed below and we hope to have now reached an overall satisfactory manuscript.

**Comments from Anonymous Reviewer #1**

> Dear authors,
> The authors considered many points that I raised last time. I found the paper excellent. Very frankly, I believe that not many readers can follow the entire details of the paper, as subjects are deep and complex. In addition, as for the numerical simulation, readers must just trust the given results. I could read it from a viewpoint of a glaciology scientist. I must confess that very details of the physical calculations were hard to follow. For these points, I let further criticisms be done by future readers. The section "Outlook" became more sound; unknown or unclear points were described concretely. I will write some comments that I ask to the authors to consider before finalizing the paper. They are basically minor points on introductory part related to glaciology.

We would like to thank the reviewer for the time invested, the encouraging words and the additional comments, which we hope to have adequately addressed.

> L1-L2: I did not understand the grammatical structure of the sentence.

Sentence has been split in two.

> L22: A term recrystallization suddenly appeared. Basically, crystal rotation due to strain and recrystallization is related but different phenomena. I guess that the authors wanted to say non-isotropic c-axes distribution here. If it is so, please rephrase.

This was also pointed out by the second reviewer, who enforced that the IceCube side is probably dominated by roation instead of recrystallization. The paragraph has been modified to avoid potential errors and to de-emphasize the importance of the type of mechanism leading to a non-isotropic fabric. Please see the new paragraph in the response to the second reviewer.

> L24-25: The authors' expression seem complex. I believe that the expression will confuse readers. An expression unimodal fabric is new for me. At the same time, I feel the expression is a bit tricky. Both single pole fabric and girdle fabric are caused by uniaxial strain (from compression and extension). With a view point of a-axis, single pole c-axes fabric is girdle type a-axes. I feel that single pole fabric or girdle fabric is good enough. The expressions in this paper makes meaning unnecessarily complex.

We watned to borrow the term from statistics, referring to a distribution having a single mode/local maximum, as is the case in axis space for single pole fabrics. To conform with the more common terminology single pole has been adopted here and in most other instances, with some exceptions in Appendix B where the statistical term is more appropriate.

> L26: Again, the term "recrystallization". Was it intended to discuss recrystallization as used in crystal physics? Recrystallization is related to molecular diffusion, grain growth, grain boundary migration or polygonization. Were these intended in the manuscript? Or do you intend to mean evolution of microstructures and ice fabrics? Please clarify. Possibly, misunderstanding is going on.

The possible connection of grain sizes and shapes to the fabric evolution has now been removed so to not include any potentially false/misleading information.

> L26: "Alternatively" I did not understand the context. What do you compare, method or phenomena?

This was intended to compare methods. Simply dropped the "*alternatively*" here.

> L25-L32: I find that negative tones for introduction of ice core studies drilled for climate research purposes. Sampling volume limitation is OK. Core orientation can be deduced from ice fabric or other methods. Grain volume or elongation can be deduced from analysis of 3D thin sections. Ground truth is very important. I prefer more positive tone.

Deducing the core orientation from the fabric (while certainly good to first order) becomes a circular argument when trying to understand the interplay between present day flow and fabric. While we are certainly not intimately familiar with the glaciological literature, we have in our studies also so far not encountered "3D thin sections." Information on this would be appreciated.

Regarding the discussed section, the tone was certainly not meant to be negative and we absolutely agree on the value of ground-truth information. Yet we believe the limitations as outlined to be fairly described and these were all limitations which we also encountered in modeling the ice optical anisotropy.

The wording of the paragraph has been slightly altered and it now reads as:

> While ice core analysis uniquely delivers ground-truth information, it is limited by its small sampling volume, and often unable to resolve the absolute direction of fabric orientation as the core orientation is not preserved in the drilling process. Volumetric quantities such as grain volumes and shapes are generally not directly accessible through the commonly employed techniques. Grain sizes and elongations evaluated through the microscopy of thin slices cut from ice cores in turn often depend on the sample plane.

> L31: In firn, grain boundaries are still undetectable by X-ray CT. Thus, it does not seem an exception. We can observe just boundaries of pore structure and ice matrix. It is so in bubbly ice zone (100-1300 m depth), too.

Fair enough. X-ray only sees density fluctuations and even in the firn it may not be single but grains with air pockets in between. We have omitted the X-ray tangent.

> L50 and at many points in the manuscript: What do you mean with a term "Grain boundary crossings"? Is it your original term? Is the widely used term "grain boundaries" not suitable? Do you specifically mean triple junctions?

"*Grain boundary crossings*" was meant to denote the process of a photon traversing/crossing the grain boundary. The term was replaced in all instances, in this case simply switching to "*... is refracted or reflected on many grain boundaries ...*".

> L51-53: Did Chirkin and Rongen investigate effects from shapes of grain boundaries? Please specify.

Yes, but only on length scales larger than $400\,\text{nm}$. On the length scale relevant to individual refractions/reflections, that is the wavelength of light, the grain boundaries are assumed to be simple planes (see section 5.2). The macroscopic shape is accounted for in the probability density of encountered grain boundary orientations, as elaborated on throughout the manuscript and detailed in Appendix C. To

emphasize the connection between the grain shape and the distribution of grain boundary orientations the paragraph has been updated as follows:

> *Thus, the spacing of grain boundaries and the distribution of encountered grain boundary orientations, both of which are a function of the average grain shape, must be accounted for in addition to the fabric.*

> L51: Light propagation is along the preferred orientation of the a-axes. It seems meaningful to let readers know about it.

We do not understand this comment, could the referee clarify? The model accounts for light propagating in any direction as is also the case in reality. (With the initial emission being random and scattering being present.)

> L5̶1̶7: The term "crystal realizations" is not a standard term in physics but might be used in specific contexts to refer to particular instances or examples of crystalline structures. Please specify your intended meaning to readers.

Changed here to "*configurations/realizations*" to specify the intent going forward.

> L150: Soot does not have volcanic origin. Please find a review paper below. Bond, T. C., et al. (2013), Bounding the role of black carbon in the climate system: A scientific assessment, J. Geophys. Res. Atmos., 118, 5380–5552, doi:10.1002/jgrd.50171. If you still claim that soot is from volcanic eruptions, please provide convincing reference paper.

Thank you for noticing this. This was an oversight. Volcanic ash layers are detectable for example with the dust logger but were not meant to be mentioned in this context. The "*(volcanic)*" has been deleted.

> L492: Crystal dielectric tensor can change both gradually and also abruptly. Please find Figure 7 in the following paper.
>     Saruya, T. et al.: Development of crystal orientation fabric in the Dome Fuji ice core in East Antarctica: implications for the deformation regime in ice sheets, The Cryosphere, 16, 2985–3003, https://doi.org/10.5194/tc-16-2985-2022, 2022.
> In terms of radar sounding, it will be a topic of further discussions.

That is an interesting plot. We have omitted the statement about gradual changes for correctness.

> Section 2.3 Size distributions of sea salts and terrestrial dusts in ice sheets are well discribed in the paper below. It will give more concrete information to readers. Oyabu, I., Iizuka, Y., Kawamura, K., Wolff, E., Severi, M., Ohgaito, R., et al. (2020). Compositions of dust and sea salts in the Dome C and Dome Fuji ice cores from Last Glacial Maximum to early Holocene based on ice-sublimation and single-particle measurements. Journal of Geophysical Research: Atmospheres, 125, e2019JD032208. https://doi.org/10.1029/2019JD032208

There is a LOT of literature on particle sizing of impurities in ice cores. Since in this context only the conceptual connection to the scattering asymmetry as well as order of magnitudes are of relevance, we would prefer to not include papers of specific measurements here.

**Comments from Reviewer #2, David Lilien**

> This is my second review of this paper, and I would like to thank the authors for carefully responding to all my comments. I still think it will be a valuable contribution well-suited to The Cryosphere.
>
> I find the manuscript to be significantly improved, although I still struggled a bit with what I view as a lack roadmap and lack of clarity on what exactly was done in the present work (or the proceedings which it collects for a geophysical audience). As the paper goes on, I think that this improves somewhat, and so this general comment focuses on the abstract and sections 1 and 2.

We would like to thank the reviewer for the time invested, the encouraging words and the additional comments, which we hope to have adequately addressed.

> I recognize that it is not really my role as a reviewer to demand grammatical changes that are a matter of preference, but I think that one reason that clarity is still an issue is the almost exclusive use of present and present perfect tenses and the heavy use of passive voice. Those choices make it very, very difficult for somebody without intimate knowledge of the various efforts by the IceCube team to know what happened when and what is really part of the current ice model or current study. I do not think there is anything technically wrong with using present tense, but it causes confusion. Take, for example, line 3: does "has been examined" mean we will read about it here or that it was in previous work? Or line 60: between two sentences I think there is a transition between a pre-existing truth and a roadmap of the present paper, but I would not have known that without having read the whole paper previously. Or 682, where the authors write "could" to describe something that they actually did. I think one way to address the lack of clarity is by hewing closer to common scientific use of tenses (i.e., present for pre-existing truths, past for things that happened in the past, including work that has been completed for the present paper). That is one suggestion for improving clarity, and I imagine there would be a number of other ways to reach the same goal of clarifying what is "this paper" and what not. However, if this suggestion is not taken, I think there still need to be substantial changes in the abstract and sections 1—2 to distinguish between what is happening in this paper and what was previous knowledge. That is not to say that this should be difficult, a careful edit that considers how little most glaciologists know about Ice Cube should make the start of the paper more readable.

We would like to thank the reviewer to be able to pinpoint specific grammatical structures that he found to hinder the accessibility of the manuscript. We found the specific recommendation (active voice and present for pre-existing truths, past for things that happened in the past, including work that has been completed for the present paper) thought provoking. In particular, active voice has traditionally been somewhat of a taboo. Re-examining some recent literature we agree that this has changed and probably improves clarity overall. While we hesitate to embrace a full rewrite of the manuscript for voice and tense, for fear of adding new inconsistencies, we will certainly take the advice for future publications.

Regarding the specific examples of confusion raised by the reviewer:

- Line 3: The wording has been changed to: "*We here examine birefringent light propagation through the polycrystalline ice microstructure as a possible explanation for this effect.*"

- Line 60: The overall effect was first described in Chirkin and Rongen, 2020, while the parametrization and application in photon propagation is described in Rongen et al. 2021a. To de-emphasize a temporal relation "*The new simulations, ...*" in line 63 has been changed to "*These simulations,*

...", the tense has been changed to be self consistent with the prior use of present perfect and a further citation of Rongen et al. 2021a has been added.

- Line 682: Yes this could easily be misread as "to be done in the future". The wording has been changed to: *"This has been improved upon by ..."*

The following further clarifications have been added:

- Line 75: Changed *"prior to this work"* to *"prior to the discovery of the ice optical anisotropy"*

- Line 150: Added a further reference to Ackermann et al., 2006.

- Line 180: Changed *"This leaves 6 global parameters ... to be determined."* to *"Thus 6 global parameters ... are required to describe the layered ice properties."* so to not give the impression that these parameters are of primary interest in this manuscript.

> At this point, other than a couple of the detailed comments below, I do not have qualms that would prevent publication, but I think it would be a missed opportunity not to further improve the clarity of the manuscript.
>
> L18: The introductory phrase is misleading—the reason for ease of deformation is not the crystal shape

Fixed by omitting the shape:

*The viscosity of an individual ice crystal strongly depends on the direction of the applied strain and it will most readily deform as shear is applied orthogonal to the c-axis*

> L20: This is still incorrect; it describes the mechanism of migration recrystallization but not lattice rotation. At IceCube, ice is cold and flow is reasonably slow, so lattice rotation is the likely cause of fabric development (the circa 1990 Alley papers are still fine references for this). That is to say, the crystals are not reorganizing to minimize strain energy, they are simply being rotated by the physical deformation changing place; to the extent that they minimize strain energy, in this case that is incidental.

Thank you for elaborating on this. This actually sheds some light on the apparently missing quantitative relationship between fabric strength and grain elongation, which is something we have been looking for in the literature for a while... The paragrah has been rephrased so not to be explicit on the fabric forming mechanism (as it does not directly matter for the optical properties) and now reads as (without citations):

*Ice flows under its own weight, either through basal sliding or through plastic deformation, which is mediated by the deformations of individual grains as well as interactions between grains. The viscosity of an individual ice crystal strongly depends on the direction of the applied strain and it will most readily deform as shear is applied orthogonal to the c-axis (crystal symmetry axis, normal to the hexagonal basal planes), leading to slip of the individual basal planes. In polycrystalline ice subjected to strain the crystals may undergo lattice rotation or recrystallization, both of which result in non-isotropic c-axes distributions and a bulk anisotropic viscosity.*
*In this work we only consider scenarios where the c-axes are distributed isotropically (uniform fabric), are aligned in a single direction (single pole fabric) or lie in a plane (girdle fabric). The*

*later is of primary importance for the studied ice.*
*The crystal orientation fabric is experimentally most commonly observed through the use of polarized light microscopy on thin sections of ice core samples.*

> L60: Combination of passive voice and present tense with a mid-paragraph transition in topic make this hard to parse

Please see the response above.

> L167-180: This is really the start of the development of a model of this particular parcel of ice, whereas the rest of the section is more about physical properties of glacial ice in general. I would suggest making these two paragraphs a separate section, since they do not really describe "ice as an optical medium"—maybe consider moving them to 3.2 if not.

Point taken. In isolation these paragraphs probably fit best into section 3.4, yet the concepts introduced here are already required going into section 3.2. With section 3.2 being dedicated to light curves and photon propagation, we have decided to give these paragraphs a new subsection titled "*Describing the depth dependence*".

> L320: I think that this is a little deceptive—the azimuthal orientation of a core is not retained but the elevation angle is generally known.

While still potentially of interest, the statement as given was indeed not detailed enough, nor is it actually vital to the rest of the manuscript. The sentence has been removed.

> L533 and 570, maybe elsewhere: 2 and 3 should be subscripts on the eigenvalues

This has been fixed.

> L659: second clause has no subject

This has been fixed.

---

## Author Response (AR3)

**Comments for the production files for**
**"In-situ estimation of ice crystal properties at the South Pole using LED calibration data from the IceCube Neutrino Observatory"**

The IceCube Collaboration

November 7, 2023

With regard to the previously reviewed version the following changes have been made:

- The editorial support criticized figures 6 and B1 for not being color-blind accessible. For these figures the color pallet has been changed and distinct line styles have been added.

- Switched from "Figure" to "Fig." when referencing figures as outlined in the journal guidelines.

- All figures with multiple panels have been merged. Due to the change to the 2-column layout a number of multi-panel figures have been rearranged.

- For Martin Rongen his current affiliation (Erlangen Centre for Astroparticle Physics, Friedrich-Alexander-Universität Erlangen-Nürnberg) has been added.

We further note a formatting problem in the bibliography to be addressed during typesetting:
Six-digit 'page numbers' seem to be broken up into two 3-digit numbers. For example, in the first reference, A. Aab et al., the bibliography reads '125 121 106' The volume number is 125, but the page number as specified in the BibTex-record should be '121106'.